# LLMs Can Leverage Graph Structural Information in Text-Attributed Graphs

## Abstract

A recurring claim in recent LLM-as-predictor work on text-attributed graphs (TAGs) is that in-context learning (ICL) benefits mainly from the textual attributes of neighboring nodes (often via homophily), while general-purpose LLMs cannot reliably exploit graph structure—especially edge direction and local topology. This paper re-evaluates that claim by asking a focused question: Can general-purpose LLMs genuinely leverage graph structural information in TAGs via ICL, once we remove confounding factors and provide an architecture explicitly designed for structural reasoning? We first introduce controlled neighborhood rewiring tests that keep node texts and label distributions fixed while perturbing structure. Across seven LLMs and four low-homophily WebKB graphs, both first-order flipping and two-hop extreme rewiring consistently degrade accuracy ($-2.06 \sim -23.15\%$ average relative drop), demonstrating genuine structural sensitivity. After flipping, structural sensitivity strongly increases with model capability, and the performance advantage of stronger models arises primarily from correct structure rather than better text-only processing. We further show that apparent "structure misuse" in weaker models can be corrected by adding explicit step-by-step instructions. The previous claim is due to confounding factors—the traditional ICL framework lacks a dedicated mechanism for graph structure reasoning and handling lengthy multi-hop neighborhood contexts, rather than the inherent nature of LLMs themselves. Motivated by these findings, we propose the Text Attributes Passing Thoughts Network (TAPTN), an edge-aware, MPNN-like ICL framework that iteratively summarizes multi-hop neighborhoods using a structure-aware template and self-generated instructions. TAPTN substantially outperforms zero-shot CoT and GraphICL-style baselines on five TAG datasets by at least $+13.98\%$, especially on malignant heterophilic graphs (with $+15 \sim +25\%$ gain), and when used to produce structurally enriched texts for downstream fine-tuning, achieves performance competitive with state-of-the-art GNN pipelines. Collectively, the results establish that LLMs can exploit structure information in TAGs through ICL at a level competitive with SOTA GNNs under a controlled comparison, once equipped with an appropriate architecture that mitigates the confounding factors.

## 1 Introduction

Text-attributed graphs (TAGs), in which each node is associated with a rich textual description and edges encode relations between entities, are a fundamental abstraction for many real-world systems, including citation networks, e-commerce platforms, and financial transaction networks. A central task on TAGs is node classification, where the goal is to predict node labels by leveraging both node text and graph structure. Traditionally, LLM-based methods rely on fine-tuning (Zhao et al., 2025; Tang et al., 2024; Ye et al., 2024). Recently, large language models (LLMs) have emerged as powerful universal learners, and a growing body of work has explored whether LLMs can replace or complement graph neural networks (GNNs) on such graph reasoning tasks via in-context learning (ICL) rather than task-specific training due to its domain transferability and zero-shot learning capability. To integrate graph structural information, open-source LLMs like Llama can fuse Graph Neural Networks (GNNs) (Sun et al., 2025; Tian et al., 2023; Qin et al.,

2023; Wang et al., 2024), while more powerful closed-source models like GPT-5.2 use In-Context Learning (ICL) for direct predictions (Chen et al., 2024; Guo et al., 2023) or additional insights (He et al., 2023).

Existing empirical evidence, however, has led to a prevailing misconception: LLMs are believed to benefit from neighboring nodes in TAGs primarily because of homophily in textual attributes with edges serving primarily as a retrieval mechanism for additional similar texts, rather than any genuine utilization of graph structure. Huang et al. (2024) systematically studied whether LLMs "can effectively leverage graph structural information through prompts," concluding that while LLMs do benefit from incorporating neighborhood information, the gains are strongly correlated with local homophily: nodes with neighbors sharing similar labels benefit most, and there is limited evidence that performance improvements stem from deeper structural reasoning beyond textual similarity. More recently, GraphICL (Sun et al., 2025) introduced a comprehensive ICL benchmark for graph reasoning with a unified prompt template that combines anchor-node text, task description, structure-aware neighbor texts, and labeled demonstrations. Their ablation studies show that the dominant source of improvement often comes from carefully selecting homogeneous neighbors and demonstrations (e.g., most similar or class-aligned nodes), whereas simply adding more structure-aware content yields smaller incremental gains.

Yet in many TAGs, edges and topological structures themselves are semantically meaningful. In financial transaction graphs, for example, specific topological patterns—such as multi-hop money-laundering chains, dense fraud rings, or temporal motifs—are strongly correlated with fraudulent behavior, even when the textual attributes of participating accounts are heterogeneous (Wei & Lee, 2025; Luo & Zhang, 2024; Tong & Shen, 2023). Similarly, in heterophilic graphs, neighbors often carry different labels, and what matters is not merely that a neighbor exists, but which role a neighbor plays in the local structure (buyer vs. seller, authority vs. hub, citing vs. cited). These observations suggest that graph structure information should be understood as more than "extra node texts": it includes edges and topological structures as relational evidence, providing context-dependent signals that determine how neighboring node information should be interpreted. If the misconceptions in existing research are corrected, when and how LLMs utilize graph structure information in in-context learning (ICL) are understood, and the necessary architectures to exhibit this capability are identified, then the application areas of graph learning based on ICL will be greatly expanded.

We argue that two limitations in current ICL-based, LLM-as-Predictor approaches conspire to mask LLMs' latent ability to use such structural information, rather than indicating a lack of this capability in the LLMs themselves:

- **Homophily-centric interpretation of neighborhood gains**: Because prior analyses largely rely on high-homophily graphs and aggregate neighbors by simple textual concatenation, improvements from neighborhood information can be explained away as coming solely from label- and text-homogeneous neighbors, rather than from any genuine reasoning over graph structure. Under these settings, ablation experiments based on structural perturbations are unlikely to eliminate this misconception, because the effect of homophilous semantic redundancy (similar texts with similar labels) with the effect of graph connectivity can be conflated, making it difficult to determine whether LLMs are actually leveraging edges as relational signals.

- **Lack of a dedicated mechanism addressing lengthy high-order neighborhood contexts**: Existing ICL prompts typically linearize large ego-networks by listing first- and second-order neighbors verbatim. This leads to long, noisy contexts in which fine-grained structural cues (e.g., which neighbor is connected through which relation, or how multiple neighbors form a motif) are diluted. Long-context studies show that LLMs tend to attend most strongly to information at the beginning and end of the context, while information in the middle is disproportionately ignored, a phenomenon known as "lost in the middle" (Das et al., 2023; Liu et al., 2024). As a result, subtle structural patterns encoded in the interior of long neighborhood descriptions are unlikely to be consistently exploited, even if they are present.

- **Lack of a dedicated mechanism for structural reasoning:** Existing ICL-based TAG methods rely on prompt templates that treat neighbors as additional textual evidence or as labeled examples,

but do not provide an explicit mechanism for representing and propagating edge information or multi-hop structure. Under this setting, as we found in Section 2, although LLMs can utilize graph structure information, some of this information is discarded or even misinterpreted, leading to the appearance that structural information is unhelpful or even detrimental to performance. Therefore, the correct way to utilize graph structure information should be to extract it from the LLM's parameter knowledge base and explicitly provide it to guide the LLM to consistently follow it.

In this work, we revisit the central question: **Can general-purpose LLMs genuinely leverage graph structural information in text-attributed graphs via ICL, once we remove these confounding factors and endow them with an architecture explicitly designed for structural reasoning?**

Our answer proceeds in three stages:

First, in Section 2, we re-examine the conclusion that "LLMs cannot effectively leverage graph structure via ICL" by revisiting the experimental setup that led to it. Rather than relying on high-homophily citation graphs with verbose second-order neighbor descriptions, we use directed, low-homophily datasets and design rewiring schemes that perturb first-order neighborhoods while keeping node texts fixed. This setting prevents good performance from being explained purely by treating neighbors as a bag of similar texts. Evaluating seven modern LLMs (including GPT-3.5, Phi, Gemma, Llama-3 family, and Qwen) on four heterophilic datasets, we find that all models exhibit systematic accuracy drops ($-2.06 \sim -23.15\%$ average relative drop), providing direct evidence that LLMs are capable, in principle, of using graph structure information when prompts do not conflate structure with homophily. Experimental results also show that the sensitivity of LLMs to graph structure perturbations is positively correlated with their capabilities, and that better utilization of graph structures is the main source of their performance gains.

Second, in Section 3, motivated by the above diagnosis, we propose the Text Attributes Passing Thoughts Network (TAPTN), an ICL architecture explicitly designed to expose and exploit graph structure for node classification. TAPTN is inspired by Message Passing Neural Networks (MPNN) (Gilmer et al., 2017): it iteratively extracts and aggregates neighborhood information in textual form, guided by LLM-generated step-by-step instructions, but processes only first-order neighborhoods at each iteration. Instead of simply concatenating all neighbor texts, TAPTN constructs structured, layer-wise summaries of a node's multi-hop neighborhood, which simultaneously incorporates the text attributes of neighboring nodes, edge semantics, and the semantics of specific topological structures. This design addresses the two major shortcomings of existing ICL-based node classification methods: it decouples multi-hop reasoning into a sequence of shorter, structurally grounded steps (mitigating long-context issues) and provides a dedicated mechanism for reasoning over edges and connectivity. TAPTN thus provides, to the best of our knowledge, the first ICL-based LLM-as-Predictor architecture for TAG node classification that is explicitly engineered to leverage graph structure, not just treating neighbors as undifferentiated extra text.

Third, in Section 4, we revisit the comparison between LLM-based architectures and GNNs from a structural perspective. We fine-tune a small pre-trained language model on TAPTN-enhanced text attributes and compare its node classification performance to state-of-the-art GNN baselines on both homophilic and heterophilic TAGs. Rather than asking whether a naive zero-shot prompt can match a fully supervised GNN, we ask whether, once equipped with a reasonably designed ICL architecture that foregrounds structural information, LLMs' structural utilization capability is still inferior to that of GNNs.

Our contributions are summarized as follows:

First, we show that prior negative or homophily-centric conclusions are largely artifacts of high-homophily datasets and verbose, poorly structured prompts. Under low-homophily, structurally perturbed settings, LLMs exhibit clear sensitivity to graph structure, even when node texts attributes are held fixed, revealing LLMs' graph structure exploiting ability. Under flipping rewiring settings, structural sensitivity is strongly and significantly correlated with model capability, indicating that stronger models derive more of their gains from edge-based reasoning rather than from homophilic neighbor texts.

Second, we introduce TAPTN, a structure-aware ICL architecture for TAG node classification addressing lengthy high-order neighborhood contexts and enabling dedicated structural information leveraging mech-

anism. Considering both homophilic and heterophilic TAG benchmarks, TAPTN consistently outperforms zero-hop chain-of-thought and GraphICL-style baselines by at least +13.98%. Ablation studies show that these gains persist after controlling for self-reflection and are amplified by explicit step-by-step instructions. This result indicates that with an appropriate framework like TAPTN, LLMs can indeed exploit graph structure information effectively.

Third, we show that a modest LM trained on TAPTN-generated textual representations achieves node classification performance competitive with state-of-the-art GNNs on both homophilic and heterophilic TAG benchmarks. This indicates that, when supported by an appropriate ICL architecture, LLMs' ability to exploit graph structure, including edges and local topology, matches or surpasses that of specialized graph neural networks rather than inherently weaker.

## 2 Are LLMs Really Unable to Leverage Graph Structural Information through ICL?

In this section, we revisit the claim that LLMs "cannot effectively leverage graph structural information through ICL and only treat neighbors as linear text," originally drawn from experiments on high-homophily citation graphs with long, verbose neighborhood descriptions. As discussed in the introduction, such settings make it difficult to distinguish whether performance gains come from (i) genuine use of graph structure (edges and their patterns), or (ii) graph homophily and textual similarity between neighbors. Moreover, long concatenated neighborhood descriptions are precisely the regime where LLMs tend to be "lost in the middle" and under-utilize information buried in the middle of the context.

Our goal here is to design an evaluation that removes these confounding factors and asks a sharper question:

**When we keep node texts fixed and modify only edges in low-homophily graphs, do LLM predictions change in a systematic way?**

A positive answer would provide direct evidence that LLMs interpret neighborhood information as graph structural data rather than as an unordered bag of texts, because any change in accuracy must arise from how the LLM interprets edge patterns.

### 2.1 Preliminary: homophily and the role of graph structure

To measure homophily for a graph $\mathbf{G} = \{\mathbf{V}, \mathbf{E}\}$, we define it as:

$$H = \frac{\sum_{i \in \mathbf{V}} \frac{\left|\left\{j | e_{ij} \in \mathbf{E}, y_i = y_j\right\}\right|}{\left|\left\{e_{ij} | e_{ij} \in \mathbf{E} \right\}\right|}}{|\mathbf{V}|}, \tag{1}$$

where $e_{ij}$ is the edge between node $i, j$, $\mathbf{E}, \mathbf{V}$ are the edges, nodes set of $\mathbf{G}$ respectively, $y_i$ is the label of node $i$. This formula measures the average proportion of a node's neighbors sharing its label. Table 1 presents the homophily values for each dataset. The citation datasets used by Huang et al. (2024) exhibit high homophily ($H \geq 0.63$), whereas the WebKB datasets we use (Cornell, Texas, Washington, Wisconsin) have substantially lower homophily ($H \leq 0.19$).

On high-homophily graphs, an LLM can achieve good node classification accuracy by effectively treating neighbors as extra similar texts and ignoring graph structure. This is exactly the mechanism highlighted in previous ablation studies (Huang et al., 2024; Sun et al., 2025), where performance gains are largely attributed to carefully selecting homogeneous neighbors and demonstrations rather than to sensitivity to edge patterns or multi-hop topology.

However, in heterophilic graphs, as neighbors often have different labels and graph structure carry semantics and determine how neighbor information should be interpreted, what matters is which node links to which through which relationship, not merely that a neighbor exists. For example, in WebKB, hyperlink patterns between student, faculty, department, and course pages encode roles and authority relationships that are not recoverable from page texts alone.

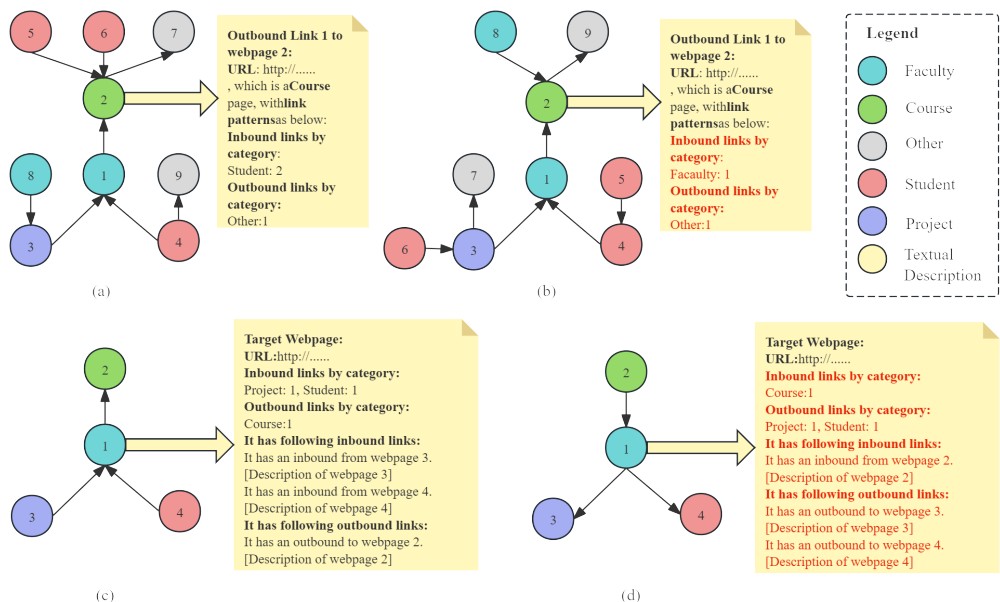

Figure 1: Illustration of "extreme" and "flipping" rewiring methods applied to a webpage neighborhood structure, with node 1 as the target node. Sub-figures (a) and (b) show "extreme" rewiring effects, while (c) and (d) depict "flipping" rewiring effects. Bold text highlights basic description components, and red text indicates changes post-rewiring. The target node's label is excluded from the second-order neighborhood description.

Table 1: Homophily of datasets used to evaluate the impact of neighborhood rewiring in this paper (upper half) and Huang et al. (2024) (lower half).

| **Dataset** | Cornell | Texas | Washington | Wisconsin | OGBN-Arxiv | Pubmed | Cora |
|---|---|---|---|---|---|---|---|
| **Homophily** | 0.1308 | 0.1448 | 0.1599 | 0.1869 | 0.6358 | 0.7924 | 0.8252 |

## 2.2 Experiments

To remove the confounding factors and reassess the core claim of previous research that LLMs "simply treat neighborhood descriptions as linear text" and cannot effectively leverage graph structural information through ICL, we design experiments with the following features:

- Use low-homophily graphs, where simple homophily-based majority voting is not sufficient.

- Manipulate edge patterns while holding node texts fixed, so any accuracy changes must come from how the LLM interprets changed connectivity (i.e., structure and relations), not from changed textual attributes.

- Avoid extremely long neighborhood prompts that bury structural cues in the middle of the context, which is known to reduce their impact due to "lost in the middle" behavior of LLMs on long inputs.

### 2.2.1 Datasets and Settings

We select four low-homophily WebKB (Ghani et al., 2001) graphs—Cornell, Texas, Washington, Wisconsin—with homophily between 0.13 and 0.19. These datasets consist of webpages from university CS departments, with nodes labeled as student, faculty, staff, department, course, or project. We exclude the "other"

category from evaluation to avoid diluting accuracy with pages whose hyperlink and content patterns are highly heterogeneous.

To reduce context length and avoid the "lost in the middle" effect on long prompts, we construct a GraphICL-style prompt template that include: (i) first-order neighbor (linked webpages) descriptions, including hyperlink direction, their URLs, content abstracts and labels; (ii) the number of second-order neighbors by category; (iii) link-count statistics (in/out degree). This template emphasizes graph structure (who links to whom by which relationship type in what pattern) rather than extra textual content.

To make it possible for the LLM to notice and process all relevant graph structural cues within a manageable context window while keeping node texts fixed across original and rewired graphs, we adopt and adapt the "extreme" rewiring from Huang et al. (2024) and introduce a new "flipping" rewiring that directly perturbs first-order link directions. Figure 1 illustrates these rewiring schemes and how they modify the neighborhood description in the prompt. Specifically:

- **Flipping**: We introduce a new method that reverses link directions of all first-order edges touching the target node. For example, if a student page originally links to a course page (outgoing), after flipping the course page has an incoming edge from the student page. This changes the incoming/outgoing pattern and role interpretation of each neighbor (who links to whom), while keeping the set of neighbors, their text descriptions and labels unchanged.

- **Extreme**: For each target node, we keep its original first-order neighbors fixed, but reconnect all its second-order neighbors to a single randomly chosen first-order neighbor. This rewiring dramatically changes the local 2-hop topology (e.g., funnels all paths through one neighbor) while preserving which nodes are in the 2-hop neighborhood and their labels. We represent second-order information only by category counts, not by verbose textual descriptions, to emphasize the change in path structure rather than added text.

These settings directly target graph structure as relational evidence: if the LLM were using neighbors only as an unordered bag of texts (homophily-driven), both extreme and flipping should have little impact, especially given that category counts and node texts remain fixed. Substantial performance changes would therefore indicate that the model is sensitive to how neighbors are connected and what role they play in the hyperlink structure.

To assess whether this structural sensitivity is model-specific or holds more generally, we conduct our experiments on a panel of seven widely-used LLMs spanning a wide range of capabilities (with their text Arena scores in LMArena leader board from 1224-1319), parameter scale (from 9B-72B), and providers: GPT-3.5-Turbo-0125, Phi-4, Gemma-2-9B, Llama-3-70B-Instruct, Llama-3.1-70B-Instruct, Llama-3.3-70B-instruct, Qwen2.5-VL-72B-Instruct.

### 2.2.2 Results

Table 2 shows the effects of the two rewiring methods on ICL-based node classification accuracy across 4 datasets and 7 models.

For flipping rewiring, all models show consistent performance degradation across the all WebKB datasets. Averaged over all models and datasets, the relative performance change is $-10.67\%$ ($p < 0.01$ with t-test). The strongest sensitivity is observed for Qwen-2.5-VL-72B, whose average accuracy decreases from 71.28% to 54.62%, a $-23.15\%$ relative drop, while the weakest sensitivity is observed for GPT-3.5-turbo-0125, where accuracy changes from 46.67% to 45.89% on average, i.e., $-2.06\%$ relatively. These drops cannot be attributed to losing homophilic neighbors or to changing the textual attributes of the context. Instead, they show that all tested models rely on directional structure (whether a node mostly points to certain page types or is referenced by them) as a useful signal for classification. When we invert this signal, performance suffers. The sensitivity to flipping rewiring significantly strengthens for more capable LLMs.

For extreme rewiring, we again observe universal degradation, but with a different profile. The average relative performance change over models and datasets is $-4.40\%$. Although it's noticeably smaller in magnitude

Table 2: Impact of neighborhood rewiring on node classification accuracy (%), where $\mathbf{O_F}, \mathbf{R_F}, \mathbf{O_E}, \mathbf{R_E}$ are accuracy before and after flipping and extreme rewiring respectively. With 54 of total 56 settings, rewiring significantly decreases accuracy, indicating LLMs incorporate graph structural information despite their intelligences, parameter scales or providers.

| Model | Arena Score | #Parameters | Dataset | $\mathbf{O_F}$ | $\mathbf{R_F}$ | $\mathbf{O_E}$ | $\mathbf{R_E}$ | $\mathbf{\Delta_F}$ | $\mathbf{\Delta_E}$ |
|---|---|---|---|---|---|---|---|---|---|
| GPT-3.5-Turbo-0125 | 1224 | ~20B | Cornell | 49.39 | 55.06 | 80.57 | 78.95 | +6.67 | −1.62 |
| | | | Texas | 43.87 | 39.53 | 69.57 | 64.82 | −4.34 | −4.75 |
| | | | Washington | 51.36 | 49.81 | 69.65 | 67.70 | −1.55 | −1.95 |
| | | | Wisconsin | 42.04 | 39.17 | 67.52 | 63.38 | −2.87 | −4.14 |
| Phi-4 | 1255 | 14B | Cornell | 51.52 | 43.15 | 63.31 | 58.07 | −8.37 | −5.24 |
| | | | Texas | 41.01 | 39.45 | 60.94 | 55.08 | −1.56 | −5.86 |
| | | | Washington | 53.01 | 52.63 | 69.17 | 60.53 | −0.38 | −8.68 |
| | | | Wisconsin | 49.84 | 48.60 | 65.73 | 60.75 | −1.24 | −4.98 |
| Gemma-2-9B | 1265 | 9B | Cornell | 49.39 | 43.95 | 71.77 | 63.71 | −5.44 | −8.06 |
| | | | Texas | 44.53 | 35.94 | 67.58 | 55.86 | −8.59 | −11.72 |
| | | | Washington | 54.51 | 50.38 | 72.56 | 63.91 | −4.13 | −8.65 |
| | | | Wisconsin | 60.44 | 51.41 | 76.01 | 71.34 | −9.03 | −4.67 |
| Llama-3-70B-Instruct | 1276 | 70B | Cornell | 73.79 | 52.02 | 82.66 | 83.06 | −21.77 | +0.40 |
| | | | Texas | 64.06 | 50.00 | 83.21 | 78.52 | −14.06 | −4.69 |
| | | | Washington | 67.67 | 58.27 | 84.96 | 77.82 | −9.40 | −7.14 |
| | | | Wisconsin | 71.65 | 59.81 | 84.42 | 83.80 | −11.84 | −0.62 |
| Llama-3.1-70B-Instruct | 1293 | 70B | Cornell | 77.02 | 55.24 | 83.87 | 83.06 | −21.78 | −0.81 |
| | | | Texas | 69.53 | 52.73 | 84.38 | 80.47 | −16.80 | −3.91 |
| | | | Washington | 70.68 | 60.53 | 81.96 | 77.82 | −10.15 | −4.14 |
| | | | Wisconsin | 74.45 | 57.94 | 85.05 | 83.80 | −16.51 | −1.25 |
| Qwen2.5-VL-72B-Instruct | 1302 | 72B | Cornell | 73.39 | 55.65 | 81.04 | 72.98 | −17.74 | −8.06 |
| | | | Texas | 69.92 | 48.44 | 76.95 | 73.04 | −21.48 | −3.91 |
| | | | Washington | 63.91 | 54.89 | 68.80 | 62.78 | −9.02 | −6.02 |
| | | | Wisconsin | 77.88 | 59.50 | 83.49 | 80.69 | −18.38 | −2.80 |
| Llama-3.3-70B-Instruct | 1319 | 70B | Cornell | 75.40 | 51.61 | 78.63 | 77.02 | −23.79 | −1.61 |
| | | | Texas | 70.70 | 52.73 | 85.54 | 83.98 | −17.97 | −1.56 |
| | | | Washington | 69.55 | 62.03 | 77.07 | 74.06 | −7.52 | −3.01 |
| | | | Wisconsin | 74.45 | 60.44 | 81.93 | 78.19 | −14.01 | −3.74 |

than under flipping, average accuracies of all models still drop across all datasets. The strongest sensitivity is seen for Gemma-2-9B (with −11.66% relative drop) while the weakest sensitivity appears for Llama-3.3-70B-instruct (−3.09% relative drop). Thus, perturbing 2-hop topology while keeping 1-hop neighbors fixed still harms performance for every model, but the effect is more uniform and generally weaker than for first-order flipping. Crucially, in both cases, node texts and label distributions are held constant. Performance differences between original and rewired graphs must therefore be attributed to graph structural information, not to different collections of homophilic neighbor texts.

Exceptionally, GPT-3.5-Turbo-0125 and Llama-3-70B-Instruct show modest improvements (+6.67% and +0.40%) with flipping and extreme rewirings on Cornell dataset respectively. These anomalous improvements suggests that LLMs sometimes misapplies structural knowledge: they appear to use link patterns, but not always in the correct way.

Table 3: Impact of flipping rewiring with step-by-step instructions on node classification accuracy of GPT-3.5-Turbo-0125. Rewiring consistently decreased accuracy across all datasets.

| Dataset | Original | Rewired | $\Delta$(Acc.) |
|---|---|---|---|
| Cornell | 52.14 | 50.97 | -1.17 |
| Washington | 61.54 | 59.92 | -1.62 |
| Wisconsin | 50.96 | 49.04 | -1.92 |
| Texas | 54.55 | 50.20 | -4.35 |

### 2.2.3 Is Structure Misapplication Caused by Knowledge Shortage or Inconsistent Adherence?

LLMs' occasional misapplication of graph structural information may result from prior knowledge errors or inconsistent correct method adherence due to lack of explicit instructions. To evaluate this, we appended LLM-generated step-by-step instructions to the prompts and re-evaluated performance after flipping rewiring with GPT-3.5-Turbo-0125.

As shown in Table 3, with explicit step-by-step instructions, structural rewiring decreased performance by over $-1.17\%$ on all datasets, averaging a $-2.27\%$ decrease. This demonstrates that LLMs, even weak LLMs such as GPT-3.5-Turbo-0125, possess correct methods for using graph structural information (it knows that who links to whom and how many neighbors of each type matter), but without explicit guidance, it does not consistently adhere to this method in its chain-of-thought, sometimes applying it in the wrong direction.

### 2.2.4 Structural Sensitivity and Model Capability

We now relate structural sensitivity under rewiring to model capability, as measured by LMArena scores. For a model $M$, we define its sensitivity $S_R^M$ to a rewiring operation $r$ as:

$$S_r^M = O_r^M - R_r^M, \tag{2}$$

where $O_r^M, R_r^M$ are average accuracy over 4 datasets before and after rewiring $r$ of $M$. Larger sensitivity indicates larger performance loss when structure is perturbed.

We examine how $O_r^M, R_r^M$ and $S_r^M$ vary with the LMArena score $C^M$ of each model $M$ through linear regression, as shown in Table 4, Figure. For flipping rewiring, we observe that the model's sensitivity to rewiring is strongly and linearly correlated with its LMArena score (Pearson $r = 0.916, p = 0.0038$; Spearman $\rho = 0.893, p = 0.0068$), with a 100-point increase in the score corresponding to a $+19.43\%$ increase in sensitivity. The LMArena score also strongly explains the variance in sensitivity: a linear regression of sensitivity $S_F$ on the LMArena score yields $R^2 = 0.8391$, meaning that within our group's score range (1224–1319), this regression can explain over 80% of the variance in $S_F$. This finding suggests that the ability of LLMs to utilize graph structures is highly correlated with intelligence, and more intelligent models possess a stronger ability in this regard. $O_F$ and $R_F$ also show credible linear positive correlations with the LMArena score (Pearson $p < 0.01$, $R^2 > 0.7$), with slopes of 3.369e-3 and 1.426e-3 respectively, demonstrating that more powerful models have a significantly greater advantage on graph tasks with reliable structural information than when such information is unavailable. However, for extreme rewiring, although all LLMs remain sensitive, the sensitivity lacks a statistically significant correlation with model capabilities.

Flipping directly changes the direction of first-order edges, which significantly impacts how node roles are interpreted (e.g., "student page points to multiple course pages" vs. "course page is referenced by multiple student pages"). This type of rewiring is highly informative for heterophilic hyperlink graphs, and powerful LLMs can easily read it when presented in the form of concise neighborhood descriptions. As model capabilities improve, LLMs can gain a deeper understanding and utilization of the graph structure, and the accuracy of this process highly depends on these directional cues. Therefore, when the graph structure is flipping rewired, more powerful models suffer greater performance losses. In contrast, extreme rewiring preserves all first-order edges and only perturbs two-hop connections, which we expose to the model through coarse category counts (e.g., counts of two-hop student/teacher/course neighbors). This representation is

Table 4: Correlation and regression between model capability (LMArena score) and performance under rewiring on WebKB. Regression slopes are reported per 100 LMArena points.

| Target | Statistic | Flipping | Extreme |
|---|---|---|---|
| $S_r^M$ | Pearson $r$ | +0.9160 | −0.2521 |
| | Pearson $p$ | 0.0038 | 0.5855 |
| | Spearman $\rho$ | +0.8928 | −0.5714 |
| | Spearman $p$ | 0.0068 | 0.1802 |
| | $R^2$ | 0.8391 | 0.0635 |
| | Intercept | −2.3737 | 0.2685 |
| | Slope | +0.1943 | −0.0176 |
| | RMSE | 0.0251 | 0.0199 |
| Orig Acc | Pearson $r$ | +0.8999 | +0.6422 |
| | Pearson $p$ | 0.0058 | 0.1199 |
| | Spearman $\rho$ | +0.8929 | +0.6429 |
| | Spearman $p$ | 0.0068 | 0.1194 |
| | $R^2$ | 0.8099 | 0.4124 |
| | Intercept | −3.6786 | −1.0747 |
| | Slope | +0.3369 | +0.1440 |
| | RMSE | 0.0481 | 0.0507 |
| Rewired Acc | Pearson $r$ | +0.8414 | +0.5858 |
| | Pearson $p$ | 0.0176 | 0.1669 |
| | Spearman $\rho$ | +0.7857 | +0.6071 |
| | Spearman $p$ | 0.0362 | 0.1482 |
| | $R^2$ | 0.7080 | 0.3432 |
| | Intercept | −1.3049 | −1.3433 |
| | Slope | +0.1426 | +0.1616 |
| | RMSE | 0.0270 | 0.0659 |

too coarse, and its compression method results in information loss. Even high-capacity models may only be able to use these descriptions in a relatively simple way, unable to translate their additional intelligence into more complex two-hop reasoning. Therefore, the sensitivity of models with different capabilities to this type of rewiring is relatively uniform. Additionally, when utilizing two-hop neighborhood information, the average accuracy of all models across 4 datasets is still higher than when only using one-hop neighborhoods, even when the second-hop graph structure is extreme rewired. The additional performance gain may come from utilizing the overall category distribution patterns of all second-order neighbors, which does not require high intelligence.

In summary, the results of the statistical analysis support the following two conclusions:

- **Structural sensitivity is an intelligence-dependent behavior.** Stronger models benefit more from having the correct edge directions and lose more when those directions are corrupted. That is, their performance improvement is not just "better text understanding" but better structural utilization.

- **Most of the extra gain of strong models comes from exploiting structure.** Because $O_F$ rises significantly faster than $R_F$ with model capability, the additional headroom of frontier LLMs manifests primarily on the true graph, not on the structurally corrupted one. This indicates that, on low-homophily WebKB graphs, a substantial portion of their improvement over weaker models comes from using edges and edge directions as predictive signals, rather than from reading neighbor texts alone.

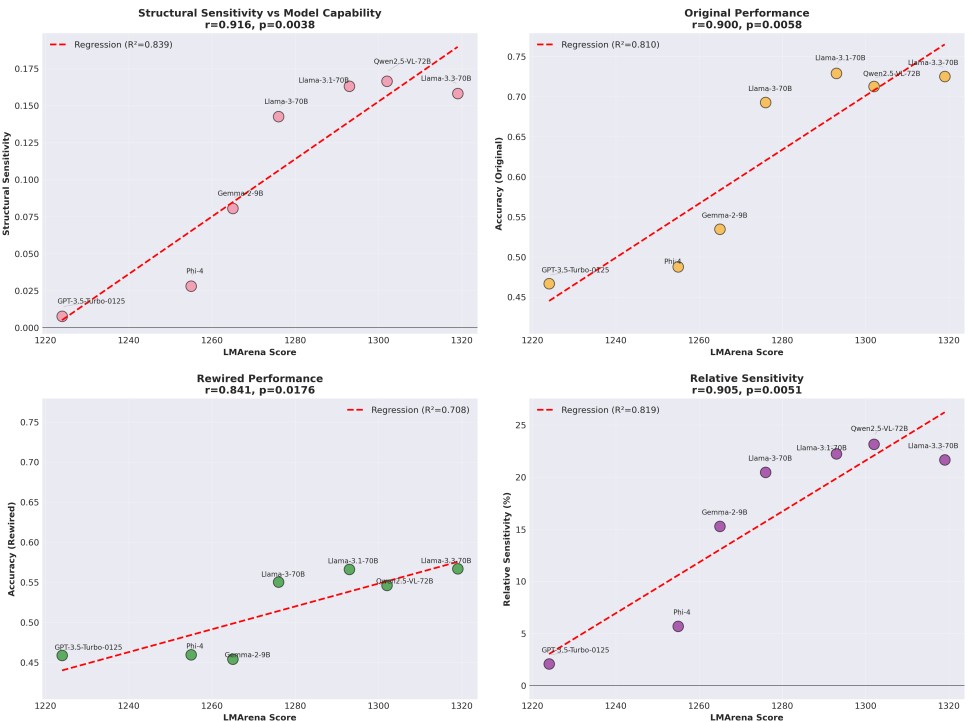

Figure 2: Correlation analysis between model capability and its sensitivity to flipping rewiring. (a) Stronger models show greater structural sensitivity (r=-0.92, p<0.01). (b-c) Unified y-axis scales reveal stronger models gain more from original structure than from rewired structure. (d) Relative sensitivity by percentage. All panels include regression lines (red dashed).

### 2.2.5 Case Study

As shown in Figure 4, flipping rewiring alters the LLM's prediction of a student page to a staff page from Cornell dataset by reversing the causal logic of the graph structure, even when textual attributes remain identical.

In the original graph, the target node receives an inbound link from a "Student Directory" page (".../students.html") and has outbound links to various "Staff" and "Project" pages. The model correctly interprets the inbound link from a directory as a membership signal ("The webpage has an incoming link from... a directory of students... This suggests that the webpage is related to a student"). In this reasoning process, LLMs correctly extracted the most informative clues for downstream tasks from the local topological structure, accurately understood and utilized the actual semantics implied by these clues, thereby achieving correct predictions, rather than relying on homophily.

After flipping rewiring, all edge directions are reversed. The target node now points outward to the "Student Directory" and receives inbound links from "Staff" and "Project" pages. The model re-interprets the target's role. It views the target as a hub or authority that is referenced by staff members and provides resources (links) to the student directory ("Outgoing link to a directory... suggests a connection to the academic community... possibly a staff member's profile"). The model still attempts to extract clues from the topological structure and understand its semantics, but misinterpreting the structural role due to the reversed edges thus changes its prediction to Staff (with 0.8 confidence).

This failure mode confirms that LLMs do not merely treat neighbors as a "bag of words" or rely on homophilic heuristics that neighboring nodes sharing the same label. If they did, the prediction would remain unchanged

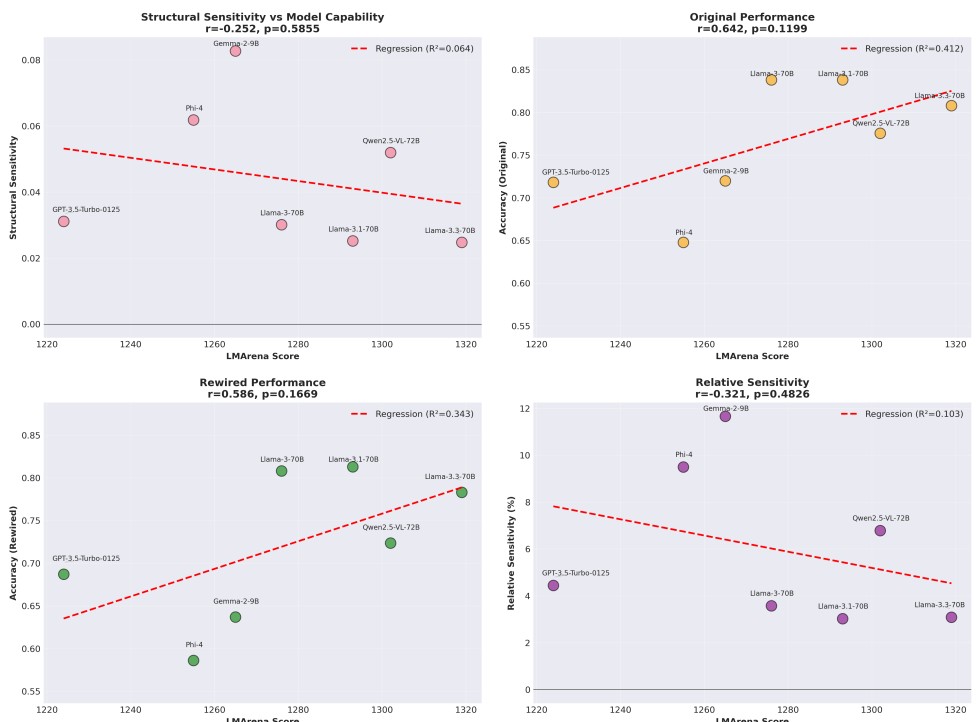

Figure 3: Correlation analysis between model capability and its sensitivity to extreme rewiring. (a) No correlation between capability and structural sensitivity (r=-0.25, n.s.), indicating uniform dependence. (b-c) Unified y-axis scales show nearly identical slopes for original and rewired performance, contrasting with Hop=1. (d) Relative sensitivity shows no capability-based pattern.

because the neighbor texts were identical in both scenarios. Instead, the specific direction of the edges (Inbound vs. Outbound) acted as a critical semantic signal for defining the node's role.

### 2.3 Inference Setting and a Label-Free Control

**What supervision is available at inference.** We clarify the supervision used in the rewiring experiments above. In the structure-aware template of Appendix B, each *first-order* neighbor is described together with its `category`, and—unlike standard semi-supervised node classification, in which only training/validation nodes carry labels—these categories are shown for *all* first-order neighbors. We make two points about this choice. First, this section is a deliberate *probe* of whether LLMs are sensitive to graph structure at all, not a deployable node-classification system: we intentionally provide a structure-rich context (neighbor roles, edge directions, degree statistics) and then perturb *only* the structure while holding node texts and the global label distribution fixed, so that any change in accuracy is attributable to structural interpretation rather than to text or label-distribution shifts. Second, on these low-homophily graphs ($H \leq 0.19$) a neighbor's category is *not* a homophily shortcut: knowing that a neighbor is, e.g., a `course` page does not let the model copy that label for the target, since neighbors predominantly carry *different* labels. The category is useful only as a *relational* attribute—the model must reason about which role links to which (e.g., a `student` page typically links out to `course`/`faculty` pages and is linked from `project` pages). This is exactly the structural reasoning that flipping/extreme rewiring perturbs, which is why accuracy degrades systematically (Table 2).

**Removing neighbor categories.** To verify that the observed sensitivity does not hinge on neighbor labels, we add a label-free control: we remove every first-order neighbor's `category` from the prompt and instead provide only its *full content abstract* (neighbor text) together with the inbound/outbound *link-*

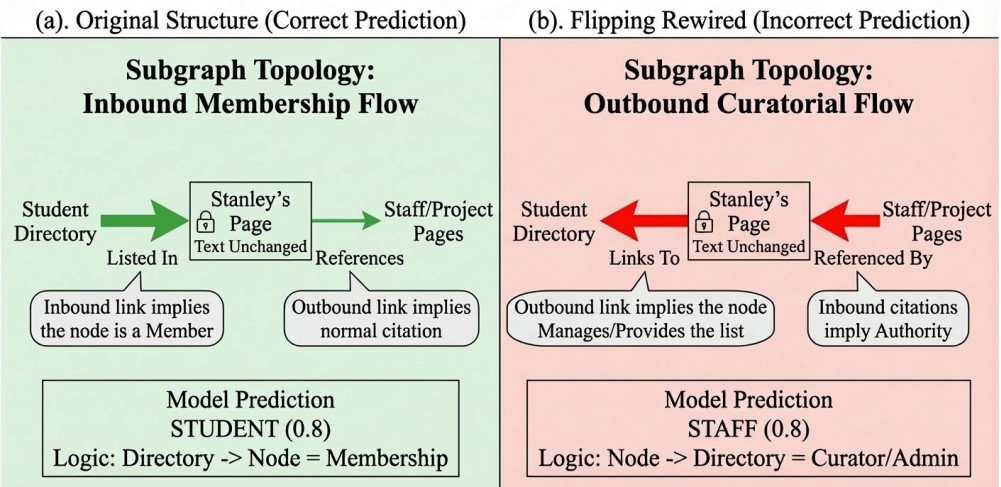

Figure 4: Case Study of flipping rewiring from Cornell dataset. This comparison illustrates how reversing edge directions alters the LLMs' interpretation of social roles, despite identical node text. (a) In the original graph, an inbound link from the "Student Directory" correctly signals membership (Prediction: Student). (b) After flipping rewiring, the edge is reversed; the node now links out to the directory, which the LLM re-interprets as a curatorial or administrative role (Prediction: Staff). This demonstrates that the model relies on structural directionality, not just textual homophily, for classification.

*pattern* statistics. No first-order neighbor label is revealed. We re-run the full panel of seven LLMs on the four WebKB graphs under original vs. flipping-rewired structure. As summarized in Table 5 and Figures 5– 7, structural sensitivity persists without neighbor categories: six of the seven models show non-positive change after rewiring (mean absolute drop 1.76 points), with the five 9B–72B instruction-tuned models dropping by $\sim 5\%$–$7\%$ relatively and Phi-4 essentially unchanged; GPT-3.5-Turbo—the weakest model, already identified as barely exploiting structure—remains the sole, expected outlier. Crucially, the capability– sensitivity relationship is preserved and remains statistically significant: the correlation between LMArena score and structural sensitivity is Pearson $r = 0.841$ ($p = 0.018$) and Spearman $\rho = 0.929$ ($p = 0.0025$), with a linear regression $R^2 = 0.71$ (Table 6). Because neighbor categories are no longer present, this rules out the possibility that the degradation in Table 2 was driven by neighbor labels: with *only neighbor text and link patterns*, more capable models still lose more accuracy when structure is perturbed, strengthening our conclusion that LLMs genuinely leverage graph structural information via ICL.

**Scope of the label-free control.** We apply this control to flipping (first-order) rewiring only, because that is the only rewiring scheme for which a label-free variant is well-posed. In the flipping experiment the perturbation is fully expressible in label-free terms—which pages link to which, and in which direction—so removing categories leaves the perturbation intact and the control measures exactly what it should. In the extreme-rewiring experiment, by contrast, second-order neighbourhoods enter the prompt exclusively as per-neighbour link-count statistics aggregated *by category*—the same coarse representation already identified above as the reason sensitivity is more uniform there. This leaves a label-free variant no well-posed design: removing the categories reduces the second-order description to bare in/out link totals, and since the rewiring procedure reassigns the pooled second-order neighbours in equal shares, the only label-free trace of the perturbation is a flattening of those totals toward their mean—an arithmetic artifact of the reassignment that carries no relational information, so a null result would be guaranteed by construction (no predictor, human or machine, could register the perturbation) rather than informative about structural utilisation; retaining the category statistics would instead reintroduce, one hop deeper, precisely the supervision the control is designed to remove. We therefore read the extreme-rewiring results as conditional on their count- based representation, and note that the paper's quantitative capability–sensitivity analysis rests on the flipping experiments, for which the label-free control above confirms that no conclusion depends on neighbor categories.

Table 5: Label-free control on WebKB: neighbor categories are removed from the prompt (only neighbor text + link patterns are provided). $\bar{O}, \bar{R}$ are accuracy (%) averaged over the four datasets before/after flipping rewiring; $\Delta = \bar{R} - \bar{O}$ (points); Rel. is the relative change (%). Models are ordered by LMArena score.

| Model | Arena | $\bar{O}$ | $\bar{R}$ | $\Delta$ | Rel. |
|---|---|---|---|---|---|
| GPT-3.5-Turbo-0125 | 1224 | 61.26 | 74.09 | +12.83 | +26.53 |
| Phi-4 | 1255 | 71.50 | 71.43 | −0.06 | +0.08 |
| Gemma-2-9B | 1265 | 77.03 | 72.47 | −4.57 | −5.96 |
| Llama-3-70B-Instruct | 1276 | 83.50 | 79.17 | −4.34 | −5.18 |
| Llama-3.1-70B-Instruct | 1293 | 87.26 | 82.06 | −5.21 | −5.98 |
| Qwen2.5-VL-72B-Instruct | 1302 | 83.29 | 77.67 | −5.62 | −6.69 |
| Llama-3.3-70B-Instruct | 1319 | 85.31 | 79.95 | −5.37 | −6.21 |
| **Mean** | | 78.45 | 76.69 | −1.76 | −0.49 |

Table 6: Label-free control: correlation and regression between model capability (LMArena score) and performance under flipping rewiring, with neighbor categories removed. Sensitivity $S^M = \bar{O}^M - \bar{R}^M$. Regression slopes are reported per 100 LMArena points.

| Target | Statistic | Value |
|---|---|---|
| Sensitivity $S^M$ | Pearson $r\,(p)$ | +0.8409 (0.0178) |
| | Spearman $\rho\,(p)$ | +0.9286 (0.0025) |
| | $R^2$ | 0.7071 |
| | Slope | +0.1773 |
| | RMSE | 0.0336 |
| Orig Acc $\bar{O}$ | Pearson $r\,(p)$ | +0.9150 (0.0039) |
| | Spearman $\rho\,(p)$ | +0.8214 (0.0234) |
| | $R^2$ | 0.8373 |
| | Slope | +0.2675 |
| Rewired Acc $\bar{R}$ | Pearson $r\,(p)$ | +0.7081 (0.0750) |
| | Spearman $\rho\,(p)$ | +0.7143 (0.0713) |
| | $R^2$ | 0.5014 |
| | Slope | +0.0902 |

**Persistence in current-generation models.** To verify that this structural sensitivity is not an artifact of the model generation we study, we repeat the label-free flipping-rewiring control on four 2025–26 models drawn from four distinct families and spanning a higher capability band—a 10–120B open-weight panel together with an open-weight flagship—reading their LMArena scores from the same snapshot used above (these models extend the capability axis up to 1475). Every new model degrades when structure is perturbed (15 of the 16 model–dataset pairs drop, with a single tie), and pooling them with the original seven models leaves 10 of 11 models degrading (sign test $p = 0.012$). Structural sensitivity therefore persists in newer and stronger models rather than vanishing; the full per-model tables, a capability–sensitivity analysis under two complementary metrics, a structural-channel decomposition of the prompt, and a current-model TAPTN-vs-GraphICL panel are reported in Appendix G.

## 3 How to Leverage Graph Structural Information More Effectively through ICL?

### 3.1 Text Attribute Passing Thoughts Network

In this section we propose a novel ICL framework that removes the key shortcomings which led previous studies to conclude that LLMs cannot utilize graph structures. We then demonstrate two things: the frame-

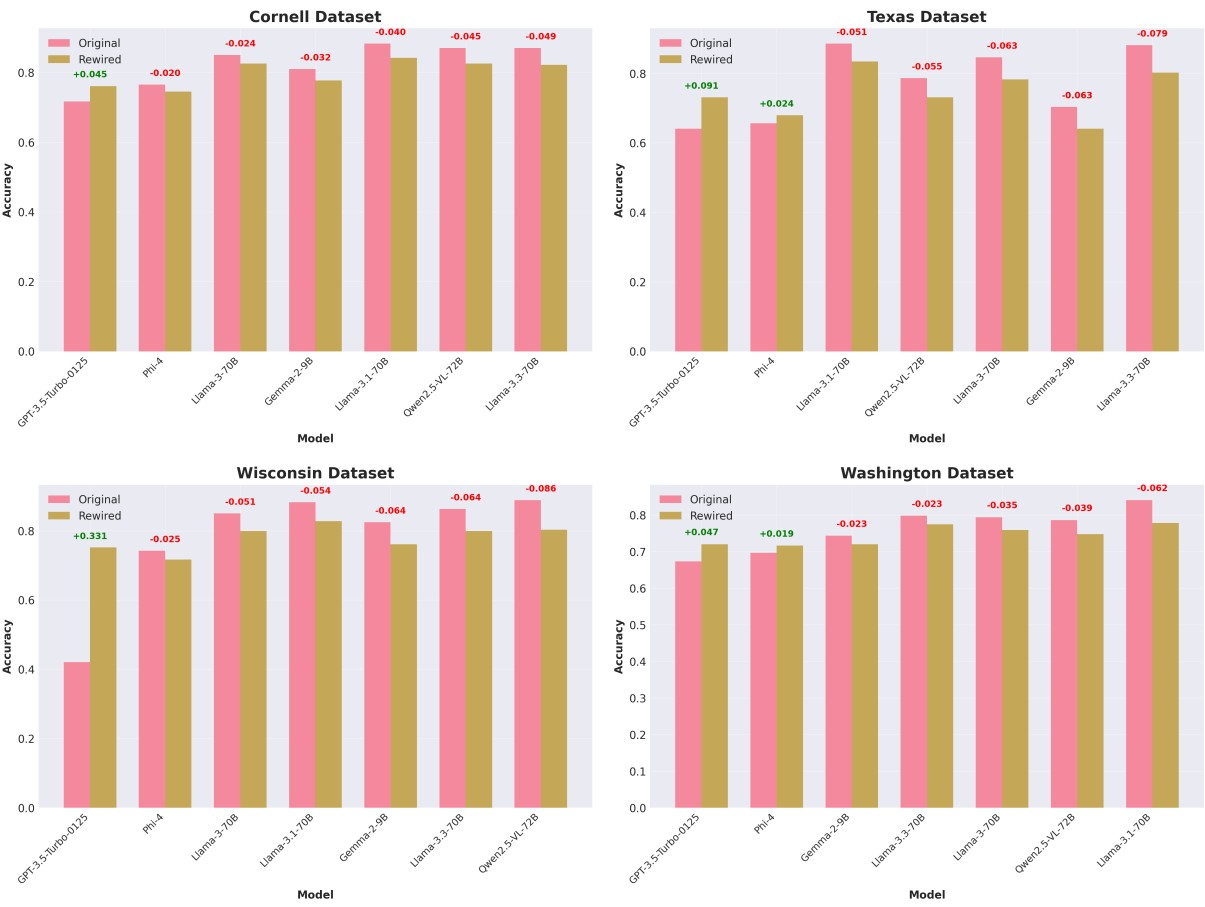

Figure 5: Label-free control (neighbor categories removed): per-dataset accuracy of each model before (Original) and after (Rewired) flipping rewiring. Even without neighbor labels, accuracy consistently drops after rewiring across datasets and models.

work gains additional performance from graph structure—beyond what traditional ICL methods obtain from homophily and neighbor attributes alone—and it remains effective on heterophilic graphs. An affirmative answer further supports the core argument of this paper: LLMs genuinely leverage graph structural information in text-attributed graphs via ICL, once we remove these confounding factors and endow them with an architecture explicitly designed for structural reasoning.

**Two limitations of ICL on node classification. Lack of an explicit structural reasoning mechanism**:

- **Template without structure:** Prior ICL-based TAG methods tend to treat neighbors either as extra textual evidence or as labeled exemplars, without an explicit representation of edge semantics or multi-hop topology.

- **Inconsistent reasoning:** Flipping-rewiring experiments showed that LLMs already possess a correct method for using structural information (e.g., interpreting hyperlink direction and local link patterns). However, without explicit guidance, LLMs—especially low-capability models—sometimes fail to apply this method consistently (e.g., before instructions are added, GPT-3.5-Turbo-0125 uses structural cues in the wrong direction on the Cornell dataset anomaly), leading to unstable performance.

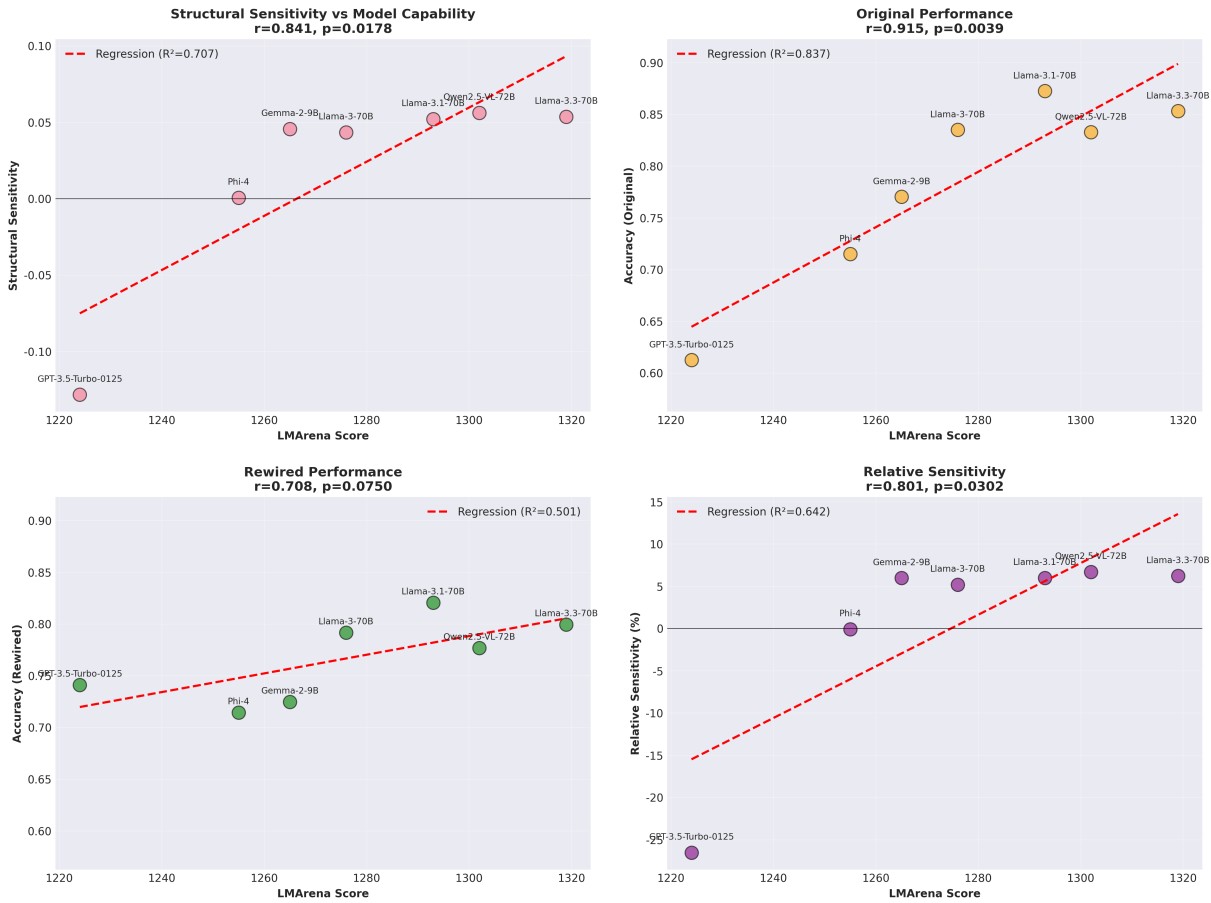

Figure 6: Label-free control: correlation between model capability (LMArena score) and structural sensitivity when neighbor categories are removed. Stronger models remain more sensitive to structural perturbation ($r = 0.85$, $p = 0.016$; $\rho = 0.93$, $p = 0.0025$), mirroring the with-label setting.

**Lack of a dedicated mechanism for lengthy high-order neighborhood contexts**: Existing ICL-based methods typically linearize large $K$-hop neighborhoods into long, noisy lists of raw node descriptions. In such prompts, subtle structural cues (who links to whom, directionality, etc.) are easily drowned out even when they are present in the template, because of the "lost in the middle" effect: information near the center of a long prompt is used much less effectively than information at its beginning or end. Conversely, as the extreme-rewiring experiments revealed, compressing high-order neighborhood information into a usable length at the price of significant information loss also caps the achievable benefit. This dilemma makes high-order structural signals hard to exploit unless the high-order neighborhood—neighbor attributes *and* graph structure—can be summarized with low information loss.

**Three design components.** TAPTN addresses these limitations with three components. *(i) Structure-aware template (SAT).* To integrate structural information into the prompt, we construct the textual description of a node's first-order neighborhood with a structure-aware template that contains the node's own attributes, its neighbors' attributes, and edge directions and semantics (with node/edge types as optional components); details are given in Appendix B. Unlike GraphICL-style templates, the SAT treats edges and topology as relational evidence: it explicitly describes which nodes link to which others and in what role, rather than listing neighbor texts as an unordered bag. *(ii) Iterative aggregation.* To avoid verbose, noisy contexts, we adopt an MPNN-like scheme that iteratively extracts effective information from $K$-order neigh-

**Structural Sensitivity Heatmap Across Datasets**

Figure 7: Label-free control: per-model, per-dataset structural-sensitivity heatmap (accuracy loss under rewiring) when neighbor categories are removed. Sensitivity is positive for nearly all (model, dataset) pairs.

bors and aggregates it into $(K-1)$-order neighbors, generating enhanced text attributes. Over $K$ iterations, the LLM thus utilizes $K$-order neighborhood information while processing only a first-order description per call; task-relevant multi-hop topological patterns (such as motifs and directed flows) can be detected, extracted, and emphasized along the way. *(iii) Self-generated instructions.* To avoid inconsistent reasoning, LLM-generated step-by-step instructions guide the model to adhere to the correct structure-usage strategy in every iteration.

We call the resulting method the Text Attributes Passing Thoughts Network (TAPTN), a new ICL method for node classification illustrated in Figure 8 and Figure 9. Like MPNNs, TAPTN consists of a message-passing phase and a readout phase. During message passing, $K$ iterations extract and summarize information from the first-order neighborhood to enhance the node's text attributes. This raises the abstraction level of the enhanced attributes, makes them more relevant to the downstream task, and condenses task-relevant $K$-order neighborhood information—including structural features such as anomalous trading patterns in financial graphs—into a single abstract. Unlike MPNNs, TAPTN passes text attributes instead of numerical tensors and relies on self-generated instructions grounded in the LLM's prior knowledge rather than on minimizing a loss over a training set, making it essentially a zero-shot learning method.

**Workflow overview.** Before formalizing each component, we summarize one TAPTN inference pass in plain terms (Figure 8; pseudocode in Algorithm 1). In each of the $K$ message-passing iterations, every node (i) collects its neighbors' current text attributes as messages, (ii) assembles these messages, together with the local topology, into a structured textual description of its first-order neighborhood using the structure-

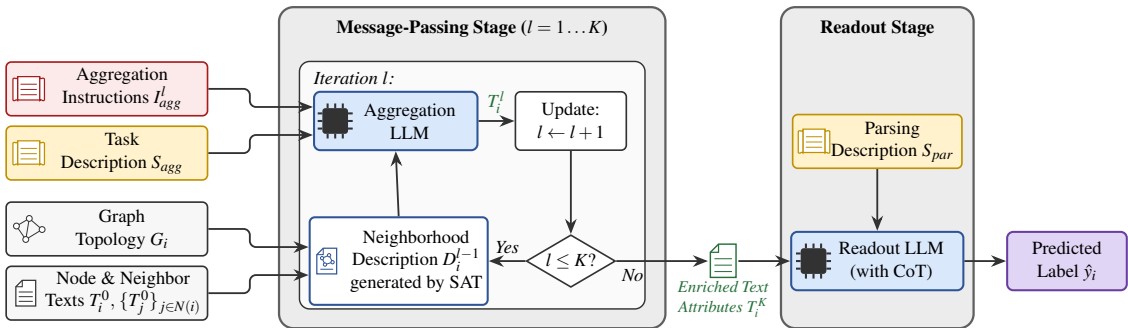

Figure 8: Overview of the TAPTN workflow. Given a target node $i$, each of the $K$ message-passing iterations assembles the node's and its neighbors' current text attributes and the local topology into a structure-aware neighborhood description $D_i^{l-1}$ (via the SAT), and an aggregation LLM call — guided by the self-generated instructions $I_{agg}^l$ and the task description $S_{agg}$ — rewrites the node's text attribute into $T_i^l$. After $K$ iterations, a readout LLM call parses the final attribute $T_i^K$ into the predicted label $\hat{y}_i$. Colors encode component roles and are shared with Figure 9: red for self-generated instructions, yellow for task descriptions, blue for the structure-aware template and LLM calls, green for enriched text attributes, purple for the final prediction, and neutral (uncolored) for raw graph data. The complete procedure is given as pseudocode in Algorithm 1.

Table 7: Notation used in the TAPTN framework.

| Symbol | Meaning |
|---|---|
| $G = (\mathbf{V}, \mathbf{E})$ | text-attributed graph with node set $\mathbf{V}$ and edge set $\mathbf{E}$ |
| $N(i)$; $G_i$ | first-order neighborhood of node $i$; topology of $N(i)$ |
| $K$; $l$ | total number of message-passing iterations; current iteration index |
| $T_i^0$; $T_i^l$ | original text attribute of node $i$; enriched attribute after $l$ iterations |
| $m_{ij}^l$ | message sent from neighbor $j$ to node $i$ in iteration $l$ (Eq. 5) |
| $D_i^l$ | SAT-generated textual description of $N(i)$ after $l$ iterations (Eq. 4) |
| $I_{agg}^l$ | self-generated step-by-step instructions for the $l$-th aggregation |
| $S_{agg}$; $S_{par}$ | task descriptions for aggregation and for label parsing |
| $F^{SAT}$; $F^{MSG}$; $F^{PAR}$ | template renderer; message function; label parser |
| $\hat{y}_i$ | predicted label of node $i$ (Eq. 7) |

aware template, and (iii) invokes the LLM — guided by the self-generated step-by-step instructions — to rewrite its own text attribute so that it absorbs the neighborhood evidence. After $K$ such iterations every text attribute summarizes the task-relevant content of the node's $K$-hop neighborhood, and a final readout call extracts the predicted label. The remainder of this subsection formalizes each step; Table 7 collects the notation, and readers primarily interested in the empirical findings may skim the equations and rely on Algorithm 1.

Specifically, the message-passing phase of the TAPTN framework is represented by:

$$T_i^l = LLM\left(T_i^{l-1}, D_i^{l-1}, I_{agg}^l, S_{agg}\right), \tag{3}$$

where $T_i^l$ is the enhanced text attribute of node $i$ after $l$ iterations, starting with $T_i^0$, the node's original text attribute. $I_{agg}^l$ is the LLM-generated instruction for the $l$-th aggregation that explicitly tells the model how to use neighbors and edges, and $S_{agg}$ is the task description for the downstream task. $D_i^l$ is the text description of node $i$'s first-order neighborhood after $l$ iterations, generated as:

$$D_i^l = F_{j \in N(i)}^{SAT}\left(G_i, m_{ij}^l\right), \tag{4}$$

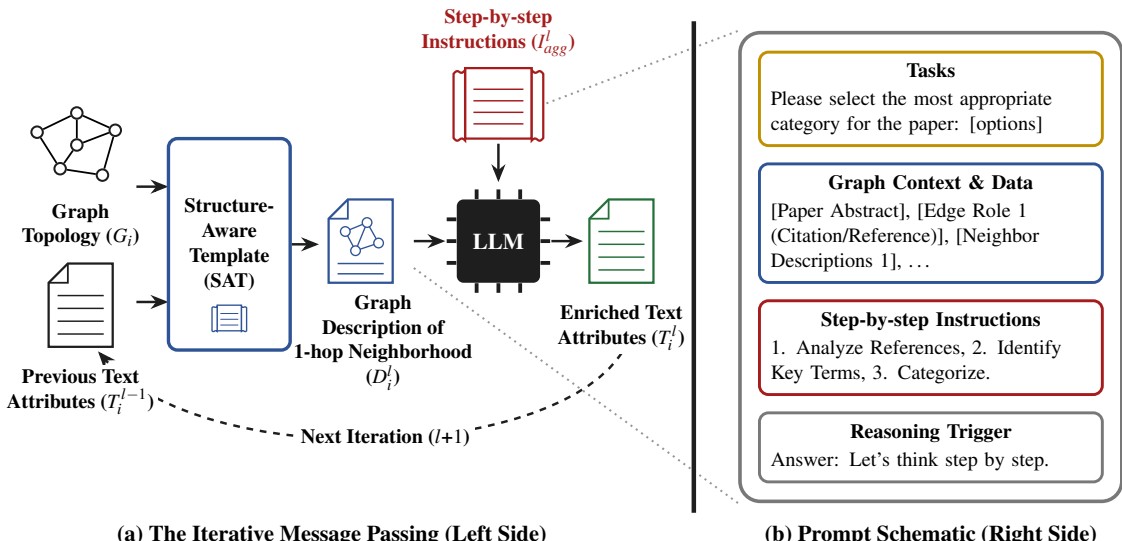

**(a) The Iterative Message Passing (Left Side)**     **(b) Prompt Schematic (Right Side)**

Figure 9: Subgraph (a) shows the iteration process of TAPTN. Subgraph (b) shows how the LLM-generated step-by-step instructions are attached to the prompt. Colors follow the semantic code of Figure 8.

where $N(i)$ is the neighborhood of node $i$, $G_i$ is the topology of $N(i)$, $F_{j \in N(i)}^{SAT}$ renders the textual description of $N(i)$ according to the template SAT, and $m_{ij}^l$ is the message sent from neighbor $j$ to target node $i$ during the $l$-th aggregation iteration. The SAT preserves edge semantics, which makes edge information usable and supports heterophilic graphs: in citation graphs it separates references and citations into distinct blocks; in e-commerce graphs it distinguishes co-purchased from also-viewed edges; and in transaction graphs it can encode sender/receiver roles. The message is generated as:

$$m_{ij}^l = F^{MSG}\left(T_j^l\right), \tag{5}$$

where $F^{MSG}$ is the message generation function. In our experiments, similar to most MPNN variants, $F^{MSG}$ is simply the identical function, but it can be adapted to different downstream tasks. For instance, if $T_j^l$ is too lengthy, a summarization approach can be used, i.e.,

$$F^{MSG}\left(T_j^l\right) = LLM(T_j^l, I_{sum}^l, S_{sum}), \tag{6}$$

where $I_{sum}^l$ is an instruction guiding LLMs to extract effective information, and $S_{sum}$ is a task description for summarizing text attributes.

For the node classification task, the readout phase of TAPTN is defined by:

$$\hat{y}_i = F^{PAR}(T_i^K), \tag{7}$$

where $F^{PAR}$ is a parser function extracting the predicted label $\hat{y}_i$ for node $i$ from the enhanced text attribute $T_i^K$ after $K$ iterations. While methods like regular expressions can implement $F^{PAR}$, owing to LLMs' limited instruction-following capability the labels they produce may sometimes fall outside the candidate range, or may not be sorted by relevance even when the reasoning process is correct. Therefore, we use the following $F^{PAR}$:

$$F^{PAR}(T_i^K) = LLM(T_i^K, S_{par}), \tag{8}$$

where $S_{par}$ is the task description used to extract the node label. Step-by-step instructions, task descriptions, graph description generation templates and prompts templates in our experiments are detailed in Appendix B. Algorithm 1 summarizes the complete procedure.

---

**Algorithm 1** TAPTN for zero-shot node classification

---

**Require:** Text-attributed graph $G = (\mathbf{V}, \mathbf{E})$ with initial text attributes $\{T_i^0\}_{i \in \mathbf{V}}$; number of message-passing iterations $K$; aggregation task description $S_{agg}$; parsing task description $S_{par}$

**Ensure:** Predicted labels $\{\hat{y}_i\}_{i \in \mathbf{V}}$

1: Generate the step-by-step instructions $I_{agg}^1, \ldots, I_{agg}^K$ with the LLM from the task description $S_{agg}$
2: **for** $l = 1$ **to** $K$ **do**
3:    **for all** nodes $i \in \mathbf{V}$ **do**
4:       $m_{ij}^{l-1} \leftarrow F^{MSG}(T_j^{l-1})$ for each neighbor $j \in N(i)$ {messages; identity by default, Eq. 5}
5:       $D_i^{l-1} \leftarrow F_{j \in N(i)}^{SAT}(G_i, m_{ij}^{l-1})$ {structure-aware neighborhood description, Eq. 4}
6:       $T_i^l \leftarrow LLM(T_i^{l-1}, D_i^{l-1}, I_{agg}^l, S_{agg})$ {instruction-guided aggregation, Eq. 3}
7:    **end for**
8: **end for**
9: **for all** nodes $i \in \mathbf{V}$ **do**
10:   $\hat{y}_i \leftarrow LLM(T_i^K, S_{par})$ {readout with CoT, Eq. 8}
11: **end for**
12: **return** $\{\hat{y}_i\}_{i \in \mathbf{V}}$

---

As a flexible framework, TAPTN can also be easily enhanced with an attention mechanism, implemented as follows:

$$W_{ij}^l = LLM(T_i^{l-1}, T_j^{l-1}, I_{att}^l, S_{att}), \tag{9}$$

$$D_i^l = F_{j \in N(i)}^{SAT}(G_i, m_{ij}^l, W_{ij}^l), \tag{10}$$

where $W_{ij}^l$ indicates the importance of neighbor $j$ to node $i$ in the $l$-th iteration, $I_{att}^l$ is the instruction guiding the LLM to calculate the importance of each neighbor, and $S_{att}$ is the task description for this calculation. The attention mechanism is optional: prior work has already demonstrated its effectiveness for ICL-based node classification, so we do not enable it in our experiments.

By incorporating these methods, TAPTN effectively addresses LLMs' challenges with overly verbose contexts and reasoning inconsistency through iterative extraction and aggregation guided by step-by-step instructions, improving performance in node classification tasks within a zero-shot ICL setting.

## 3.2 Experiments

To verify the effectiveness of TAPTN in addressing the shortcomings of existing ICL-based graph classification methods, and to demonstrate that graph structure can indeed be effectively understood and utilized by LLMs with these shortcomings mitigated by TAPTN framework, rather than just benefit from homophily and neighbor attributes, we tested two hypotheses:

**H1: Stable performance gain:** TAPTN consistently delivers performance gains exceeding the GraphICL-style baselines.

**H2: Stable gain from structure on heterophilic graph:** TAPTN consistently obtains performance improvements on heterophilic graphs after integrating neighborhood information, with larger neighborhood hops leading to greater gains.

If these two Hypotheses hold, TAPTN will necessarily be an effective framework with which LLMs can correctly understand and utilize graph structure information. Otherwise, TAPTN will fail to achieve stable performance superior to Naive Zero-Shot CoT which simply uses the target node's text attributes on heterophilic graphs lacking homophily.

We evaluate on five text-attributed graphs spanning two regimes from LLMNodeBed (Wu et al., 2025) benchmark:

Table 8: Node classification accuracy (%) across five text-attributed graphs. GraphICL denotes the GraphICL-style baseline that concatenates neighborhood text without a dedicated mechanism for leveraging graph structure or addressing verbose high-order neighborhood context.

| Method | Cora | Arxiv-2023 | Texas | Wisconsin | Cornell |
|---|---|---|---|---|---|
| 0-hop (zero-shot CoT) | 59.40 | 76.19 | 66.93 | 68.85 | 72.58 |
| GraphICL 1-hop | 62.55 | 83.49 | 65.23 | 76.01 | 76.21 |
| GraphICL 2-hop | 63.47 | 85.40 | 66.41 | 70.72 | 72.18 |
| TAPTN 1-hop | 72.69 | 87.30 | 89.84 | 85.67 | 85.08 |
| TAPTN 2-hop | 73.80 | 88.57 | 91.80 | 87.23 | 86.69 |

- **Homophilic citation graphs:** Cora (Yang et al., 2016) and Arxiv-2023 (Huang et al., 2024). Due to the large number of nodes in Arxiv-2023, we selected subgraphs consisting of seven categories of nodes for the experiments.

- **Heterophilic webpage graphs:** Texas, Wisconsin, Cornell from WebKB (Ghani et al., 2001) dataset. These datasets fall in (or close to) the mid-homophily pitfall range $(0.1, 0.3)$ (Luan et al., 2024b) and are classified as malignant heterophilic graphs considered particularly challenging for node classification (Luan et al., 2024a).

For each dataset, we compare:

- **0-Hop (Zero-shot CoT).** A graph-agnostic ICL baseline that uses only the target node's own text with chain-of-thought prompting, without any neighbors.

- **1-Hop / 2-hop GraphICL.** The original GraphICL-style prompt that linearizes one-hop or two-hop neighborhoods into a single context, without TAPTN-style iterative aggregation. For a clear comparison between the structure information leveraging, the GraphICL baseline in this paper is an enhanced version with the same explicit structure-aware templates as TAPTN. To ensure a fair comparison, GraphICL employs a random neighbor sampling strategy, allowing it to access the same set of neighbors as TAPTN.

- **1-Hop / 2-hop TAPTN.** Our full method with structure-aware template, $K = 1$ or $K = 2$ message-passing iterations and LLM-generated step-by-step instructions for each aggregation step and for the final classification.

We select GPT-3.5-Turbo-0125 as the backbone model for citation graphs and Llama-3.3-70B-Instruct for WebKB graphs. More details of our experiment settings are provided in Appendix C and E.

Table 8 shows that TAPTN's advantage persists and becomes substantially larger on malignant heterophilic graphs. Across all five datasets, 2-hop TAPTN is the top-performing zero-shot ICL method. Averaged over datasets, 2-hop TAPTN reaches 85.62% accuracy, compared to 68.79% for 0-hop CoT, 72.70% for 1-hop GraphICL, 71.64% for 2-hop GraphICL and 84.12% for 1-hop TAPTN. These results correspond to 2-hop TAPTN improves by +16.83% on average over 0-hop CoT, by +12.92% on average over 1-hop GraphICL, by +13.98% over 2-hop GraphICL and by +1.50% over 1-hop TAPTN. On Texas/Wisconsin/Cornell datasets, 2-hop TAPTN improves over 2-hop GraphICL by +25.39%, +16.5%, and +14.51% respectively, and by +24.87%, +18.38%, and +14.11% over 0-hop CoT. Notably, on these datasets, GraphICL exhibits little benefit from expanding to 2-hop neighborhoods (and can even degrade relative to 1-hop), and on Texas dataset 1-hop and 2-hop GraphICL even degrade from 0-hop zero-shot CoT, while TAPTN gains consistently from neighborhood information, and its gain consistently rises when considered neighborhood expands to 2-hop.

The WebKB dataset exhibits low homogeneity and is near the mid-homophily pitfall, meaning that simple heuristics relying on "neighboring nodes sharing the same label" or "neighboring nodes having different labels" are not feasible. In this case, the gains of TAPTN compared to GraphICL cannot be simply explained

by "reading similar neighbors"; they must reflect a better utilization of the graph structure (e.g., specific hyperlink patterns). This gain becomes even more significant with the introduction of multi-hop neighborhood information, which is sometimes detrimental to GraphICL, further demonstrating that TAPTN's performance advantage over GraphICL stems from its unique mechanism for utilizing structural information.

In summary, both hypotheses H1 and H2 are valid, and with TAPTN, our effective ICL framework mitigating shortcomings of traditional ICL for graph tasks, LLMs can indeed understand and utilize graph structure information. Previous research suggested that LLMs lacked this ability due to limitations of traditional ICL frameworks, rather than an inherent limitation of LLMs themselves.

### 3.3 Ablation Study

To demonstrate that TAPTN successfully mitigates the shortcomings of traditional ICL frameworks and achieves performance gains through its dedicated multi-hop neighborhood information processing mechanism and structural information reasoning mechanism, rather than just improves performance simply by providing LLMs with more opportunities for "self-reflection" through iterative neighborhood processing, we evaluate three hypotheses:

**H3: Enhanced utilization of K-hop neighborhoods through MPNN-like processing:** The reason TAPTN improves classification accuracy by leveraging multi-hop neighborhood information more effectively than GraphICL-style baselines, which directly concatenate neighborhood text, is iterative information extraction and aggregation (MPNN-like) specifically designed to address verbose multi-hop neighborhood contexts rather than mere self-reflection.

**H4: Improved adherence through instructions:** LLM-generated step-by-step instructions increase the consistency with which LLMs apply correct, structure-aware reasoning procedures, thereby contributing additional gains.

**H5: Component attribution and mutual reinforcement:** The structure-aware template, the step-by-step instructions, and iterative aggregation each contribute to accuracy and are mutually reinforcing and jointly necessary; the gains cannot be attributed to any single component—in particular, not to prompt construction alone in the absence of iterative aggregation.

Each hypothesis maps onto a dedicated experiment. To prove H3, we compared the accuracy of the following approaches on Cora and Arxiv-2023 datasets with GPT-3.5-Turbo-0125 (Table 9):

- TAPTN incorporating first-order and second-order neighborhood information;

- GraphICL incorporating first-order and second-order neighborhood information after 1-2 iterations of self-reflection. That is, the model first produces a preliminary label and explanation, then is asked to reflect on and revise its answer.

H4 and H5 are tested by a pair of complementary $2 \times 2$ ablations. Table 10 crosses the structure-aware template with the instructions while iterative aggregation is switched *off* throughout (a GraphICL ablation), and Table 11 crosses the instructions with iterative aggregation while the structure-aware template is held *on*; the latter contains precisely the GraphICL-with-instructions-but-without-aggregation baseline that isolates iterative message passing from prompt construction. The instruction toggles of both tables jointly evidence H4 (instruction gains with and without aggregation), and the two designs together attribute the overall gain among the three components and expose their interactions (H5). All ablation runs use the per-dataset backbones of Section 3.2—GPT-3.5-Turbo-0125 for the citation graphs and Llama-3.3-70B-Instruct for the WebKB graphs—each cell being a single run at temperature 0.

**Iterative aggregation versus self-reflection.** As shown in Table 9, For Cora and Arxiv-2023, adding self-reflection to naive GraphICL sometimes helps slightly but also frequently harms. For example, on Cora, 1-hop GraphICL improves from 62.55% (no self-reflection) to 69.19% (with 1 iteration of self-reflection), but

Table 9: Comparison between TAPTN and GraphICL with self-reflection reported in percentage. "RI" denotes the number of self-revision iterations. TAPTN consistently outperforms naive GraphICL with self-reflection, with an average improvement of +6.59%, confirming the superiority of its framework in utilization of graph structural information.

| Method | Order | RI | Cora | Arxiv-2023 |
|---|---|---|---|---|
| GraphICL | 1 | 0 | 62.55 | 83.49 |
| | | 1 | 69.19 | 74.92 |
| | | 2 | 68.45 | 73.65 |
| | 2 | 0 | 63.47 | 85.40 |
| | | 1 | 69.18 | 75.87 |
| | | 2 | 68.45 | 80.00 |
| TAPTN | 1 | / | 72.69 | 87.30 |
| | 2 | / | **73.80** | **88.57** |

2-hop GraphICL with self-reflection can underperform its non-reflective counterpart on Arxiv-2023, and the second iteration of self-reflection harms 1-hop GraphICL on both datasets. There is no consistent trend that "more reflection" yields monotonic gains. This comparison indicates that self-reflection alone is insufficient and inconsistent. TAPTN still dominates even after reflection. Even when naive GraphICL is given its best-performing RI configuration, TAPTN's 2-hop variant remains substantially better. Aggregating across datasets and configurations, TAPTN improves node classification accuracy by about +6.6% over the strongest self-reflective baseline. This confirms that TAPTN's advantages on multi-hop neighborhood information utilization stem from MPNN-like structured aggregation, not from merely allowing the LLM to re-think its answers more times.

**Decoupling structure-aware templates and instructions.** TAPTN is an MPNN-like architecture built from three co-designed components: a structure-aware template (SAT) that exposes edge semantics and topology (Eq. 4), self-generated step-by-step instructions that force the model to apply the correct structural-reasoning procedure at each step (Eq. 3), and iterative aggregation that summarises multi-hop neighbourhoods with low information loss. None of these is a detachable prompt trick: the instructions enter the message-passing recursion itself and directly target the inconsistent-adherence failure diagnosed in Section 2, exactly as the SAT instantiates the message/description function, much as message, aggregation, and readout functions are constitutive parts of a GNN. Our main results (Table 8) show that the integrated architecture is effective; to understand how that gain is distributed, we attribute it *among* these components, rather than separating "architecture" from "prompting."

We therefore isolate each component while holding the others fixed, first varying the SAT and the instructions independently on the single-pass *GraphICL baseline*—i.e. with iterative aggregation switched off, so this is an ablation of GraphICL rather than of TAPTN—under 1-hop and 2-hop prompting (Table 10). At 1-hop the structure-aware, instruction-guided GraphICL setting coincides with 1-hop TAPTN, since a single hop involves no multi-hop aggregation; the values therefore agree with Table 8.

Holding aggregation off, adding instructions improves accuracy in all twenty instruction toggles of Table 10, with the largest gains on heterophilic graphs (e.g. Texas 1-hop: +33.21 without the SAT and +24.61 with it). The SAT likewise helps on its own—most clearly on heterophilic graphs (Texas 1-hop without instructions, $45.31 \rightarrow 65.23$; Cornell 1-hop, $56.45 \rightarrow 76.21$)—while on the strongly homophilic Cora it can hurt when used without instructions (1-hop, $69.37 \rightarrow 62.55$), because exposing edge structure pays off only once the instructions make the model use it correctly. The two components are thus mutually reinforcing rather than independent: the SAT supplies the structural evidence while the instructions enforce its correct use, and in nine of the ten rows the best accuracy is attained with both components on (the sole exception, Arxiv-2023 2-hop, trails the instructions-only cell by a mere 0.32 points, on a strongly homophilic citation graph where structural evidence has the least to add). This interaction is itself evidence for our thesis—LLMs possess

Table 10: Decoupling the structure-aware template and the step-by-step instructions *on the GraphICL baseline*—a single-pass GraphICL-style prompt *without* iterative aggregation, not TAPTN—under 1-hop and 2-hop prompting. We independently toggle the structure-aware (non-anonymous) edge-labelled template ("Struct.") and the LLM-generated instructions ("Instr."), so the table isolates the marginal effect of each and their interaction while aggregation is switched off throughout. Backbones follow Section 3.2: GPT-3.5-Turbo-0125 for the citation graphs, Llama-3.3-70B-Instruct for the WebKB graphs. Accuracy in %.

| Dataset | Hop | w/o Struct. Template | | w/ Struct. Template | |
|---|---|---|---|---|---|
| | | w/o Instr. | w/ Instr. | w/o Instr. | w/ Instr. |
| Cora | 1-hop | 69.37 | 71.40 | 62.55 | 72.69 |
| | 2-hop | 62.55 | 72.14 | 63.47 | 73.25 |
| Arxiv-2023 | 1-hop | 81.59 | 85.71 | 83.49 | 87.30 |
| | 2-hop | 82.54 | 86.35 | 85.40 | 86.03 |
| Texas | 1-hop | 45.31 | 78.52 | 65.23 | 89.84 |
| | 2-hop | 57.03 | 73.44 | 66.41 | 86.72 |
| Wisconsin | 1-hop | 55.45 | 81.31 | 76.01 | 85.67 |
| | 2-hop | 60.75 | 80.37 | 70.72 | 83.18 |
| Cornell | 1-hop | 56.45 | 76.61 | 76.21 | 85.08 |
| | 2-hop | 62.10 | 76.61 | 72.18 | 81.45 |

Table 11: Factorial ablation at 2 hops (all rows use the structure-aware template), crossing LLM-generated step-by-step instructions with iterative aggregation. The two middle rows each add a single component to the GraphICL baseline; the last row is the full TAPTN. Backbones as in Table 10. Accuracy in %.

| Method (2-hop, structure-aware) | Cora | Arxiv-2023 | Texas | Wisconsin | Cornell | Avg. |
|---|---|---|---|---|---|---|
| GraphICL (no instr., no aggr.) | 63.47 | 85.40 | 66.41 | 70.72 | 72.18 | 71.64 |
| + instructions (no aggr.) | 73.25 | 86.03 | 86.72 | 83.18 | 81.45 | 82.13 |
| + iterative aggregation (no instr.) | 70.85 | 87.30 | 66.79 | 73.21 | 77.82 | 75.19 |
| TAPTN (instr. + aggr.) | **73.80** | **88.57** | **91.80** | **87.23** | **86.69** | **85.62** |

the right structural method but apply it reliably only when the architecture both exposes the structure and compels adherence to it.

**Disentangling instructions and iterative aggregation.** Restricting attention to 2-hop structure-aware prompting, Table 11 crosses the two remaining factors—instructions and iterative aggregation—in a $2 \times 2$ design, so that each component's marginal effect and their interaction can be read directly. The two middle rows add a single component to the GraphICL baseline, and the last row is the full TAPTN.

Each component carries an independent contribution, and the components reinforce one another. Averaged over the five datasets, adding instructions improves accuracy by +10.49 without aggregation and by +10.42 with it, while adding iterative aggregation improves accuracy by +3.56 without instructions and by +3.49 with them; every single-component configuration remains strictly below the full TAPTN on every dataset. The interaction is strongest exactly where our thesis predicts—on malignant heterophilic graphs: on Texas,

aggregation alone adds only +0.38, but +5.08 once instructions are present, because multi-hop message passing helps only when the model is compelled to read edge structure correctly at each step. No single component reproduces the full method and none is dispensable; the structure-aware template, the instructions, and iterative aggregation are jointly necessary and mutually reinforcing parts of one structure-reasoning architecture. The instructions, in particular, are not generic prompt engineering but the adherence-enforcing element of the message-passing recursion (Eq. 3) that turns the latent structural method—identified in Section 2—into consistent behaviour; their large measured effect is therefore confirmation of the paper's central claim rather than a confound to be discounted. The same pattern holds at scale: on the 13,482-node Amazon co-purchase graph, under the budget backbone of Table 13, second-order aggregation without instructions brings no benefit over the one-hop and GraphICL baselines, and only the combination of iterative aggregation and instructions improves accuracy—a large-scale corroboration that the components are jointly necessary rather than independently sufficient.

Among the citation graphs, TAPTN's improvements are larger on Cora (+10.33 over the 2-hop GraphICL baseline, versus +3.17 on Arxiv-2023), where baseline accuracy was lower, indicating its particular effectiveness in tasks with ambiguous classification standards and overlapping node categories. Although there are concerns about reliance on instructions and potential error propagation, TAPTN's iterative process demonstrated resilience against cumulative errors. Since instructions are self-generated by LLMs, TAPTN remains scalable and practical without requiring manual intervention. A small annotated subgraph could further refine instructions if needed, enhancing accuracy and reducing risks. The specifically designed guidance mechanism is a core innovation overcoming LLMs' inconsistent adherence to correct graph structural methods.

In conclusion, the experiments confirm all three hypotheses. TAPTN's iterative extraction and aggregation effectively utilize K-order neighborhood information, achieving higher accuracy than naive methods, even after self-reflection. Performance gains result from better utilization of high-order neighborhood structural information. Additionally, LLM-generated step-by-step instructions significantly enhance accuracy, particularly with first-order information, ensuring consistent adherence to correct graph structural usage. Disentangling the components further shows that the structure-aware template, the step-by-step instructions, and iterative aggregation are mutually reinforcing and jointly necessary: each is individually beneficial, their interaction is strongest on heterophilic graphs, and no single component reproduces the full method. The gains therefore stem from TAPTN as an integrated structure-reasoning architecture—in which the instructions are an intrinsic part of the message-passing process that compels correct structural reasoning, not a detachable prompt—rather than from any isolated component. TAPTN overcomes LLMs' limitations with verbose contexts and inconsistent method application through ICL, broadening LLMs' applicability in node classification tasks by leveraging zero-shot learning and interpretability.

### 3.4 Scalability and Portability

Cora and Arxiv-2023 are citation graphs with fewer than 3,000 nodes, so concerns about TAPTN's scalability to large graphs and its portability across domains may arise. To address both, we evaluate TAPTN on Amazon, an e-commerce co-purchase network with 13,482 nodes—five times larger than our largest citation graph, an order of magnitude larger than the WebKB graphs, and from an entirely different domain. This graph is a deliberately *sparsified* subgraph of the 2.45M-node ogbn-products network, obtained by uniformly subsampling 50k nodes, taking the induced subgraph, and keeping its largest connected component (construction details in Appendix C); the induced-subgraph step prunes most edges and is our cost-control device, bounding the neighbourhood text and the number of LLM calls per node. Because exhaustively labelling such a graph with a large LLM is costly, and because the benefit of TAPTN is realised *per target node* (inference cost grows linearly in the number of queried nodes; Appendix A), we evaluate on a fixed 400-node test set sampled from the graph rather than on the full node set; all methods are scored on the *same* 400 nodes (verified by intersecting the result sets), so the reported gaps are not an artefact of differing samples. This pay-per-node property is itself a scalability advantage over transductive GNNs, which must be trained over the whole graph.

Our primary scalability evidence uses a uniform Llama-3.3-70B-Instruct backbone (Table 12). TAPTN's structural mechanisms carry over cleanly to this large, non-citation graph: 2-hop TAPTN reaches 86.75%

Table 12: Primary scalability/portability result: TAPTN on the Amazon (ogbn-products) co-purchase subgraph (13,482 nodes) under a uniform Llama-3.3-70B-Instruct backbone, on a fixed 400-node test set (identical nodes across all configurations, verified by intersection). "GraphICL 2-hop" is the structure-aware baseline without instructions or iterative aggregation; "dense full-graph nbhd." is the same baseline but fed each node's *full-graph* neighbourhood (test-node mean degree $\approx$ 106 vs. $\approx$ 2.8 in the sparsified subgraph), saturating the per-call neighbour budget. A constrained budget-backbone configuration is reported in Table 13.

| Method | Accuracy (%) |
|---|---|
| GraphICL 2-hop (no instr., no aggr.) | 76.50 |
| GraphICL 2-hop (dense full-graph nbhd.) | 76.75 |
| TAPTN 1-hop | 83.75 |
| TAPTN 2-hop | **86.75** |

and 1-hop TAPTN 83.75%, against 76.50% for the structure-aware GraphICL baseline (no instructions, no iterative aggregation)—gains of +10.25% and +7.25%, respectively. Notably, this margin is even larger than on the small citation graphs: while we refrain from inferring a monotone size effect from a single large graph (the domain, label space, and homophily also differ), the result establishes the claim that matters for scalability—the advantage of correctly guided structural reasoning does *not* diminish when moving to a much larger, out-of-domain graph; expanding the receptive field from one to two hops continues to help, exactly as our thesis predicts.

Nor is the gap an artefact of the sparsified subgraph starving GraphICL of context. To rule this out, we re-ran the GraphICL baseline on the same 400 nodes but supplied each node's *dense* neighbourhood from the full 2.45M-node graph: the test nodes' mean degree rises from $\approx$ 2.8 to $\approx$ 106 (median 74)—a $\approx 38\times$ density increase under which the per-call neighbour cap (40 first-hop and 10 second-hop descriptions in our configuration) binds for most nodes, saturating every prompt with neighbour text. Its accuracy is essentially unchanged (76.50% $\rightarrow$ 76.75%). This saturated configuration is GraphICL's empirical *context ceiling* on this graph—no further neighbourhood text can be bought—yet TAPTN operating on the heavily sparsified subgraph still exceeds it by 10 points (and even 1-hop TAPTN exceeds it by 7). Sparsification thus does double duty: it keeps TAPTN's per-node cost at the level of our small benchmarks, while TAPTN extracts more from this impoverished structure than GraphICL extracts from the full graph; the gain comes from guided iterative aggregation, not from the sheer volume of neighbourhood text, which inflates the per-call token count roughly three-fold for no accuracy return (Appendix A).

We further ask whether these gains survive a *constrained compute budget*, since a practical appeal of an ICL approach on a large graph is the option to run the bulk of its calls on a cheap small model. Table 13 reports such a budget configuration, in which the first (neighbour-classification) iteration is delegated to Llama-3.1-8B and only the final refinement uses Llama-3.3-70B. Even under this constraint, 2-hop TAPTN still outperforms GraphICL, 1-hop TAPTN, and 2-hop TAPTN without instructions (by +1.75%, +2.00%, +2.75%), confirming that the gains are not an artefact of using a powerful model at every step but follow from the architecture itself. The guidance mechanism is effective only at the second iteration here, because the weak 8B first pass cannot produce neighbour reasoning good enough to serve as exemplars; consistently with the cost analysis (Appendix A), this budget backbone trades accuracy for cost, so the uniform backbone above is preferable when affordable. Crucially, *without* TAPTN's two mechanisms, incorporating second-order neighbourhoods brings no benefit at all on this graph—the same conclusion we draw on the small graphs, now confirmed at scale.

Appendix A provides both the asymptotic $O(|\mathbf{V}| \cdot L)$ time-complexity analysis and an empirical study of inference cost, token consumption, and wall-clock latency on this graph (Table 21), quantifying the practical overhead of iterative prompting and instruction generation and the economic rationale for preferring TAPTN over GraphICL at scale.

Table 13: Budget-backbone robustness check on the Amazon graph: 2-hop TAPTN using the economical Llama-3.1-8B for the first iteration and Llama-3.3-70B only for refinement. Even under this constrained budget, 2-hop TAPTN with instructions outperforms GraphICL, 1-hop TAPTN, and 2-hop TAPTN without instructions, confirming the gains do not require a powerful model at every step. The uniform-backbone results are in Table 12.

| Method | Hop | w/o instr- uction(%) | w/ instru- ction(%) |
|--------|-----|--------|--------|
| GraphICL | 1 | 74.75 | / |
|  | 2 | 75.00 | / |
| TAPTN | 1 | 74.75 | 74.75 |
|  | 2 | 74.00 | **76.75** |

# 4    Are LLMs' Ability of Leveraging Graph Structure Information on Par with GNNs?

Sections 2 and 3 have shown that general-purpose LLMs are structurally sensitive: when we perturb edges while keeping node texts fixed on low-homophily graphs, accuracy drops systematically, especially for stronger models. Moreover, TAPTN converts this latent sensitivity into reliable gains by performing edge-aware, iterative aggregation in text space. We now ask a more stringent question:

**When provided with a reasonably designed ICL architecture such as TAPTN, is the structural utilization capability of LLMs still inferior to state-of-the-art GNNs on text-attributed graphs?**

A naive comparison—zero-shot TAPTN accuracy against fully supervised GNNs—cannot answer this question: zero-shot TAPTN may trail simply because of systematic mismatches between the LLM's pre-training data and a dataset's labelling standards, a textual and supervision confound that says nothing about structural utilisation. Answering the question therefore requires a comparison where (i) both sides have access to the same node text encoder, (ii) both operate over the same edge structure, and (iii) any remaining performance gap can be attributed to differences in how they use graph structure, not to differences in raw textual capacity or training data.

To support this, we fine-tuned a small-scale language model (LM) using TAPTN-generated enhanced text attributes to mitigate potential biases. In the control group, we fine-tuned the LM using only the nodes' original text attributes (TA) and then used the LM-generated embeddings to train various GNNs. Detailed pipelines for both methods are shown in Figure 10.

## 4.1   Experiment Setup: Aligning TAPTN and GNN

We consider homophilic graphs with standardized GNN baselines: the **Cora** and **Arxiv-2023** citation networks and the **ogbn-products** (Hu et al., 2020) co-purchase graph. We deliberately include ogbn-products—a larger and more diverse OGB benchmark from a non-citation (e-commerce) domain—in this same TAPTN+LM versus GNN comparison, so that the structural-competitiveness claim is not confined to small citation graphs but is tested on a substantially larger and topologically different graph. For products we use the same 13,482-node connected subgraph as in Section 3 (construction in Appendix C); the fixed 400-node pool of the ICL experiments is re-split 60/20/20 (240/80/80 train/validation/test) under each random seed, so that the ICL and supervised evaluations rest on an identical node population. For all datasets, we align the LLM and GNN pipelines around a shared text encoder:

- **TAPTN+LM.** As shown in Figure 10(b), we first run 2-hop TAPTN with the same LLM backbones as in Section 3 to generate an enhanced text attribute $T_i^{(2)}$ for each node, using the edge-aware templates and instructions from Section 3. We then fine-tune DeBERTa-base as a standard text classifier on these enhanced texts, using the node labels as supervision. Importantly, during fine-

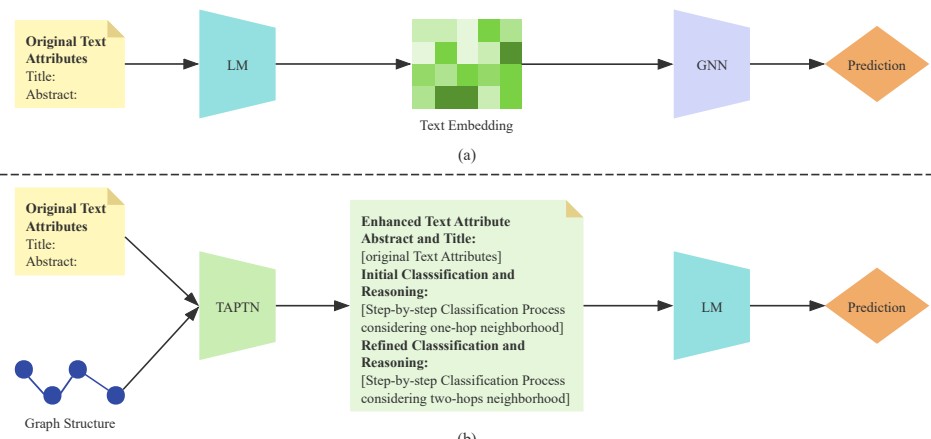

Figure 10: Subgraph (a) shows the pipeline of training GNNs on text embeddings of original text attributes. Subgraph (b) shows the pipeline of fine-tuning LMs on Enhanced Text Attributes generated by TAPTN.

tuning the LM sees no neighborhood structure—all structural information must be encoded into $T_i^{(2)}$ by TAPTN's ICL stage.

- **TA+GNN+LM (control).** As shown in Figure 10(a), in the control pipeline, we fine-tune the same DeBERTa-base LM using only the original text attributes (TA) of nodes, again without any graph structure. We then use this LM to provide node embeddings for GNNs trained on the original graph, so that structural information is exploited only by the GNN (via the adjacency matrix), not by the LM. We consider this control in two training regimes: a *frozen-feature* regime, in which the GNN is trained on embeddings from the independently fine-tuned encoder, and a *jointly-trained* regime, in which the text encoder and the GNN are optimised end-to-end on the same graph. The jointly-trained regime removes any bias arising from decoupling the encoder from the structural learner and grants the GNNs the most favourable, fully consistent training conditions.

- **TA+LM and GraphICL+LM.** We also include TA+LM (LM classifier on original texts only, no graph) and GraphICL+LM (GraphICL-style neighborhood descriptions fed directly to the LM, then fine-tuned) as baselines, to isolate the contribution of TAPTN's structured utilization from both "no structure" and "unstructured structure" controls.

To ensure that TAPTN+LM and TA+GNN+LM see neighborhoods of comparable depth, we use 2-hop TAPTN and set the number of graph convolution layers in each GNN to 2. We conducted experiments on the Cora (Yang et al., 2016), Arxiv-2023 (Huang et al., 2024), and ogbn-products (Hu et al., 2020) datasets using DeBERTa-base as the LM. We evaluated 5 widely used GNN architectures (GAT (Velickovic et al., 2017), GraphSAGE (Hamilton et al., 2017), GATv2 (Brody et al., 2022), ChebNet (Defferrard et al., 2016), and DGI (Velickovic et al., 2019)) and 9 SOTA models (GCNII (Chen et al., 2020), ASDGN (Gravina et al., 2023), DirGNN (Rossi et al., 2024), ACM-GNN (Luan et al., 2022), DMP (Yang et al., 2021), Graphsaint (Zeng et al., 2019), FSGNN (Maurya et al., 2022), APPNP (Gasteiger et al., 2018), and the recent linear graph Transformer GraphTARIF (Hu et al., 2026)), matching the groupings of Table 14.

During LM fine-tuning, no structural information from the neighborhood was provided. Thus, TAPTN+LM can match the classification performance of TA+GNN+LM only if TAPTN's ability to leverage neighborhood structural information is comparable to that of GNNs.

**Evaluation protocol.** Unless otherwise stated, every reported accuracy is averaged over *five random seeds*, and we summarise each pipeline by its mean and standard deviation; all pairwise comparisons against TAPTN+LM use two-sided paired *t*-tests at the seed level. With five seeds, the Student-*t* 95% confidence

Table 14: Node classification accuracy (%) of TAPTN+LM versus GNNs trained on frozen TA embeddings, on the homophilic Cora, Arxiv-2023, and ogbn-products graphs. All methods include "+LM" by default; results are mean±std over five random seeds. Per-dataset paired-$t$-test significance against TAPTN+LM is summarised in Table 16, and the jointly-trained GNN counterparts appear in Table 17.

| Method | Cora | Arxiv-2023 | Products |
|---|---|---|---|
| *DeBERTa on raw text attributes (TA+LM)* | | | |
| TA | 76.79±1.54 | 90.52±1.57 | 63.25±10.63 |
| *Normal GNNs on frozen TA embeddings* | | | |
| GAT | 84.91±0.93 | 91.04±1.17 | 68.25±5.97 |
| SAGE | 84.24±1.11 | 91.50±1.09 | 69.25±6.94 |
| ChebNet | 84.13±0.95 | 90.12±1.11 | 68.00±6.16 |
| DGI | 84.84±0.88 | 92.89±0.93 | 72.75±4.54 |
| GATv2 | 83.17±2.06 | 91.96±1.03 | 69.25±4.89 |
| *SOTA GNNs on frozen TA embeddings* | | | |
| GCNII | 78.30±1.56 | 91.33±0.85 | 61.25±10.86 |
| ASDGN | 77.60±1.31 | 91.97±0.85 | 63.50±11.71 |
| DirGNN | 84.80±1.17 | 91.91±1.37 | 72.50±6.61 |
| ACM-GNN | 82.18±2.85 | 91.68±0.38 | 63.75±7.55 |
| DMP | 31.70±7.61 | 86.76±13.76 | 66.25±7.91 |
| GraphSaint | 84.21±1.77 | 91.15±1.09 | 69.50±9.00 |
| FSGNN | 85.46±1.62 | 92.89±1.41 | 73.00±5.35 |
| APPNP | **88.63±0.98** | 92.43±0.62 | 75.75±6.99 |
| GraphTARIF | 81.88±2.00 | 91.27±1.33 | 63.75±7.65 |
| *GraphICL+LM (DeBERTa fine-tuned on GraphICL-enhanced text)* | | | |
| GraphICL | 82.18±1.14 | 93.18±1.59 | 81.50±3.47 |
| *TAPTN+LM (DeBERTa fine-tuned on TAPTN-enhanced text)* | | | |
| TAPTN | 84.32±1.44 | **93.81±0.70** | **89.50±2.59** |

interval has half-width $t_{0.975,4}/\sqrt{5} \approx 1.24$ times the reported standard deviation, so a 95% CI can be read directly off any mean±std entry in Tables 14–17; Table 16 additionally reports the CIs for TAPTN+LM explicitly. To ensure that any residual performance gap reflects differences in structural utilisation rather than differences in textual capacity, training data, or optimisation budget, all pipelines share the same DeBERTa-base text encoder, the same label sets and graph splits, the same edge structure, and an identical hyper-parameter search protocol. Section 4.4 additionally reports an expanded, higher-powered, and bias-controlled comparison that broadens the GNN panel (including the recent linear graph Transformer GraphTARIF (Hu et al., 2026)) and adds a jointly-trained regime in which the text encoder and the GNN are optimised end-to-end.

## 4.2 Results

Table 14 shows that TAPTN+LM clearly improves over the structure-agnostic and unstructured-structure controls: averaged over the three datasets it exceeds TA+LM by +12.36 points and GraphICL+LM by +3.59 points, with the largest margins on ogbn-products (+26.25 and +8.00 points). Against the frozen-feature GNN panel, TAPTN+LM is on par with or better than every architecture on average; it attains the best accuracy on Arxiv-2023 and ogbn-products, and on Cora, among the frozen-feature GNNs, only APPNP significantly surpasses it (paired-$t$-test significance for all configurations is summarised in Table 16). On the larger, heavily sparsified ogbn-products subgraph, every GNN trained on frozen text features degrades substantially (best GNN APPNP 75.75% versus TAPTN+LM 89.50%), indicating that converting edge structure into text via TAPTN's ICL stage is more robust than message passing over a fixed text encoder.

Because all GNN baselines and TAPTN+LM: (i) use the same DeBERTa-base TA encoder, (ii) train on the same labels and graph splits, and (iii) have access to the same edge structure, These gains cannot be explained by more LM parameters or more supervision—both pipelines fine-tune the same LM on the same label sets. The only difference is that TAPTN+LM fine-tunes on structurally enriched texts produced by an explicit structure-aware ICL architecture, whereas TA+LM and GraphICL+LM fine-tune on structure-agnostic or unstructured neighborhood text. In other words, TAPTN's ICL phase successfully converts edge structure (including edge directions and local motifs) into textual representations that make downstream supervised learning easier, even for a relatively small LM. Once equipped with TAPTN-generated structural representations, the LLM's ability to exploit graph structure is on par with that of SOTA GNNs. The performance gap is no longer about "LLMs cannot use edges," but about differences at the level of optimization, parameterization, and inductive bias—all of which TAPTN+LM handles surprisingly well despite being built on an ICL-style textual interface.

## 4.3 Complementary Evidence from Heterophilic graphs

The comparisons in Table 14 are conducted on homophilic graphs. To test whether TAPTN-based LLMs remain structurally competitive when homophily is low and potentially misleading, we further evaluate on three malignant heterophilic WebKB datasets (Texas, Wisconsin, Cornell), whose homophily falls in or near the "mid-homophily pitfall" regime (roughly $(0.1, 0.3)$) where many standard GNNs either fail to benefit from neighborhood aggregation or degrade due to heterophilic mixing.

We reuse the same GNN panel as in the homophilic comparison, comprising heterophily-oriented designs (DirGNN (Rossi et al., 2024), ACM-GNN (Luan et al., 2022), FSGNN (Maurya et al., 2022), AS-DGN (Gravina et al., 2023)), classical message-passing and attention models (GAT (Velickovic et al., 2017), GATv2 (Brody et al., 2022), GraphSAGE (Hamilton et al., 2017), ChebNet (Defferrard et al., 2016), APPNP (Gasteiger et al., 2018), GCNII (Chen et al., 2020), Graphsaint (Zeng et al., 2019), DGI (Velickovic et al., 2019), DMP (Yang et al., 2021)), and the linear graph Transformer GraphTARIF (Hu et al., 2026), all trained on the same frozen TA embeddings. GraphICL+LM was not run on WebKB, which the reviewers did not require.

Table 15 reports the frozen-feature comparison on the three malignant heterophilic datasets. TAPTN+LM remains highly competitive, achieving 94.66% average accuracy—essentially tied with the strongest GNN (DirGNN, 95.83%) and ahead of the heterophily-specialized FSGNN and ACM-GNN (93.69% and 92.90%). It attains the best accuracy on Cornell, ties the best on Texas, and crucially no frozen-feature GNN significantly surpasses it on any of the three datasets under paired $t$-tests; the only significant heterophilic case, a jointly-trained DirGNN on Wisconsin, is reported in Table 17 (see also Table 16). This supports our thesis: once equipped with a reasonably designed ICL architecture, an LLM-based predictor matches specialized GNNs even in the most challenging heterophilic regime, where many standard GNNs collapse (e.g. GAT 66.0%, DMP 45.7% on average).

## 4.4 Statistical Significance and a Bias-Controlled Comparison

We close with two checks that address the reviewers' concerns about statistical power and the fairness of the comparison. As described in the setup, every method is run over five random seeds with mean±std and two-sided paired $t$-tests, the GNN panel includes the recent linear graph Transformer GraphTARIF (Hu et al., 2026), and each GNN is evaluated both on frozen text features and jointly trained with the encoder. Table 16 reports the resulting significance verdicts, and Table 17 reports the jointly-trained accuracies. Over the six datasets, each compared against TAPTN+LM, this amounts to 27–28 competing pipelines per dataset: the GNN panel in both the frozen and jointly-trained regimes, plus the TA+LM and (on the homophilic graphs) GraphICL+LM baselines.

As summarised in Table 16, across the six datasets and the full panel of competing pipelines, only three pipeline–dataset combinations significantly exceed TAPTN+LM under paired $t$-tests: APPNP and a jointly-trained FSGNN on Cora, and a jointly-trained DirGNN on Wisconsin, each by a small margin. On Arxiv-2023, Products, Cornell and Texas, *no* pipeline—including the jointly-trained SOTA models—significantly surpasses TAPTN+LM, and on Arxiv-2023, Products, Cornell and Texas not a single competing pipeline

Table 15: Node classification accuracy (%) on malignant heterophilic WebKB graphs (Texas, Wisconsin, Cornell), mean±std over five random seeds. GNNs are trained on frozen TA embeddings. Avg is the mean over the three datasets. Jointly-trained GNN counterparts and paired-$t$-test significance are reported in Table 17 and Table 16.

| Method | Texas | Wisconsin | Cornell | Avg. |
|---|---|---|---|---|
| *DeBERTa on raw TA (TA+LM)* | | | | |
| TA | 96.54±2.11 | **91.69±2.34** | 95.60±1.67 | 94.61 |
| *Normal GNNs on frozen TA embeddings* | | | | |
| GAT | 71.15±6.80 | 59.69±6.29 | 67.20±6.42 | 66.01 |
| GATv2 | 82.31±4.98 | 85.54±9.08 | 78.00±2.00 | 81.95 |
| SAGE | 95.39±2.58 | 91.08±2.96 | 94.80±3.03 | 93.76 |
| ChebNet | 89.62±3.99 | 84.31±4.27 | 84.80±3.35 | 86.24 |
| DGI | 83.08±3.16 | 77.54±2.34 | 75.60±4.34 | 78.74 |
| *SOTA / heterophily-specialized GNNs on frozen TA* | | | | |
| GCNII | 94.62±2.51 | 90.77±2.43 | 94.40±2.61 | 93.26 |
| ASDGN | 93.84±3.44 | 90.46±1.69 | 93.60±3.85 | 92.63 |
| DirGNN | **97.69±1.61** | 91.39±2.07 | 98.40±1.67 | **95.83** |
| ACM-GNN | 95.77±2.11 | 88.92±3.67 | 94.00±5.10 | 92.90 |
| DMP | 48.08±23.39 | 38.15±13.16 | 50.80±11.63 | 45.68 |
| GraphSaint | 86.16±7.25 | 81.85±11.22 | 77.60±13.52 | 81.87 |
| FSGNN | 95.77±2.51 | **91.69±1.76** | 93.60±2.19 | 93.69 |
| APPNP | 79.23±6.58 | 71.38±6.12 | 78.00±4.69 | 76.20 |
| GraphTARIF | 94.62±3.44 | 90.15±2.80 | 91.60±3.29 | 92.12 |
| *TAPTN+LM* | | | | |
| TAPTN+LM | **97.69±1.61** | 87.08±2.79 | **99.20±1.10** | 94.66 |

even attains a higher mean. Crucially, these conclusions hold under the bias-controlled jointly-trained regime: even when the GNN and its text encoder are optimised together on the same graph and labels, specialised graph models obtain no statistically significant advantage over an LM fine-tuned on TAPTN-enhanced text. This directly addresses the concern that the earlier comparison might have favoured TAPTN+LM by coupling it to a separately fine-tuned encoder.

Table 17 reports the bias-controlled jointly-trained regime. Even when the text encoder and the GNN are optimised end-to-end—the condition most favourable to the GNNs and free of any encoder-decoupling effect—specialised graph models do not gain a statistically significant advantage over TAPTN+LM, with the only exceptions being a jointly-trained FSGNN on Cora and DirGNN on Wisconsin. The most recent linear graph Transformer, GraphTARIF, does not significantly outperform TAPTN+LM on *any* of the six datasets in either the frozen-feature (Tables 14 and 15) or the jointly-trained regime: on ogbn-products

Table 16: Significance summary over five random seeds. For each dataset we count, among all 27–28 competing pipelines (GNN configurations in both the frozen and jointly-trained regimes, plus the TA+LM and, where available, GraphICL+LM baselines), how many *significantly* exceed TAPTN+LM (two-sided paired $t$-test, $p < 0.05$ *and* higher mean) and how many merely have a higher mean. TAPTN+LM accuracy is mean±std with the Student-$t$ 95% confidence interval ($n = 5$).

| Dataset | Homophily | TAPTN+LM | 95% CI | #sig. >TAPTN+LM | #mean >TAPTN+LM |
|---|---|---|---|---|---|
| Cora | homophilic | $84.32 \pm 1.44$ | $[82.53, 86.11]$ | 2 | 9 |
| Arxiv-2023 | homophilic | $93.81 \pm 0.70$ | $[92.94, 94.68]$ | 0 | 0 |
| Products | homophilic | $89.50 \pm 2.59$ | $[86.28, 92.72]$ | 0 | 0 |
| Cornell | heterophilic | $99.20 \pm 1.10$ | $[97.83, 100]$ | 0 | 0 |
| Texas | heterophilic | $97.69 \pm 1.61$ | $[95.69, 99.69]$ | 0 | 0 |
| Wisconsin | heterophilic | $87.08 \pm 2.79$ | $[83.62, 90.54]$ | 1 | 15 |

Table 17: Jointly-trained regime: the DeBERTa text encoder and each GNN are optimised end-to-end on the same graph, removing the bias of training GNNs on embeddings from a *separately* fine-tuned LM. Accuracy (%) is mean±std over five random seeds; the self-supervised DGI and the sampling-based GraphSaint have no jointly-trained variant and are omitted. TAPTN+LM is repeated as a reference (its encoder is not jointly trained with a GNN). Even under these fully consistent training conditions, no GNN—including the SOTA GraphTARIF—significantly surpasses TAPTN+LM except a jointly-trained FSGNN on Cora and DirGNN on Wisconsin (Table 16).

| Method | Cora | Arxiv-2023 | Products | Texas | Wisconsin | Cornell |
|---|---|---|---|---|---|---|
| *Normal GNNs, jointly trained with the LM* | | | | | | |
| GAT | 68.56±22.15 | 36.88±20.63 | 46.50±20.77 | 56.15±24.42 | 47.38±19.71 | 54.80±11.71 |
| GATv2 | 59.85±25.35 | 91.50±1.92 | 22.25±9.82 | 71.54±33.24 | 66.15±19.73 | 66.40±26.74 |
| SAGE | 84.87±2.45 | 93.70±1.18 | 65.50±14.21 | 94.23±3.04 | 89.54±2.01 | 92.00±5.66 |
| ChebNet | 84.24±2.92 | 92.48±1.24 | 67.25±5.96 | 89.23±2.19 | 85.54±4.43 | 89.60±4.56 |
| *SOTA GNNs, jointly trained with the LM* | | | | | | |
| GCNII | 77.82±5.17 | 92.02±0.48 | 66.00±6.02 | 92.69±4.39 | 88.00±3.67 | 90.80±3.35 |
| APPNP | 87.49±2.54 | 85.20±5.39 | 73.25±7.53 | 63.46±13.53 | 60.62±6.49 | 52.00±21.54 |
| ASDGN | 77.23±2.26 | 80.35±18.24 | 64.50±9.30 | 93.46±3.50 | 88.00±3.83 | 92.00±3.74 |
| DirGNN | 86.35±1.29 | 93.12±0.94 | 71.25±7.55 | 96.15±1.36 | **92.00±1.29** | 97.20±1.79 |
| ACM-GNN | 81.33±3.07 | 91.04±4.64 | 65.75±1.90 | 95.77±2.51 | 87.69±6.62 | 92.80±2.28 |
| DMP | 25.24±7.63 | 93.29±1.15 | 58.50±10.21 | 48.85±13.84 | 22.46±15.02 | 41.20±10.92 |
| FSGNN | **87.23±1.65** | 93.70±0.75 | 73.50±7.20 | 96.54±2.11 | 90.15±1.38 | 92.40±3.58 |
| GraphTARIF | 82.40±1.04 | 92.02±1.22 | 64.00±4.87 | 76.54±33.6 | 88.31±4.94 | 86.80±8.44 |
| *Reference* | | | | | | |
| TAPTN+LM | 84.32±1.44 | **93.81±0.70** | **89.50±2.59** | **97.69±1.61** | 87.08±2.79 | **99.20±1.10** |

and on the heterophilic Cornell and Texas graphs it trails TAPTN+LM by wide, significant margins (e.g. $-25.5$ points on Products, $p < 10^{-3}$), and where it leads (Wisconsin) the gap is not significant ($p = 0.08$ frozen, $p = 0.73$ jointly trained). Taken together, the expanded, higher-powered, and consistently-trained evaluation reinforces our central claim: once equipped with TAPTN-generated structural representations, an LLM-based predictor is statistically on par with—and on heterophilic and large-scale graphs often superior to—state-of-the-art GNNs, including the latest graph Transformer architectures.

Table 18: Downstream transferability of the TAPTN representation. "P5" feeds the TAPTN-enriched text (rather than raw text) to an otherwise identical frozen-feature GNN, on the same encoder, graphs, splits and seeds as Tables 14–17. **#(P5>raw GNN)**: number of GNN architectures (of 14) for which P5 *significantly* beats the raw-text GNN (paired *t*-test, $p < 0.05$, higher mean)—the enrichment carries structural signal a convolution on raw text cannot recover. **#(P5>TAPTN+LM)**: number for which a GNN on top of the enriched embedding significantly beats TAPTN+LM. $\Sigma$ totals over the six datasets.

| Dataset | Homophily | #(P5>raw GNN) | #(P5>TAPTN+LM) |
|---|---|---|---|
| Cora | homophilic | 2 | 5 |
| Arxiv-2023 | homophilic | 5 | 0 |
| ogbn-products | homophilic | 11 | 0 |
| Cornell | heterophilic | 4 | 0 |
| Texas | heterophilic | 1 | 0 |
| Wisconsin | heterophilic | 0 | 0 |
| $\Sigma$ | | 23 | 5 |

### 4.5 Two Controls: Transferability of the Representation and Encoder Generality

Finally, two controls—run on the same encoder, graphs, splits and seeds—probe the *nature* and *generality* of this competitiveness, rather than just its existence.

**Is the enriched text structurally informative, and already structure-complete?** We keep the graph, splits, seeds and GNN architectures fixed and change *only* the node input from raw text (the TA+GNN pipeline) to the TAPTN-enriched text encoded by the *same* frozen DeBERTa; call this pipeline P5. Two readings follow (Table 18). First, swapping raw text for the enriched embedding under an otherwise identical GNN *significantly* improves accuracy (paired *t*-test, $p < 0.05$, higher mean) in 23 of 84 (dataset×GNN) cases and never significantly degrades on the homophilic graphs: a graph convolution operating on raw text cannot recover this signal, so the enrichment is *structurally* informative, not merely more fluent text. Second, stacking any GNN—including the linear graph Transformer GraphTARIF—*on top of* the enriched embedding fails to significantly beat TAPTN+LM on 5 of 6 datasets; the structural information a GNN would extract has already been captured by TAPTN's in-context stage. The sole exception is homophilic Cora, where five GNNs add a small significant increment—the same additive APPNP-on-Cora effect of Table 14: on graphs that reward plain neighbour averaging, cheap message passing complements the LLM, whereas on the heterophilic graphs that genuinely test structure use, TAPTN's representation already subsumes the GNN.

**Does the conclusion survive a different encoder?** Repeating the raw-vs-enriched comparison with RoBERTa-base in place of DeBERTa-base leaves the conclusion intact on 5 of 6 datasets—including two of the three heterophilic graphs (Cornell, Texas)—where no raw-text pipeline (neither TA+LM nor any of the twelve TA+GNN) significantly surpasses TAPTN+LM (Table 19; per-pipeline accuracies in Table 20); "TAPTN-enriched ≥ raw" is therefore not a DeBERTa artefact. The lone exception is the small heterophilic Wisconsin graph under RoBERTa, and it is a significance artefact rather than an encoder-specific reversal: the raw−TAPTN+LM margin there is essentially identical under both encoders (+4.6 points, $p = 0.058$ for DeBERTa; +4.9 points, $p = 0.035$ for RoBERTa) and merely crosses the threshold under RoBERTa's lower-variance features. Wisconsin is also the one heterophilic dataset on which a structure-based model significantly beats TAPTN+LM in our main comparison (jointly-trained DirGNN, Table 17), and structure-sensitivity is otherwise intact—reversing its edges still costs 13 points in the in-context setting (Table 29).

## 5 Related Works

ICL-based methods have proven effective in tasks such as Knowledge Graph Question Answering (KGQA) and topological structure understanding. For example, Guo et al. (2023) demonstrated that LLMs can

Table 19: Encoder generality. Number of raw-text pipelines (TA+LM and 12 TA+GNN, 13 total) that *significantly* surpass TAPTN+LM (paired *t*-test, $p < 0.05$, higher mean) under two text encoders. "$\geq$ raw" holds on 5/6 datasets for *both* encoders; the sole exception is the small heterophilic Wisconsin graph ($\sim$51 test nodes) under RoBERTa—a significance artefact (an identical +4.6/+4.9-point margin under both encoders, with $p$ crossing 0.05 only under RoBERTa) on the dataset already identified as our soft spot (Table 17).

| Dataset | Homophily | DeBERTa | RoBERTa |
|---------|-----------|---------|---------|
| Cora | homophilic | 1 | 0 |
| Arxiv-2023 | homophilic | 0 | 0 |
| ogbn-products | homophilic | 0 | 0 |
| Cornell | heterophilic | 0 | 0 |
| Texas | heterophilic | 0 | 0 |
| Wisconsin | heterophilic | 0 | 8 |

Table 20: Per-pipeline node-classification accuracy (%) under the **RoBERTa-base** text encoder, mean±std over five random seeds, on the same datasets, splits and seeds as Tables 14–17. All raw-text pipelines— TA+LM and twelve TA+GNN trained on frozen RoBERTa features—are compared against TAPTN+LM (RoBERTa fine-tuned on the TAPTN-enriched text). Best accuracy per dataset is in bold. [†] marks a raw-text pipeline that *significantly* exceeds TAPTN+LM (two-sided paired *t*-test, $p < 0.05$ and higher mean). Significant cases occur only on Wisconsin (8 of 13 raw pipelines), the heterophilic graph already identified as TAPTN+LM's soft spot (Table 17); on the other five datasets no raw-text pipeline significantly surpasses TAPTN+LM, matching the counts in Table 19.

| Method | Cora | Arxiv-2023 | Products | Cornell | Texas | Wisconsin |
|--------|------|------------|----------|---------|-------|-----------|
| *RoBERTa on raw text attributes (TA+LM)* | | | | | | |
| TA | 63.54±19.81 | 90.75±1.92 | 66.75±7.74 | 93.60±1.67 | 95.00±1.05 | 92.62±2.53[†] |
| *Normal GNNs on frozen RoBERTa embeddings* | | | | | | |
| GAT | 71.51±24.05 | 89.60±1.43 | 62.50±6.79 | 63.20±6.42 | 71.54±7.37 | 65.54±4.56 |
| GATv2 | 71.40±23.93 | 91.68±1.27 | 73.50±6.75 | 79.20±4.60 | 87.31±7.14 | 75.69±8.02 |
| SAGE | 71.77±24.14 | 91.21±0.67 | 71.00±7.26 | 93.60±2.61 | 95.00±1.05 | 92.62±2.01[†] |
| ChebNet | 71.48±24.96 | 90.46±0.87 | 69.00±6.93 | 83.60±4.10 | 88.08±2.50 | 87.08±2.57 |
| *SOTA / heterophily-specialized GNNs on frozen RoBERTa embeddings* | | | | | | |
| GCNII | 66.01±21.14 | 90.69±1.13 | 63.75±8.88 | 92.00±2.83 | 93.46±2.19 | 91.69±3.19[†] |
| DirGNN | 71.51±24.89 | 91.85±1.42 | 71.75±3.60 | 97.20±3.03 | 95.00±2.19 | **93.23±1.37**[†] |
| ACM-GNN | 70.74±23.59 | 91.62±1.65 | 66.75±5.90 | 94.00±2.45 | 94.23±1.36 | 92.62±2.01[†] |
| DMP | 31.55±11.77 | 91.04±1.53 | 63.75±9.64 | 46.80±11.71 | 59.23±10.74 | 40.00±8.91 |
| FSGNN | 74.17±23.48 | 92.48±1.34 | 75.00±4.42 | 92.80±2.28 | 93.46±2.19 | 92.00±2.53[†] |
| APPNP | 75.17±26.08 | 91.85±0.48 | 75.50±6.65 | 74.00±8.37 | 76.54±5.68 | 72.61±5.03 |
| GraphTARIF | 70.19±23.27 | 90.87±0.97 | 66.25±7.18 | 93.20±1.79 | 94.61±1.61 | 92.92±2.33[†] |
| RevGAT | 75.50±12.99 | 92.14±1.03 | 71.75±6.29 | 93.60±1.67 | 94.23±1.36 | **93.23±1.75**[†] |
| *TAPTN+LM (RoBERTa fine-tuned on TAPTN-enriched text)* | | | | | | |
| TAPTN+LM | **81.11±1.73** | **93.35±1.37** | **89.25±2.74** | **98.40±0.89** | **98.08±1.36** | 87.69±3.08 |

extract basic structural information from adjacency lists, but not whether LLMs can exploit it. Sun et al. (2023) showed LLMs performing beam search on knowledge graphs to achieve SOTA KGQA, yet this involved selecting relevant triplets rather than applying graph structure.

Node classification tasks demand effective extraction and utilization of graph structural information through ICL. While Guo et al. (2023) introduced ICL to node classification and found neighborhood information enhancing performance, their accuracy did not exceed 60%. Similarly, Hu et al. (2023) reported only slight

enhancements or even negative effects when incorporating neighborhood information, with GPT-3.5's accuracy on OGBN-Arxiv and Cora not exceeding 65%.

Research on ICL-based node classification has primarily focused on prompt construction, with limited investigation into effectively leveraging graph structural information. Existing methods often depend on high-quality labeling or underperform compared to GNNs. Notably, there are no practical zero-shot ICL-based node classification methods. Chen et al. (2024) found that adding neighborhood information improves accuracy due to homophily but did not clarify whether it was interpreted as graph structure or linear text. Das et al. (2023) compared different modalities, finding text to be most effective, yet the study relied on high-quality labels and did not assess additional benefits from graph structure narratives. Wang et al. (2023) proposed a Retrieval-Augmented Generation (RAG) strategy that outperformed GNNs by creating few-shot exemplars from similar training samples, but it heavily depended on high-quality annotations, lacking zero-shot learning capabilities.

As for exploration of LLMs' ability to leverage graph structural information through ICL, Huang et al. (2024) concluded that LLMs treat neighborhood information merely as linear text through rewiring experiments. GraphICL (Sun et al., 2025) systematizes this direction by proposing a unified ICL benchmark and prompt template for graph reasoning tasks, combining anchor-node text, task description, structure-aware neighbor texts, and labeled demonstrations. Importantly, its ablations highlight that selecting homogeneous neighbors/demonstrations (e.g., most similar or class-aligned) often contributes the largest portion of performance gains, which can reinforce the interpretation that "structure helps mainly as a retrieval mechanism for homophilic texts". However, their methodology was flawed: the absence of performance changes after removing structural descriptions might reflect high homophily rather than an inability to leverage graph structure. The prompt templates they use do not encode edge semantics, and the ICL framework they employ lacks a dedicated mechanism for handling lengthy multi-hop neighborhood contexts and performing graph-structure reasoning. Xu et al. (2025) points out that when LLMs are used as the backbone network, the structural signals encoded by GNNs do not provide any benefit and may even be detrimental. However, they overlooked exploring whether LLMs can understand and utilize the structural information of text attribute graphs within the text space through a specific ICL framework.

## 6 Conclusion

This work revisits the central question: **Can general-purpose LLMs genuinely leverage graph structural information in text-attributed graphs via ICL, once we remove confounding factors and endow them with an architecture explicitly designed for structural reasoning?** Our answer is affirmative.

First, by moving away from high-homophily citation graphs and instead operating on low-homophily WebKB datasets with carefully controlled rewiring, we demonstrate that LLMs are inherently sensitive to graph structure. Flipping first-order edge directions produces consistent and often substantial accuracy drops across seven diverse LLMs, with structural sensitivity strongly correlated with model capability. Extreme two-hop rewiring yields smaller but still universal degradation, with sensitivity more uniform across models due to the coarse, count-based representation of second-order neighborhoods. In all cases, node texts and label distributions are held fixed, so performance differences can only be explained by changes in how the models interpret connectivity patterns, not by differences in homophilic neighbor texts. The main barriers to harnessing this latent structural ability are architectural rather than inherent: traditional GraphICL prompts linearize large ego-networks, lacking a dedicated mechanism for graph structure reasoning and handling lengthy multi-hop neighborhood contexts, and suffering from inconsistent adherence to correct graph structure utilization method.

Second, we show that TAPTN addresses these issues by (i) encoding first-order neighborhoods with a structure-aware template that foregrounds edge directions and roles, (ii) iteratively passing and summarizing textual "messages" to build multi-hop representations, and (iii) stabilizing reasoning with self-generated step-by-step instructions at each aggregation and classification step. Experiments on five TAG benchmarks confirm that TAPTN consistently and substantially improves over zero-hop CoT and GraphICL-style baselines, with gains growing when moving from 1-hop to 2-hop neighborhoods and being especially pronounced

on malignant heterophilic graphs, where homophily-based heuristics are ineffective. Ablation studies further show that naive self-reflection cannot reproduce these gains, whereas structured aggregation and instructions together provide stable benefits and convert structural sensitivity into reliable accuracy improvements.

Third, by aligning LLM and GNN pipelines around the same DeBERTa text encoder, we show that TAPTN's structural utilization capability is competitive with state-of-the-art GNNs. Fine-tuning an LM on TAPTN-enhanced texts (TAPTN+LM) yields performance that matches or exceeds many widely used architectures (e.g., GAT, SAGE, ChebNet) and is statistically indistinguishable from several strong SOTA models on homophilic citation graphs, while clearly outperforming both normal and heterophily-specialized GNNs on malignant WebKB datasets. An expanded, five-seed, and bias-controlled study over six datasets and a thirteen-architecture GNN panel—including a jointly-trained regime that optimises the text encoder and the GNN end-to-end, and the most recent linear graph Transformer GraphTARIF—confirms this conclusion: under paired $t$-tests almost no competing pipeline significantly surpasses TAPTN+LM, and the latest graph Transformer does so on none of the six datasets. Since all methods share the same encoder, labels, and edge structure, the remaining gap cannot be attributed to lacking access to structural information; instead, TAPTN+LM demonstrates that an ICL-style textual interface is sufficient to exploit graph structure information, including edge semantics and local topology, at a level competitive with dedicated graph neural networks under this controlled protocol.

Taken together, these results overturn the view that LLMs "cannot use graph structure" in TAG node classification. Rather, prior negative findings largely stem from homophily-centric datasets and prompt designs that obscure structural cues. Once these confounders are removed and an architecture like TAPTN is introduced to expose and guide structural reasoning, general-purpose LLMs not only exhibit clear structural sensitivity but also translate it into competitive downstream performance. This suggests a practical path forward: instead of replacing graph models outright, we can embed graph structural reasoning into LLM-centric workflows, using TAPTN-style ICL as a zero-shot or low-supervision alternative when labels are scarce, and as a complementary structural encoder when integrating with traditional GNNs. Future work includes extending TAPTN beyond node classification to link prediction and subgraph-level tasks, learning or optimizing instruction policies, and further reducing computational cost for very large graphs, thereby broadening the range of applications where LLMs can serve as structure-aware graph learners.

## Broader Impact Statement

This work makes a methodological contribution to graph reasoning with LLMs and does not raise immediate ethical concerns. Nevertheless, improved structural reasoning over text-attributed graphs could facilitate the analysis of social, financial, or communication networks, and in some settings such capabilities could be misused for large-scale monitoring or profiling of individuals within networked systems. These risks are not specific to TAPTN—they accompany graph learning methods in general—but we encourage responsible use: deployments involving personal data should comply with applicable data-protection regulations, rely on data collected with consent or another appropriate legal basis, and audit downstream decisions that affect individuals. Conversely, the zero-shot nature of the framework lowers labeled-data requirements, which can benefit socially valuable, label-scarce applications such as scientific-literature organization and fraud detection on transaction graphs.

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

**Appendix roadmap.** The appendices follow the evidence chain rather than reviewer order. Appendix A reports scalability, cost, and runtime; Appendix B gives the prompt, task, and instruction templates; Appendix C specifies datasets and splits; Appendix D provides exact LLM inputs and outputs for a representative case; Appendix E details the fine-tuning and GNN implementation; Appendix G collects the current-generation model and structural-channel controls. The remaining appendices summarize prior rewiring variants and the over-smoothing discussion.

## A Time Complexity, Inference Cost, and Runtime Analysis

### A.1 Asymptotic Time Complexity

TAPTN's complexity depends on LLM operations, where output sequence of a node includes LLM-generated categorization and reasons for it while input sequence of a node in the (l+1)-th iteration consists of enhanced text attributes of its first-order neighbors (including original attributes and LLM-outputs of the l-th iteration). For a specific graph, the enhanced text attributes and outputs length of a node float around fixed values $p, q$ respectively. According to Keles et al. (2023), time complexity for a self-attention based model to output a sequence of $m$ tokens is:

$$T_m = O(d \cdot m \cdot n^2), \tag{11}$$

where d is the number of self-attention layers, which is a constant for a specific model.

Thus, for a graph $\mathbf{G} = (\mathbf{V}, \mathbf{E})$, where $\mathbf{V}, \mathbf{E}$ is nodes and edges set of $\mathbf{G}$ respectively, TAPTN's per-node complexity is:

$$O(m \cdot n^2) = O(q \cdot (\frac{|\mathbf{E}|}{|\mathbf{V}|}p + q)^2). \tag{12}$$

As there are $|\mathbf{V}|$ nodes in $G$, the time complexity for a single iteration is:

$$O(|\mathbf{V}| \cdot q \cdot (\frac{|\mathbf{E}|}{|\mathbf{V}|} p + q)^2) = O(\frac{|\mathbf{E}|^2}{|\mathbf{V}|} + |\mathbf{E}|). \tag{13}$$

For $L$ iterations, it becomes $O((\frac{|\mathbf{E}|^2}{|\mathbf{V}|} + |\mathbf{E}|) \cdot L)$. For dense graphs, this scales as $O(|\mathbf{V}|^3 \cdot L)$, manageable for small-to-medium graphs, and for large sparse graphs which more common in real applications, it simplifies to $O(|\mathbf{V}| \cdot L)$. Additionally, as the context window of a specific LLM is limited, when a graph is become so dense that first-order neighborhood descriptions of an average node exceed context window, only a constant number $C$ neighbors will be chosen. In this case, $|\mathbf{E}| = C|\mathbf{V}|$, so its time complexity also simplifies to $O(|\mathbf{V}| \cdot L)$. In conclusion, the time complexity of TAPTN is $O(|\mathbf{V}| \cdot L)$, making TAPTN a efficient zero-shot learning algorithm.

## A.2 Empirical Inference Cost and Runtime Trade-offs

The asymptotic analysis above characterises how TAPTN scales in the number of LLM calls. To make the practical overhead of iterative prompting and instruction generation concrete, we additionally measure the actual token consumption, monetary cost, and wall-clock latency of TAPTN and the single-pass GraphICL baseline on the ogbn-products subgraph—the largest graph we study—under the uniform Llama-3.3-70B-Instruct backbone of Table 12. Token counts are read directly from the prompts and completions exchanged with the API (tokenised with the `cl100k_base` encoding); monetary cost uses the public OpenRouter list prices of \$0.10/\$0.32 per million input/output tokens for Llama-3.3-70B-Instruct and \$0.02/\$0.05 for the Llama-3.1-8B-Instruct that serves as an auxiliary label parser (and, in the budget-backbone configuration of Table 13, as the first-iteration backbone); per-call latency is the median over a small live timing probe.

Two sources of overhead distinguish TAPTN from a single GraphICL prompt. (i) *Instruction generation.* The step-by-step instruction block is a fixed addition to the system prompt; empirically it enlarges the system prompt from $\approx 0.3$k tokens (the instruction-free baseline) to $\approx 2.3$k tokens, i.e. a constant overhead of $\approx 2.0$k input tokens per call. (ii) *Iterative aggregation.* To supply each target node with its neighbours' reasoning, TAPTN first classifies the node's first-order neighbours and then refines, issuing $\approx 4.6$ reasoning calls per test node versus a single call for GraphICL. Combined, TAPTN consumes $\approx 22$k tokens per test node against $\approx 2.8$k for 2-hop GraphICL.

Table 21 translates these quantities into cost and runtime. Under a uniform 70B backbone, TAPTN costs $\approx \$2.5$ per thousand test nodes—about $8\times$ the GraphICL baseline—but buys the +10.25-point accuracy gain of Table 12; this is the operating point we recommend. The dominant cost driver is the iterative neighbour pass rather than the instruction block: the $\approx 4.6$ reasoning calls per node multiply the base cost, whereas the fixed $\approx 2.0$k-token instruction overhead adds only a constant factor to each call. A natural lever for reducing cost is to delegate the first (neighbour-classification) iteration to a smaller model and reserve the 70B model for the final refinement; however, this is not a free lunch—as Table 13 shows, an 8B first pass produces weaker neighbour reasoning and substantially shrinks the gain on this graph, so the cheaper model should be used only when a smaller accuracy improvement is acceptable. In wall-clock terms, the median latency of a 70B reasoning call is 3.5s and of an 8B parsing call 1.1s; at the request concurrency of 40 used in our runs, these latencies imply $\approx 3.5$ minutes to classify the 400-node test set end-to-end with TAPTN ($\approx 0.53$s per node amortised) versus $\approx 45$s for single-pass GraphICL. A measured end-to-end run corroborates the scale of these estimates: the dense-neighbourhood GraphICL configuration of Table 21—whose prompts are roughly three times longer than the sparse baseline's—completed all 400 nodes, including label parsing and retries, in 98s at the same concurrency. Consistent with the $O(|\mathbf{V}| \cdot L)$ analysis, both cost and latency grow linearly in the number of target nodes and iterations, and the per-node figures are stable because the per-call prompt length is bounded by the context window and the constant neighbour cap $C$; the absolute numbers therefore extrapolate directly to larger node sets.

**Is TAPTN over GraphICL economically rational at scale?** The $\approx 8\times$ premium might suggest that GraphICL is the economical choice on large graphs, but for accuracy-sensitive applications the opposite holds, for three reasons. First, the *absolute* cost is small—$\approx \$2.5$ per thousand labelled nodes—so even large

Table 21: Measured inference cost and runtime of 2-hop TAPTN (uniform Llama-3.3-70B backbone) versus the 2-hop GraphICL baseline on the fixed 400-node ogbn-products test set of Table 12; auxiliary label parsing uses Llama-3.1-8B. "Reason calls/node" counts LLM reasoning calls per test node; "Tokens/node" sums input and output over all calls; cost uses OpenRouter list prices ($0.10/$0.32 per M in/out for 70B, $0.02/$0.05 for 8B). TAPTN trades an $\approx 8\times$ cost increase for a +10.25-point accuracy gain. The "dense nbhd." row gives GraphICL the full-graph neighbourhood of each node: this roughly triples its per-call token cost yet leaves accuracy flat, showing that paying for more context—rather than for guided aggregation—does not help.

| Method | Reason calls/node | Tokens /node | Cost ($/1k) | Acc. (%) |
|---|---|---|---|---|
| GraphICL 2-hop | 1.0 | $\approx 2.8$k | 0.31 | 76.50 |
| GraphICL 2-hop (dense nbhd.) | 1.0 | $\approx 8.1$k | 0.91 | 76.75 |
| TAPTN 2-hop (uniform 70B) | 4.6 | $\approx 22$k | 2.52 | **86.75** |

query sets remain inexpensive and the +10.25-point gain is bought for a few dollars per thousand decisions. The marginal rate makes this concrete: relative to single-pass GraphICL, TAPTN's guided aggregation converts $2.21 of extra spend per thousand nodes into 10.25 accuracy points, i.e. $\approx$ $0.22 *per accuracy point per thousand nodes*. The only alternative way to spend more inference budget within the single-pass paradigm—buying more neighbourhood context (the dense row of Table 21)—converts $0.60 into 0.25 points, i.e. $\approx$ $2.40 per point, an order of magnitude worse, and saturates immediately since the context ceiling is already reached. Marginal dollars are therefore $\approx 11\times$ more productive when spent on aggregation than on context. Second, the relevant accuracy-matched alternative on a large graph is *not* GraphICL, which trails by more than ten points, but a trained GNN; TAPTN+LM matches or exceeds strong GNNs on this very graph (Section 4, Table 14: 89.50% versus the best frozen-feature GNN at 75.75%) while requiring no full-graph training and no task labels at inference time. Third, the pay-per-node, linear-in-$|\mathbf{V}|$ cost means one pays only for the nodes that actually need labelling, whereas a transductive GNN recoups its whole-graph training cost only if a large fraction of the graph is labelled. TAPTN is therefore the economically rational option when accurate labels are needed for a bounded set of nodes on a large, weakly-labelled graph; GraphICL remains preferable only where its substantially lower accuracy is acceptable.

**Generality across datasets.** We report these figures on ogbn-products because it is the largest graph we study; the operational cost above is measured on the sparsified 13,482-node subgraph, whose per-node neighbourhood—and hence prompt length and cost—is comparable to the other benchmarks. We do *not* claim products is the most expensive dataset: per-node cost is set by two graph-dependent quantities— the average degree, which fixes the number of neighbour-classification calls, and the per-call prompt length (neighbour text plus the fixed $\approx 2$k instruction block)—and these vary non-monotonically across our datasets (the WebKB graphs have denser *local* neighbourhoods than the products subgraph, whereas the citation graphs carry longer node texts), so no single dataset uniformly dominates. What makes the accounting generalise is therefore not a single worst-case dataset but the constant neighbour cap $C$ introduced in the complexity analysis above: both drivers are bounded by $C$—at most $C$ neighbour-classification calls, each prompt carrying at most $C$ neighbour descriptions plus the fixed instruction block—so per-node cost has a dataset-independent ceiling, and the absolute figures extrapolate rather than explode with graph size. The *dense* full-graph products regime (test-node mean degree $\approx 106 \gg C$) is the empirical instantiation of that ceiling: it is precisely the case where the cap binds, and even there feeding GraphICL the saturated neighbourhood only $\approx$triples its single-pass token cost while leaving accuracy flat (Table 21). Two conclusions thus transfer to every dataset: (i) once $C$ is fixed the cost ceiling is bounded and small, so our figures are representative rather than a one-off; and (ii) the GraphICL$\leftrightarrow$TAPTN accuracy gap does not shrink as more neighbourhood context is bought—it persists even at the cost ceiling—so it is guided iterative aggregation, not neighbourhood volume, that purchases the accuracy gain. The cost objection is correspondingly weakest on the moderate-sized benchmarks that anchor our main GNN comparison.

# B Prompt, Task, and Instruction Templates

## B.1 Prompt Skeleton

> **Prompt Skeleton**
>
> [Task Description Head]
> [Step-by-step Instruction]
> [Neighborhood Description]
> [Task Description Ending].

## B.2 Neighborhood Description Templates

The structure-aware neighborhood description template for Cora and Arxiv-2023 is listed as below:

> **Structure-Aware Template for Citation Graphs**
>
> #### Paper ####
> [content description of the target paper]
> #### Citations ####
> The paper has following citations:
> [content description of each citation]
> #### References ####
> The paper has the following references:
> [content description of each reference]
> , where the content of a paper is described as below:
> ### [paper Number] ###
> ## Abstract ##
> []abstract]
> ## Title ##
> [title]
> Citations Number: [citations number]
> References Number: [references number]
> ## Initial categorization ##
> [initial categorization]
> ## Reasons for initial categorization ##
> [reasons for initial categorization]
> , where paper number describes the relation between the described paper and the target paper (for example, Citation 1). For the first iteration, there won't be the "initial categorization" and "reasons for initial categorization" parts.

The neighborhood description template for Texas, Wisconsin, Washington and Cornell for rewiring experiment in Section 2 is listed as below:

> **Neighborhood Description Template for WebKB Graphs (Rewiring Experiments)**
>
> #### Webpage ####
> ## Incoming links number ##
> []incoming links number]
> ## Outgoing links number ##
> [outgoing links number]
> URL: [URL]
> #### It has inbound links from following webpages: ####

> [description of inbound link]
> #### It has outbound links to following webpages: ####
> where a linked webpage is described as below:
> ### [Webpage Number] ###
> It has outbound link to webpage [URL], which is a []category] page with content abstract as below:
> [content abstract], with link pattern as below: [link pattern]
> , where the content abstract part is only added when its category is "other", and the link pattern part is described as below:
> Inbound links by category:
> [category]: [number of second-order neighbors linking to this webpage belonging to each category]
> Outbound links by category:
> [category]: [number of second-order neighbors linked from this webpage belonging to each category].

The target page itself is not involved in these link pattern. If a first-order page has no hyperlinks except the target page, its link pattern is recorded as "private resource". In the above template, the `category` of every first-order neighbor is shown. As discussed in Section 2.3, this is a structural attribute rather than a homophily shortcut on these low-homophily graphs.

**Label-free WebKB variant.** For the label-free control of Section 2.3, the "which is a `[category]` page" clause is removed for all first-order neighbors and replaced by the neighbor's full content abstract; the link-pattern statistics are retained. Thus, in that control, no first-order neighbor label is exposed to the model, and only neighbor text and link patterns remain.

The structure-aware neighborhood description template for Texas, Wisconsin, Washington and Cornell for TAPTN in Section 3 is listed as below:

> **Structure-Aware Template for WebKB Graphs (TAPTN)**
>
> ## Incoming links number ##
> []incoming links number]
> ## Outgoing links number ##
> [outgoing links number]
> URL: [URL]
> Content Abstract: [content abstract of the target webpage]
> ## Initial categorization ##
> [initial categorization]
> ## Reasons for initial categorization ##
> [reasons for initial categorization]
> #### It has inbound links from following webpages: ####
> [description of inbound link]
> #### It has outbound links to following webpages: ####
> The webpage has the following outbound links:
> [description of each outbound link]
> where a linked webpage is described as below:
> It has inbound/outbound link from/to webpage [URL] with content abstract as below: [content abstract].
> ## Initial categorization ##
> [initial categorization]
> ## Reasons for initial categorization ##
> [reasons for initial categorization].

For the first iteration, there won't be the "initial categorization" and "reasons for initial categorization" parts.

### B.3 Task Description Templates

Task descriptions used for Cora are listed as below:

> **Task Description Head for the First Iteration (Cora)**
>
> Please predict the 2 most appropriate categories for the paper based on the titles and abstracts of the paper itself as well as its references and citations. Choose from the following categories: [candidate options]

> **Task Description Head for Other Iteration (Cora)**
>
> Further revise the initial categorization for the paper. Choose the most appropriate category for the paper from the following categories:[candidate options]Now, apply this method to the following paper. If multiple options apply, ensure these options are sorted from the most relevant to the least relevant.

> **Task description ending (Cora)**
>
> ####Question####: Predict the 2 most appropriate category for the paper. For each category you predict, give a relevance score from 0 and 1. Make double choices from the given list of categories: [candidate options]. Answer: Let's think step by step.

> **Task Description for Parser (Cora)**
>
> Please extract the most appropriate category for the paper. Make single choice from the following categories: [candidate options]. Answer:

Task descriptions used for Arxiv-2023 are listed as below:

> **Task Description Head for the First Iteration (Arxiv-2023)**
>
> Predict the 2 most appropriate arXiv Computer Science (CS) sub-category for the paper based on the titles and abstracts of the paper itself as well as its references and citations. Choose from the following categories: [candidate options]. The predicted sub-category should be in the format 'cs.XX'.

> **Task Description Head for Other Iteration (Arxiv-2023)**
>
> Further revise the initial categorization for the paper. Predict the 2 most appropriate arXiv Computer Science (CS) sub-category for the paper. The predicted sub-category should be in the format 'cs.XX'. Now, apply this method to the following paper. The predicted sub-category should be in the format 'cs.XX'. If multiple options apply, ensure these options are sorted from the most relevant to the least relevant.

> **Task description ending (Arxiv-2023)**
>
> ####Question####: Predict the 2 most appropriate category for the paper. For each category you predict, give a relevance score from 0 and 1. Make double choices from the given list of categories: [candidate options]. Answer: Let's think step by step.

> **Task Description for Parser (Arxiv-2023)**
>
> Please extract the refined most appropriate category for the paper. Choose from the following categories: [candidate options]. Answer:

Task descriptions used for Texas, Wisconsin, Washington and Cornell are listed as below:

> **Task Description Head for Rewiring Experiments (WebKB)**
>
> Please predict the 2 most appropriate categories for the webpage based on the URL and category of the webpages which the target page has outbound links to or has inbound linked from (for the non-main-page or resource pages labeled as 'other' within these webpages, their content abstract will be attached in addition) as well as the link pattern (inbound and outbound hyperlinks) of the target webpage. Choose from the following categories: [candidate options].

> **Task Description Head for TAPTN (WebKB)**
>
> Please predict the 2 most appropriate categories for the webpage based on the URL and content abstract of the webpage as well as its linked pages which the target page has outbound links to or has inbound links from. Choose from the following categories: [candidate options].

> **Task description ending (WebKB)**
>
> ####Question####: Predict the 2 most appropriate categories for the webpage with URL: . For each category you predict, give a relevance score from 0 and 1. Make double choices from the given list of categories: [candidate options]. Answer: Let's think step by step.

> **Task Description for Parser (WebKB)**
>
> Please extract the category with highest relevance score for the webpage. Make single choice from the following categories: [candidate options]. Answer:

## B.4 Step-by-Step Instructions

The step-by-step instruction used for Cora and Arxiv-2023 in the first iteration is listed as below:

> **Step-by-step Instruction for the First Iteration (Citation Graphs)**
>
> Choosing the most appropriate category for a paper based on the titles and abstracts of the paper and its references and citations involves a process of identifying key themes, methods, and subject areas. Here is a structured approach to help you categorize a paper:
> 1. **Understand the Categories**: First, familiarize yourself with the predefined list of categories available. Understand what each category entails, including the typical methodologies, subject areas, and scopes covered by each.
> 2. **Analyze the Paper's Abstract and Title**:
> - **Keywords and Phrases**: Identify key terms and phrases in the title and abstract. These often indicate the central themes and the discipline.
> - **Objective and Approach**: Look for statements about the paper's main objectives and the methods used. This can give clues about whether the paper is theoretical, empirical, review-based, etc.
> - **Subject Area**: Determine which area of study the paper belongs to based on the problems addressed and the context provided.
> 3. **Examine References**:

- **Reference Sources**: Review the titles and abstracts of the cited papers. Papers often cite sources within the same or a closely related field.
- **Patterns in Citations**: Note any recurring themes or predominant disciplines in the citations. This can indicate the community and academic discourse the paper is engaging with.
4. **Check Citations to the Paper**:
- **Citing Papers' Focus**: Look at what aspects of the paper are being cited by other authors and in what context. This might provide insights into the paper's contributions and relevance in specific fields.
5. **Synthesize the Information**:
- **Majority Rule**: If the majority of references and citations belong to a particular category, there's a good chance the paper fits there too.
- **Consistency Check**: Ensure the identified category aligns with the paper's objectives and methods.
- **Broader Context**: Consider if the paper might be interdisciplinary and whether it should be categorized under a more general or specific field based on its breadth and focus.
6. **Make a Decision**:
- **Best Fit**: Choose the category that best captures the essence of the paper based on the synthesis of the above steps.
- **Fallback Option**: If unsure, lean towards a broader category that encompasses multiple aspects of the paper.
7. **Document the Reasoning**: It's useful to keep notes on why you categorized the paper in a certain way, especially if the decision was not straightforward. This helps in maintaining consistency when categorizing other papers.
8. **Final Check**: After final decision, review the reasoning process and final categorization carefully and identify any factual errors, inconsistencies, or missing important information. If you find any issue, please fix it accordingly to ensure it logically fits with its content and its scholarly context. This final check ensures that the category reflects the paper's contributions and themes accurately. This method relies heavily on critical thinking and a good grasp of the subject areas represented in your category list. It requires an analytical approach to text and the ability to discern patterns and themes from limited information.

The step-by-step instruction used for Cora and Arxiv-2023 in other iterations is listed as below:

---

**Step-by-step Instruction for Other Iteration (Citation Graphs)**

To further revise the categorization of a paper based on the citation network and the initial categorizations, follow this structured method:
1. **Examine the Citation Network**:
- **Analyze Connections**: Look at how the paper is connected within the citation network. Identify whether it is primarily citing or being cited by papers within specific categories.
- **Identify Influential Papers**: Determine which papers in the citation network are highly influential or frequently cited. These papers can often guide you towards the core category of the subject matter.
2. **Compare Initial Categorizations**:
- **Consistency Check**: Check if the initial categorizations of the papers within the citation network align with the initial categorization of the target paper. A strong alignment suggests a correct initial categorization.
- **Majority Rule**: If the majority of the papers in the citation network belong to a particular category, this might indicate the central focus area for the target paper.
3. **Review Reasoning for Categorizations**:
- **Justifications**: Evaluate the reasons given for the initial categorizations of the papers in the citation network. Strong, well-articulated justifications can help validate the categories.
- **Identify Common Themes**: Look for common themes in the reasoning provided. If similar

reasons are repeatedly used for categorizing papers into a specific category, this strengthens the case for that category.

4. **Cross-Referencing Themes**:

- **Abstract and Title Analysis**: Re-examine the abstracts and titles of the target paper and the papers in its citation network. Look for shared keywords, phrases, and thematic overlaps.

- **Methodologies and Approaches**: Compare the methodologies and approaches described in the abstracts. Similar methods often indicate similar categorical alignment.

5. **Consider the Influence of Interdisciplinary Connections**:

- **Broader Context**: Determine if the paper spans multiple disciplines. If it does, consider which categories are most relevant based on the depth and focus of the interdisciplinary connections.

- **Primary vs. Secondary Categories**: If the paper is highly interdisciplinary, you might need to choose a primary category that best represents the core contribution and a secondary category for the supporting discipline.

6. **Iterative Adjustment**:

- **Re-Evaluate Initial Judgment**: Based on the analysis of the citation network and the comparisons made, re-evaluate the initial categorization.

- **Revise if Necessary**: Adjust the category if the evidence from the citation network strongly supports a different categorization.

7. **Final Decision**:

- **Best Fit Category**: Choose the category that now seems to best capture the essence of the paper, considering the additional information from the citation network.

- **Document the Revision**: Make notes on why the revision was made, including the influence of the citation network and any key papers that led to the change in categorization.

8. **Final Review**:

- **Self-Check**: Review the analysis process and the final categorization decision carefully and identify any factual errors, inconsistencies, or missing important information. If you find any issue, please fix it accordingly to ensure that the analysis process is logically correct and fitting with the paper itself as well as its citation network, the final categorization decision aligns with the overall analysis and that the paper is placed where it best fits within the academic landscape.

By systematically analyzing the citation network and considering the broader context provided by the initial categorizations and reasons, you can refine and improve the judgment of the most appropriate categories for the paper. This approach ensures that the categorization is robust, well-justified, and reflective of the paper's true academic context.

The step-by-step instruction used for Texas, Wisconsin, Washington and Cornell is listed as below:

---

### Step-by-step Instruction (WebKB Graphs)

To classify these pages manually without the aid of a machine learning model, you can follow a systematic approach by analyzing the available data (link patterns and content abstracts for "other" pages). Here's a step-by-step guide to help you with the classification process:

### Step 1: Analyze Link Patterns

Link patterns can provide significant clues about the category of a webpage. Consider both the inbound and outbound links:

1. **Inbound Links:**

- **From Faculty Pages:** Indicates the target page may be important for faculty members, possibly related to faculty, projects, or research. Sometimes students also have a single inbound link from faculty.

- **From Student Pages:** Indicates relevance to students, potentially a course, faculty, department, or project page.

- **From Course Pages:** Likely indicates a relationship with academic courses for example, faculty, student or course materials. - **From Department Pages:** May suggest the page is important for

the entire department, possibly an administrative or project page.
- **From Staff Pages:** Could imply relevance to staff-related activities.
- **From Project Pages:** Suggests involvement in specific projects, may be faculty or students.
- **From Other Pages:** Determine according to the content of this "other" page. For example, inbound links from webpages that list information of graduate students (such as student directories) indicate it's a student page, inbound links from course list indicate it's a course page, etc.
2. **Outbound Links:**
- **To Faculty Pages:** The target page could be related to faculty activities or information, for example, it may be a student page the faculty is supervising, the project page the faculty is leading, or the course page the faculty is teaching, etc.
- **To Student Pages:** The target page might be course, faculty pages, or it providing resources or information useful/related to students such as students directories.
- **To Course Pages:** Likely to be faculty/student pages as teacher/TA, or it is related to academic content or course information.
- **To Department Pages:** The personal mainpages (staff/faculty/student) has outbound links back to department mainpage. It indicates a broader departmental focus if it's an "other" page.
- **To Staff Pages:** Might be providing staff-related resources or information.
- **To Project Pages:** Suggests the target page is project-related, for example, it's a faculty page or student page as a participant.
- **To Other Pages:** Consider the content of the "other" page to infer the target page's category. For example, outbound links to research publications indicate a faculty/student page, outbound links to miscellaneous personal content indicate a student page, outbound links to administrative documents indicate a staff page, outbound to course materials indicate a course page, etc.
### Step 2: Content Abstract Analysis (for "Other" Pages)
For pages labeled as "other" with attached content abstracts, examine the abstracts closely to understand the primary focus of the page. Look for keywords and phrases that indicate:
- **Academic Terminology:** Course names, academic terms, syllabus details, etc., indicate course pages.
- **Research Terminology:** Research interests, publications, projects, etc., suggest faculty or project pages.
- **Administrative Language:** Departmental policies, staff roles, administrative announcements, etc., indicate department or staff pages.
### Step 3: Contextual Linking and Category Inference
Integrate the insights from link patterns and content abstracts to infer the category:
1. **Faculty Pages:**
- High number of inbound links from course, project, students or other faculty pages.
- Outbound links to research interests, publications, projects related content, courses related content or departmental resources. Sometimes to student pages.
2. **Student Pages:**
- Inbound links from course, project or other student pages. Often with an inbound link from webpages that list information of graduate students (such as student directories). Sometimes a single inbound link from faculty page.
- Outbound links to course-related content, project related content, faculty members or miscellaneous personal content. Sometimes with a single outbound link to department mainpage.
3. **Course Pages:**
- Inbound links from student pages or faculty pages.
- Outbound links to faculty members, students, syllabi, and academic resources.
4. **Department Pages:**
- High number of inbound links from all categories (faculty, student, staff, etc.).
- Outbound links to department-wide resources (including content related to research, students, faculty, course, administrative, etc.).
5. **Staff Pages:**

- Inbound links from departmental or faculty pages.
- Outbound links to administrative documents, departmental resources.
6. **Project Pages:**
- Inbound links from faculty and student pages.
- Outbound links to research-related content, publications, participants (faculty/student) and external resources.
7. **Other Pages:**
- Typically have content abstracts attached.
- Serve as auxiliary pages linked primarily to a main category page (faculty, student, etc.).
### Step 4: Cross-Verification
Verify your initial classification by cross-referencing with multiple link patterns and content abstracts. If a page seems ambiguous, check the link patterns again and reassess based on the most frequent category of linking pages.
### Step 5: **Final Review**:
Review the analysis process and the final category carefully and identify any factual errors, inconsistencies, or missing important information. If you find any issue, please fix it accordingly to ensure that the analysis process and the final category of target page are logically correct and fitting with its context.
By carefully analyzing the link patterns and content abstracts, you can systematically classify the webpages into their respective categories.

## C   Dataset Details and Splits

This appendix summarizes the datasets, preprocessing choices, and train/eval/test splits used in the experiments.

- WebKB dataset: Consisting of Cornell, Texas, Washington and Wisconsin. This data set contains WWW-pages collected from computer science departments of various universities in January 1997 by the World Wide Knowledge Base project of the CMU text learning group. The pages were manually classified into the 7 categories: student, faculty, staff, department, course, project and other. The class other is a collection of pages that were not deemed the "main page" representing an instance of the previous six classes. For example, a particular faculty member may be represented by home page, a publications list, a vitae and several research interests pages. Only the faculty member's home page was placed in the faculty class. The publications list, vitae and research interests pages were all placed in the other category. We just evaluate different methods on the first 6 categories as the intra-class similarities of both hyperlink pattern and content of pages in "other" category are weak, thus they can be categorized as pages in the first 6 categories reasonably. What's more, since most of the pages are labeled as "other", Differences in the classification performance of pages in the first 6 categories will have almost no impact on total accuracy if pages in "other" category are evaluated.

- Cora dataset: The Cora dataset consists of scientific papers categorized into different classes. The primary data structure includes a citation graph where each node represents a paper, and each directed edge represents a citation from one paper to another. The dataset includes both the text content of the papers (which is typically represented as a bag-of-words) and the citation relationships between them. The Cora dataset was originally constructed by McCallum et al. (2000) as part of their work on citation indexing and clustering. The papers were collected from the research literature on topics like machine learning, data mining, and other related fields. The papers were manually categorized into 7 classes based on their research topics, specifically Case-Based, Genetic Algorithms, Neural Networks, Probabilistic Methods, Reinforcement Learning and Theory.

- Amazon dataset: This dataset is a node classification dataset that originates from a large Amazon product co-purchasing network. Each node in the graph represents a product, and an edge between

two nodes signifies that the two products are frequently purchased together. The dataset includes products categorized into 47 top-level categories on Amazon platform. This dataset was initially introduced by Yang & Leskovec (2012) with primary goal of developing a heuristic parameter-free community detection method that easily scales to networks with more than hundred million nodes. The subgraph used in our ICL experiments is constructed from the full ogbn-products curation of this network (Hu et al., 2020) (2,449,029 nodes, mean degree $\approx 50.5$) as follows: we uniformly sample 50,000 nodes (fixed seed), take the induced subgraph—retaining only edges whose *both* endpoints are sampled—symmetrise it, and keep the largest connected component, yielding 13,482 nodes and 18,761 undirected edges (37,522 directed edge entries, the convention of Table 22; mean degree $\approx 2.8$). Induced-subgraph sampling removes every edge with an unsampled endpoint and therefore sparsifies neighbourhoods by construction; this is deliberate: it bounds the number of neighbour descriptions per prompt and the number of LLM calls per node, keeping inference cost controllable (Appendix A). The dense full-graph control in Table 12, which feeds GraphICL each node's original full-graph neighbourhood, verifies that this sparsification does not artificially favour TAPTN over the baseline.

- Products (ogbn-products) dataset: For the GNN comparison in Section 4 we use ogbn-products (Hu et al., 2020), the standardized Open Graph Benchmark curation of the Amazon product co-purchasing network, in which nodes are products described by their textual metadata and edges connect frequently co-purchased products—a widely adopted, homophilic large-scale benchmark. Concretely, the comparison runs on the same 13,482-node connected subgraph described in the Amazon item above; the fixed 400-node pool used by the ICL experiments is re-split 60/20/20 (240/80/80) under each random seed, so that the ICL and supervised evaluations share an identical node population. We use it to assess whether the parity between TAPTN+LM and GNNs persists beyond the small citation graphs.

- Arxiv-2023 dataset: This dataset was constructed by Huang et al. (2024). They sampled 668 test nodes from about 46,000 arXiv CS papers published between January 1 and August 22, 2023, and extracted references to identify one-hop and two-hop neighbors by searching for valid arXiv IDs and titles via the arXiv API. To comply with API rate limits, each paper was limited to 30 searches, and unmatched pre-2019 papers were excluded. Using this data, they constructed a citation network, where one-hop references were identified through both arXiv ID matching and title searching, and two-hop references were determined solely by arXiv ID matching. Dataset statistics reveal similar test node degrees between ogbn-arxiv and arxiv-2023. We selected a subgraph consisting of 7 categories for evaluation (specifically, "cs.GT", "cs.MA", "cs.RO", "cs.NE", "cs.IR", "cs.SI", "cs.CY"), then edges that contains nodes that not belong to these categories were deleted, finally we removed nodes with no neighbors. The consideration for the categories selection include that nodes number of these categories are close to each other, and their connotation partially overlap, ensuring a classification method must understand the semantic of every category precisely to achieve a good performance.

The detailed statistics of the datasets we used are presented in Table 22.

## D   Worked Examples: Exact LLM Inputs and Outputs

To make the prompting protocol fully concrete, this appendix traces the *exact* input messages and model outputs for a single representative node, classified by `Llama-3.3-70B-Instruct`. We use the same test node throughout: the Texas (WebKB) page `.../users/markus`, whose ground-truth label is **student**. The two examples below are reconstructed deterministically from the stored experiment artifacts; reasoning traces are quoted verbatim but abbreviated with "[. . . ]" for space, and the full transcripts are released with our code. The neighbourhood of this node is identical across all conditions and is summarised in Table 23; only the *direction* of each edge (which neighbours are inbound vs. outbound) is changed by the rewiring perturbation. Note that the neighbour descriptions never reveal a neighbour's category — only its URL and a content abstract — so the prompts are label-free in the sense of Section 2.3.

Table 22: The detailed statistics of the datasets used in experiments. Train/Eval/Test splitting is not applicable for the Washington dataset, which is not used for LM fine-tuning. For Texas, Wisconsin and Cornell datasets, the three parts split by slashes are the number of nodes for rewiring experiments in Section 2, ICL-based methods in Section 3 and finetuned LMs in Section 4 respectively. For Amazon, ICL-based methods are evaluated on a fixed 400-node pool; the GNN comparison of Section 4 re-splits this same pool into 240/80/80 train/eval/test under each random seed.

| Datasets | #Nodes | #Edges | #Train Nodes | #Eval Node | #Test Node |
|---|---|---|---|---|---|
| Cornell | 800 | 1625 | 0/0/149 | 0/0/50 | 247/248/49 |
| Texas | 788 | 1731 | 0/0/154 | 0/0/51 | 253/256/51 |
| Washington | 1119 | 2372 | / | / | 257 |
| Wisconsin | 1208 | 3464 | 0/0/193 | 0/0/64 | 314/321/64 |
| Cora | 2708 | 5429 | 270 | 542 | 542 |
| Arxiv-2023 | 2389 | 5153 | 1124 | 287 | 315 |
| Amazon | 13482 | 37522 | / | / | 400 |

Table 23: The (label-free) neighbourhood of the target node `.../users/markus` used in the worked examples. The same five neighbours and the same content abstracts are shown in every condition; the rewiring perturbation only swaps the inbound/outbound role of each edge. The `.../users/psp` research-group page is a reciprocal neighbour (both inbound and outbound), so it is unaffected by reversal.

| Neighbour (abbrev.) | Role (directed) | Gist of content abstract |
|---|---|---|
| `/docs/grad.html` | inbound | lists names/logins of UTCS *graduate students* (a student directory) |
| `/users/psp` | in + out | PSP research group (Programs, Specifications, Proofs) |
| `/users/emerson` | outbound | E. A. Emerson, Centennial *Professor* (a faculty page) |
| `/mac/main.html` | outbound | UT Macintosh freeware/shareware archive |
| `www.cs.utexas.edu` | outbound | the CS *department* main page |

### D.1 Example 1: Structure-aware TAPTN, original vs. edge-reversed graph

The model receives the WebKB step-by-step instruction (the "Step-by-step Instruction (WebKB Graphs)" box above) as its system message, followed by a user message that enumerates the inbound and outbound neighbours with their content abstracts. The *only* difference between the two conditions below is that the edge directions are reversed (the directed-graph inbound set {`grad.html`, `psp`} becomes the outbound set, and vice versa); the neighbour texts are byte-for-byte identical.

---
**Example 1 — Input/output, ORIGINAL (directed) graph**

[**system**] *(WebKB step-by-step instruction, abbreviated)* "Step 1: Analyze Link Patterns [...] From Other Pages: [...] inbound links from webpages that list information of graduate students (such as student directories) indicate it's a student page [...]"

[**user**] URL: `.../users/markus`. *Inbound from:* `/docs/grad.html` ("lists [...] UTCS Graduate Students"), `/users/psp` (PSP research group). *Outbound to:* `/users/emerson` (faculty), `/mac/main.html`, `www.cs.utexas.edu` (department), `/users/psp`. "Predict the 2 most appropriate categories [...] Let's think step by step."

[**output**] "[...] The webpage has an inbound link from `/docs/grad.html`, which lists UTCS Graduate Students. This suggests the target page is likely related to a graduate student. [...] The link from the graduate student list strongly suggests the page is about a student." ⇒ **Student** (0.8). **Correct.**

---

---

**Example 1 — Input/output, REWIRED (edges reversed) graph**

[**system**] *identical to the box above.*

[**user**] URL: `.../users/markus`. *Inbound from:* `/users/emerson` (faculty), `/mac/main.html`, `www.cs.utexas.edu` (department), `/users/psp`. *Outbound to:* `/docs/grad.html` ("lists [...] UTCS Graduate Students"), `/users/psp`. *(Same five neighbours and abstracts; only the inbound/outbound roles are swapped.)*

[**output**] "[...] inbound links from `/users/emerson` (a faculty page) [and] `www.cs.utexas.edu` [the department] [...] the outbound link to the graduate students' page [...] the page seems to be related to an individual with a significant role within the department, possibly involved in research and supervision." ⇒ **Faculty**. **Incorrect.**

---

Because the neighbour set and all neighbour texts are held fixed, the flip from **Student** to **Faculty** is attributable *solely* to edge direction: the model reads "a graduate-student directory links *to* me" as evidence of being a student, but "I link *to* a graduate-student directory (while faculty/department pages link to me)" as evidence of being a supervising faculty member. This directly demonstrates that the model consumes directional structure rather than merely the bag of neighbour texts.

### D.2  Example 2: Same node, label-free probe without the step-by-step instructions

To isolate the contribution of the structure-aware instruction template, we feed the *same* (original, directed) label-free neighbourhood but replace the system message with the plain task description used by the rewiring probe of Section 2 (no step-by-step guidance on how to read link patterns).

---

**Example 2 — Input/output, plain (no step-by-step) system message**

[**system**] "Please predict the 2 most appropriate categories for the webpage based on the URL and content abstract of the webpage as well as its linked pages [...]" *(no guidance on interpreting link direction.)*

[**user**] *identical directed neighbourhood as in Example 1 (original graph).*

[**output**] "The inbound link from `/docs/grad.html` [...] might initially suggest a connection to the 'student' category. However, [...] possibly a graduate student who is also a teaching assistant, researcher, or transitioning into a faculty/staff role. [...] Faculty (Relevance Score: 0.8): The connections to other faculty members, involvement with research groups, and the structure of the URL suggest a strong likelihood that the webpage belongs to a faculty member." ⇒ **Faculty**. **Incorrect.**

---

On the *same* correctly-oriented evidence that Example 1 resolved to the right answer, the plain prompt misreads the decisive inbound link from the graduate-student directory and predicts **Faculty**. The structure-aware step-by-step instruction is therefore not cosmetic: it is what lets the model convert the available directional signal into the correct decision, consistent with the ablation in Section 3 and Appendix G.4.

## E  Fine-Tuning and GNN Implementation Details

We evaluated LMs fine-tuned on TAPTN-generated enhanced text attributes against GNN-based models trained on the original text attributes, on six datasets (Cora, Arxiv-2023, ogbn-products, Texas, Wisconsin and Cornell). Every method is run with five random seeds, and we report mean±std with two-sided paired $t$-tests; each GNN is evaluated both on frozen text embeddings and jointly trained end-to-end with the encoder. The GNN architectures and training hyper-parameters are summarized in Tables 24 and 25, respectively. As our training pipelines are modified based on the programs provided by He et al. (2023), their training hyper-parameters are also adopted by us.

Table 24: The detailed architectures of each GNNs. The hidden dims of SAGE and GAT are set to 1024 as their fitting abilities are not as powerful as SOTA GNNs. Number of convolutional layers of all GNNs are set to 2, ensuring the order of neighborhood they consider are the same as TAPTN. "/" means this item is not applicable.

| GNN | Hidden Dim | #Layers | Special Architecture |
|---|---|---|---|
| GAT | 1024 | 2 | Number of heads is set to 8. |
| SAGE | 1024 | 2 | / |
| GATv2 | 128 | 2 | / |
| GCNII | 128 | 2 | / |
| ASDGN | 128 | 2 | We select GAT as its aggregation function. In ASDGN, the hidden dim are the same as input dim, so we first map the inputs to the hidden dim using MLP. |
| DirGNN | 128 | 2 | The basic convolutional layer we select is GCN. |
| APPNP | 128 | 2 | The teleport probability is set to 0.1. |
| DGI | 128 | 2 | We select GCN with hidden dim 128 as the encoder, and mean function as readout function. |
| ChebNet | 128 | 2 | Chebyshev filter size is set to 3. |
| Graphsaint | 128 | 2 | Implemented by GATv2 combining a Graphsaint random walk sampler. |
| DMP | 128 | 2 | / |
| FSGNN | 128 | 2 | / |
| ACM-GNN | 128 | 2 | / |
| GraphTARIF | 128 | 2 | Linear attention augmented by a gated local GAT branch (to raise the attention-map rank) and a learnable log-power function on the attention scores (to lower entropy) (Hu et al., 2026). |

Table 25: Hyper-parameters for training GNNs and fine-tuning LMs. "/" means this item is not applicable.

| Parameter | LM | GNN |
|---|---|---|
| Learning Rate | 2.02e-5 | 0.01 |
| Weight Decay | 0.0 | 0.0 |
| Dropout | 0.3 | 0.0 |
| #Training Epochs | 15 | 200 |
| Batch Size | 4 | / |
| Early Stop Patience | / | 50 |
| Attention Dropout | 0.1 | / |
| Classifier Dropout | 0.4 | / |

## F    Additional Rewiring Methods from Prior Work

"Random" keeps 1-hop neighbors and randomly reconnects 2-hop neighbors to 1-hop neighbors; "extreme" retains 1-hop neighbors and connects all 2-hop neighbors to a random 1-hop neighbor; and "path" connects all 1-hop neighbors into a path, as shown in Figure 11.

## G    Extension to Current-Generation Models

Reviewer feedback raised whether our findings hold for more recent and stronger LLMs, and anticipated that structural in-context learning should improve with capability while the marginal benefit of TAPTN

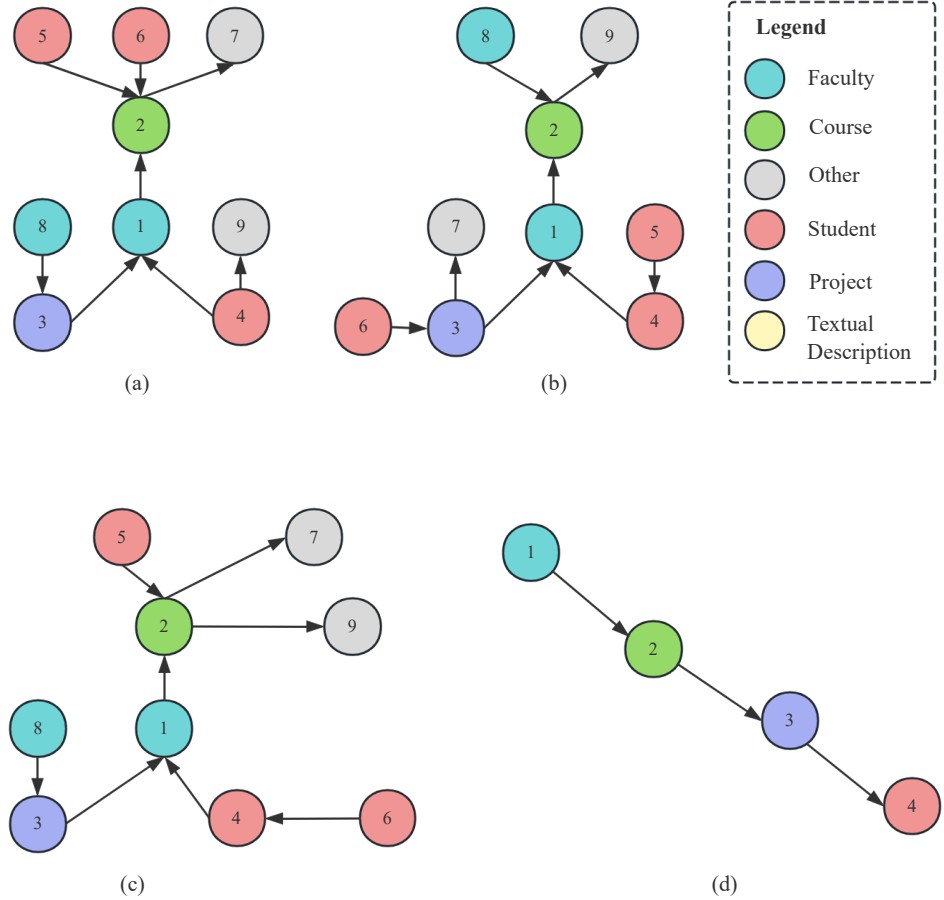

Figure 11: Examples of prior two-hop rewiring methods on the Texas dataset. "Random" keeps 1-hop neighbors and randomly reconnects 2-hop neighbors to 1-hop neighbors; "extreme" keeps 1-hop neighbors and connects all 2-hop neighbors to a random 1-hop neighbor; "path" connects all 1-hop neighbors into a path.

may shrink. This appendix reports an additive study addressing exactly this: all results here are new, and every table, figure, and statistic in the main text is unchanged. We add four 2025–26 models from four distinct families, with LMArena text-overall scores read from the same snapshot used for the original panel: GPT-OSS-120B (1353), Qwen3.5-27B (1409), Gemma-4-31B-it (1451), and the open-weight flagship GLM-5.1 (1475). They extend the capability axis from the original 1224–1319 band up to 1475. Mixed-thinking models are called in non-thinking/instruct mode for comparability with the original instruct-mode panel.

### G.1 Structural sensitivity persists under rewiring

We repeat the label-free flipping-rewiring control (neighbour categories removed; only neighbour text and inbound/outbound link-pattern statistics are shown) on all four WebKB graphs. Table 26 reports the four-dataset averages. Every new model degrades when structure is perturbed: 15 of the 16 model–dataset pairs drop, with a single tie. Pooled with the original seven models, 10 of 11 models degrade (sign test $p = 0.012$); the only exception remains GPT-3.5-Turbo, already identified as the weakest model that barely exploits structure. Structural sensitivity therefore does not disappear in newer and stronger models.

Table 26: Current-generation models under the label-free flipping-rewiring control on WebKB. $\bar{O}, \bar{R}$ are accuracy (%) averaged over the four datasets before/after rewiring; $\Delta = \bar{R} - \bar{O}$ (points); Err. inc. is the relative increase in error rate (ceiling-normalized). Models are ordered by LMArena score.

| Model | Arena | $\bar{O}$ | $\bar{R}$ | $\Delta$ | (Err. inc.) |
|---|---|---|---|---|---|
| GPT-OSS-120B | 1353 | 83.84 | 79.67 | $-4.17$ | $(+26\%)$ |
| Qwen3.5-27B | 1409 | 93.33 | 86.01 | $-7.31$ | $(+110\%)$ |
| Gemma-4-31B-it | 1451 | 96.55 | 94.67 | $-1.88$ | $(+55\%)$ |
| GLM-5.1 | 1475 | 95.17 | 93.48 | $-1.69$ | $(+35\%)$ |

Table 27: Structural-channel decomposition on Texas (accuracy %). The **ego** column uses the target-page text only (0-hop); the other three use the same first-order (1-hop) neighbourhood and differ only in the structural channel exposed. **GraphICL**: edges anonymised, so neighbours appear as an unordered bag of text with no direction or role; **GraphICL+SAT**: the full directed structure-aware template—the same baseline configuration as in Table 28, reported there at 2-hop; **TAPTN**: the directed template plus instruction, with a single fixed instruction shared across every row including the anchor. The first row is the original weak Llama backbone: for it the template is not yet usable (anonymised neighbours fall even below ego), whereas the current-generation models gain $+23$–$24$ points from the directed template alone.

| Model | ego | GraphICL | GraphICL+SAT | TAPTN |
|---|---|---|---|---|
| | 0-hop | 1-hop, anon. | 1-hop | 1-hop |
| Llama-3.3-70B (anchor) | 66.80 | 45.31 | 64.06 | 87.11 |
| Gemma-4-31B-it | 71.88 | 92.19 | 95.70 | 96.48 |
| Qwen3.5-27B | 68.75 | 69.92 | 92.19 | 94.92 |
| GLM-5.1 | 83.59 | 92.97 | 94.53 | 96.09 |

We further relate sensitivity to capability under two complementary metrics, reported together for transparency. Under the *absolute* accuracy drop, the strong linear capability–sensitivity correlation of the original seven-model panel ($r = 0.85$, $p = 0.016$) weakens once the four new models are pooled in ($r = 0.34$, $p = 0.31$). This is a ceiling artifact rather than a vanishing relationship: the strongest new models already score 0.95–0.97 without rewiring, so the absolute headroom for a drop is bounded by construction. Under the standard ceiling-normalized metric—the relative increase in error rate—the relationship is significant on the full eleven-model panel (Pearson $r = 0.64$, $p = 0.035$; Spearman $\rho = 0.71$, $p = 0.015$) and stronger still on the original seven ($r = 0.92$, $p = 0.003$). The strongest models remain clearly sensitive to structural perturbation (error rate up by 35–55%) while recovering part of their accuracy from neighbour text, consistent with the expectation that more capable models lean more on structure where it is informative.

## G.2 Current LLMs actively use the structure-aware template

To localize where the gains originate, we decompose the prompt into structural channels on Texas (Table 27). A text-only prompt that sees only the target page (ego, 0-hop) reaches 69–72% for the mid-tier models; adding the structure-aware-template neighbourhood description *without any instruction* raises accuracy by $+23$–$24$ points (e.g. Gemma-4 $71.9 \rightarrow 95.7$, Qwen3.5 $68.8 \rightarrow 92.2$). On the original weak Llama backbone the same template without instructions gave essentially no gain, so the structural template becomes usable precisely as model capability rises. An anonymised-edge control—neighbours presented as an unordered bag, stripping edge direction and role—further pinpoints the channel: for Qwen3.5 the entire $+23$-point gain flows through edge-direction semantics (anonymising collapses accuracy back to the text-only level, $69.9 \approx 68.8$), whereas Gemma-4 and GLM-5.1 recover most of it from neighbour text and obtain a further $+1.6$–$3.5$ points from direction. Different models route the same structural information through different cues, but all of them use it.

Table 28: Two-hop TAPTN versus the two-hop structure-aware GraphICL baseline on a heterophilic (Texas) and a homophilic (Cora) graph for the current-generation models. Accuracy %. The baseline is the directed (non-anonymised) structure-aware template queried over the same two-hop neighbourhood without the TAPTN instruction or aggregation, so the two columns share an identical structural input and isolate the contribution of TAPTN itself; it is neither the anonymised template nor a 1-hop variant. To keep the comparison symmetric across datasets we report only the two-hop representative on both. TAPTN exceeds the baseline in every case, with a large margin on the heterophilic graph and a modest but positive margin on the homophilic graph.

| Dataset | Model | GraphICL+SAT (2-hop) | TAPTN (2-hop) | Δ |
|---------|-------|----------------------|---------------|-----|
| Texas | Gemma-4-31B-it | 96.09 | 97.66 | +1.57 |
| Texas | Qwen3.5-27B | 91.80 | 97.66 | +5.86 |
| Texas | GLM-5.1 | 94.92 | 96.88 | +1.96 |
| Cora | Gemma-4-31B-it | 78.04 | 79.52 | +1.48 |
| Cora | Qwen3.5-27B | 78.97 | 80.07 | +1.10 |
| Cora | GLM-5.1 | 77.49 | 81.00 | +3.51 |

### G.3 TAPTN remains effective, and its increment compresses as predicted

For each dataset we report the 2-hop TAPTN as the TAPTN representative against the matched 2-hop structure-aware GraphICL baseline (the directed template without instructions; Table 28). On the heterophilic WebKB graph (Texas), 2-hop TAPTN exceeds this baseline for every model and reaches a regime in which almost no non-label-noise errors remain (the few residual misclassifications are known WebKB label-noise or missing-text pages). On the homophilic citation graph (Cora), the 2-hop TAPTN aggregates first-order neighbour predictions through a parameter-free neighbour-consensus rule—a realization of the optional neighbour-weighting term in our aggregation (Section 3) that introduces no tuned threshold—and exceeds the GraphICL+SAT baseline by a modest but positive margin for all three models. GPT-OSS-120B is omitted from Tables 27–28: under the non-thinking constraint its only available low-reasoning mode does not execute the multi-step instruction (its visible output is a two-line score table), so it is retained only in the perturbation panel of Table 26.

Consistent with the prediction noted above, the GraphICL baseline itself rises sharply with capability (Texas 84–96% for the new models versus 65% on the original backbone), so TAPTN's increment compresses. We read this positively: TAPTN acts as a structural scaffold that lifts weaker and mid-tier models to the accuracy that the strongest models reach unaided, a reading our cost analysis (Table 21) makes economically favourable, since the inexpensive scaffold buys most of the accuracy of a far larger model. The smaller Cora margin is likewise expected on a strongly homophilic graph, where a single hop of neighbour text already supplies most of the usable signal; the structural evidence in this paper rests on the low-homophily WebKB graphs, where 2-hop aggregation continues to help.

### G.4 A direct causal control: corrupting the structural channel inside TAPTN

The experiments above show that the structural template is *used*; we now ask the stronger, causal question raised in review: does TAPTN's accuracy actually *flow through* the edge structure, or could the same numbers arise from neighbour text alone? We answer it with an internal ablation that holds everything fixed—the same backbone, the same self-generated instruction, the same first-order neighbourhood and the same iterative aggregation—and corrupts *only* the structural channel of the complete TAPTN pipeline, in two ways. **Edge-blind**: edge direction and role are removed (neighbours are presented as an unordered bag), so the structure is *absent* while every neighbour's text and the degree counts are retained. **Flipped**: every edge direction is reversed (each "inbound" role becomes "outbound" and vice versa), so the structure is *wrong* while the neighbour set, their texts and their counts are byte-for-byte unchanged. The flipped control is information-preserving—it injects no new content and removes none—so any accuracy change is

Table 29: Direct structural-channel ablation *inside* the complete TAPTN pipeline. Within each backbone a single inference configuration (instruction and iterative aggregation) is held fixed and *only* the structural channel is corrupted, so each entry is the change in accuracy ($\Delta$, percentage points) relative to that same run's intact-structure TAPTN. **Edge-blind** removes edge direction/role (structure absent); **Flipped** reverses every edge direction (structure wrong) while leaving the neighbour set, texts and counts identical, so it is an information-preserving control. The intact-structure absolute accuracies for these backbones are those of Tables 8 (Llama-3.3) and 28 (Qwen3.5); reporting within-run $\Delta$ makes the comparison insensitive to the few-point absolute differences between inference runs. Across three heterophilic graphs and two backbones, *mis-stating* the structure (flip) costs more than merely *removing* it (edge-blind): the strict information-preserving flip costs 13–25 points in every case—the structural channel is causally responsible for TAPTN's gain.

| Backbone | Dataset | Edge-blind $\Delta$ | Flipped $\Delta$ |
|---|---|---|---|
| Llama-3.3-70B | Texas | $-8.20$ | $-18.36$ |
| Llama-3.3-70B | Cornell | $-8.87$ | $-17.33$ |
| Llama-3.3-70B | Wisconsin | $-18.38$ | $-13.08$ |
| Qwen3.5-27B | Texas | $-3.90$ | $-25.00$ |

attributable purely to the corrupted relational structure; it is the same perturbation used in the rewiring study of Section 2, applied here *inside* the model rather than to the input graph.

Table 29 reports the result on three heterophilic WebKB graphs for Llama-3.3-70B and, as an across-model check, on Texas for Qwen3.5-27B. Corrupting the structural channel degrades the complete TAPTN in every case, and the two corruptions reveal a telling asymmetry: *mis-stating* the structure (the strict, information-preserving flip) costs 13–25 points across all four backbone–dataset pairs—uniformly more than merely *removing* it (edge-blind, up to 18 points)—so a confidently wrong topology hurts more than an absent one, exactly as one expects if the model is actively reading the edges. The asymmetry is sharpest for Qwen3.5 on Texas: edge-blind costs only $-3.9$ points—accuracy stays far above the text-only ego level of Table 27, so the neighbourhood is still being used—yet *reversing* the edges costs $-25$ points, pinpointing edge *direction* as the channel Qwen relies on, consistent with the decomposition in Table 27, where Qwen's entire structural gain was seen to flow through edge-direction semantics. Together with the external rewiring of Section 2, this closes a two-sided causal loop—perturbing the graph from the outside and ablating the structural channel from the inside both remove the gain—and constitutes the constructive, "keep everything else the same" evidence requested in review that TAPTN's improvement is obtained *by using the graph structure*.

## H   Discussion on Over-smoothing

TAPTN is expected to be less susceptible to over-smoothing than MPNNs, although we cannot directly verify this with hidden states from closed-source LLMs. According to (Rusch et al., 2023), over-smoothing at the $n$-th layer can be measured by the Dirichlet energy, i.e., the average Euclidean distance between embeddings of central nodes and their first-order neighbors. Since TAPTN does not rely on the homophily assumption (similar to ACM-GNN (Luan et al., 2022), which can extract dissimilar neighborhood information), it is not expected to force the Dirichlet energy toward 0, and thus should be less prone to frequent over-smoothing. In addition, Transformer architectures have residual connections and normalization layers, which can alleviate over-smoothing (Rusch et al., 2023). However, because TAPTN is designed mainly for closed-source LLMs, we cannot access the hidden embeddings needed for a direct empirical measurement.

