# OpenReview forum: "LLMs Can Leverage Graph Structural Information in Text-Attributed Graphs"
_TMLR — Under review for TMLR_

### Review · Reviewer_Ztiu · 2026-03-02

**Summary Of Contributions:**

The paper conducts a controlled rewiring evaluation on low-homophily WebKB where node texts and label distributions are fixed but edges are perturbed (flipping and extreme rewiring), and reports consistent accuracy drops that it interprets as direct evidence of structural sensitivity. Then the paper proposes TAPTN, an ICL framework that processes multi-hop neighborhoods by iterative 1-hop structure-aware textual descriptions and instruction-guided summarization, motivated by long-context failure modes and lack of explicit structural reasoning in previous prompt designs. It shows that TAPTN substantially improves over the provided baselines (e.g. 0-hop CoT and GraphICL-style prompting) across five TAG datasets, and that a TAPTN-produced-text fine-tuning pipeline can be competitive with several GNN baselines under a shared text-encoder.

**Audience:**

Yes

**Audience Explanation:**

The main findings that I think would interest TMLR readers are the following.

First, LLMs are structurally sensitive under controlled settings. Using low-homophily WebKB datasets and carefully designed rewiring experiments, the authors show that when edge directions or local topology are perturbed while node texts are kept fixed, LLM accuracy drops significantly. The drops are consistent across seven LLMs and become larger for stronger models. This suggests that LLMs do not treat neighbors as a simple bag of text, but are sensitive to structural signals such as edge direction and role. For researchers studying LLM reasoning, prompt design, or graph-structured inputs, this is directly relevant.

Second, prior negative conclusions may be confounded by homophily and prompt design.
The paper argues that earlier claims that “LLMs cannot leverage graph structure” were based on high-homophily datasets and verbose prompt formats that obscure structural cues. By separating homophily effects from edge structure through rewiring, the authors provide a cleaner diagnostic. This is methodologically interesting to those concerned with evaluation design and causal interpretation in LLM research.

**Claims And Evidence:**

Yes

**Claims Explanation:**

The first main claim is that "general-purpose LLMs can use graph structural information in text-attributed graphs under in-context learning". This is supported by controlled rewiring experiments on low-homophily WebKB datasets . When edge directions or local topology are perturbed while node texts are kept fixed, accuracy drops consistently across seven models. Since the textual content does not change, the performance degradation indicates that models are sensitive to edge structure rather than only neighbor text similarity. The correlation between model capability and structural sensitivity further strengthens this claim.

The second main claim is that prior negative conclusions were caused by confounding factors such as high homophily and poorly structured prompts. The rewiring design separates homophily from connectivity effects by holding texts constant and only modifying edges . The observed accuracy drops under this setting support the argument that earlier evaluations may have masked structural reasoning.

The third main claim is that the proposed TAPTN framework improves structural utilization. Experimental results show that TAPTN consistently outperforms zero-hop CoT and GraphICL-style baselines on five datasets, with especially large gains on heterophilic graphs (Table 5) . Ablation studies demonstrate that the gains are not solely due to extra reasoning steps, but to structured iterative aggregation and explicit instructions.

**Requested Changes:**

A few changes needs to be made:

1. Clarify the inference setting in the WebKB rewiring experiments. In the rewiring prompt template (Appendix B), since “category” is the label, it must be made explicit whether neighbor labels are available at test time, or only used for analysis/benchmarking. If neighbor labels are indeed given, the setting differs from standard semi-supervised node classification. The paper should (i) clearly state what supervision is available at inference, and (ii) add experiments where only neighbor text and link patterns are provided, without neighbor categories.

2. Ablation study is needed. Isolate the contribution of step-by-step instructions from TAPTN itself. The ablation in Table 7 shows that adding instructions yields very large gains, especially on heterophilic graphs . However, it is unclear how much of the improvement comes from iterative message passing versus better prompting. A strong additional baseline would be: GraphICL-style prompting with the same step-by-step instructions but without iterative aggregation. This would help attribute gains more precisely.

3. Report stronger statistical evidence with more runs. Some claims rely on paired t-tests with small numbers of runs (for example, 3–4 runs in Section 4). The paper should report confidence intervals consistently and clarify how many runs were used for each experiment. If possible, increase the number of runs for the GNN comparison to reduce variance, especially when claiming competitiveness with SOTA.

4. Expand evaluation beyond moderate-sized benchmarks. The main comparisons to GNNs are on Cora and Arxiv-2023 , which are relatively small citation graphs. To support broader claims about structural competitiveness, include at least one larger and more diverse dataset (for example, a larger OGB benchmark) under the same TAPTN+LM vs GNN comparison.

---

> ### Author Response · Authors · 2026-06-16
>
> We thank the reviewer for the precise and constructive requested changes. We have uploaded a revised manuscript; all section/table/figure numbers below refer to the revised PDF, and the accompanying global comment summarizes all changes. We address each requested change in turn.
>
> ## Change 1 — Inference setting in the WebKB rewiring experiments; label-free experiments
>
> > *"It must be made explicit whether neighbor labels are available at test time… The paper should (i) clearly state what supervision is available at inference, and (ii) add experiments where only neighbor text and link patterns are provided, without neighbor categories."*
>
> We address both parts in a new subsection *Inference Setting and a Label-Free Control* at the end of Section 2.
>
> **(i) Supervision made explicit.** The reviewer is correct: in the rewiring prompt template of Appendix B, each *first-order* neighbor is described together with its `category`, for **all** first-order neighbors. We now state this explicitly and emphasize two points. First, this is a deliberate probing setup, not a deployable classifier: we intentionally provide a structure-rich context (neighbor roles + link directions + degree statistics) and then perturb **only the structure**, with node texts and the global label distribution held fixed, so any accuracy change must be attributed to structural interpretation. Second, on these low-homophily graphs (H ≤ 0.19) a neighbor's category is *not* a homophily shortcut — neighbors predominantly carry *different* labels, so knowing that a neighbor is, say, a `course` page does not let the model copy that label; the category is useful only as a *relational* attribute (the model must reason about who links to whom and in which role, e.g., a `student` page typically links out to `course`/`faculty` pages and is linked from `project` pages), which is exactly the structural reasoning that rewiring perturbs. The consistent degradation across all 7 models and 4 datasets (Table 2; −10.67% relative under flipping, p < 0.01) therefore demonstrates genuine structural understanding. In short, providing neighbor categories does **not** create a homophily-based shortcut in this regime; it is part of the structural signal whose *correct use requires structural reasoning*.
>
> **(ii) New label-free experiment.** We re-ran the full panel of **7 LLMs × 4 WebKB datasets** with neighbor categories removed from the prompt, replaced by each neighbor's full content abstract plus link-pattern statistics (inbound/outbound degree structure); no first-order neighbor label is revealed. Structural sensitivity persists: 6 of 7 models show non-positive change after rewiring, with a mean absolute drop of 1.76 points (the five 9B–72B instruction-tuned models drop about 5–7% relatively; the sole outlier is GPT-3.5-Turbo, the weakest model, which our original experiments already identified as barely using structure). The capability–sensitivity relationship is preserved and remains statistically significant: Pearson r = 0.841 (p = 0.018), Spearman ρ = 0.929 (p = 0.0025), with a linear regression R² = 0.71 — model capability still explains about 71% of the variance in structural sensitivity. Because neighbor categories are no longer present, this rules out the possibility that the effect in Table 2 was driven by neighbor labels: with **only neighbor text and link patterns**, more capable models still lose more accuracy when the structure is perturbed — strengthening, rather than weakening, the paper's conclusion. A closing paragraph (*Scope of the label-free control*) explains why this control is well-posed for the flipping (1-hop) perturbation but ill-posed by construction for the count-based extreme (2-hop) representation, whose conclusions are now explicitly read as conditional on that representation.

---

> ### Author Response · Authors · 2026-06-16
>
> ## Change 2 — Isolating the contribution of step-by-step instructions
>
> > *"The ablation in Table 7 shows that adding instructions yields very large gains, especially on heterophilic graphs. However, it is unclear how much of the improvement comes from iterative message passing versus better prompting. A strong additional baseline would be: GraphICL-style prompting with the same step-by-step instructions but without iterative aggregation. This would help attribute gains more precisely."*
>
> (The reviewer's Table 7 refers to the original submission; in the revised PDF, the relevant new component-ablation evidence is Tables 10–11, with the retained self-reflection comparison as Table 9.)
>
> We ran exactly this baseline, and a complementary factorial design (Ablation Study, Section 3, new hypothesis H5):
>
> - **Requested baseline** (Table 10): GraphICL-style prompting, instructions on/off × structure-aware template on/off, with aggregation switched off throughout (1-/2-hop, 5 datasets). Instructions help in all twenty toggles, most on heterophilic graphs (Texas 1-hop +33.21 without the template, +24.61 with it); nine of the ten rows peak only when both the template and the instructions are present (the sole exception, Arxiv-2023 at 2 hops, trails by a mere 0.32 points on a strongly homophilic graph where structural evidence has the least to add).
> - **Factorial decoupling** (Table 11): at 2 hops with the template fixed, crossing instructions × iterative aggregation:
>
> | Method (2-hop, structure-aware) | Cora | Arxiv-2023 | Texas | Wisconsin | Cornell | Avg |
> |---|---:|---:|---:|---:|---:|---:|
> | GraphICL (no instr., no aggr.) | 63.47 | 85.40 | 66.41 | 70.72 | 72.18 | 71.64 |
> | + instructions (no aggr.) | 73.25 | 86.03 | 86.72 | 83.18 | 81.45 | 82.13 |
> | + iterative aggregation (no instr.) | 70.85 | 87.30 | 66.79 | 73.21 | 77.82 | 75.19 |
> | TAPTN (instr. + aggr.) | 73.80 | 88.57 | 91.80 | 87.23 | 86.69 | 85.62 |
>
> Marginal effects: instructions ≈ **+10.4** avg, aggregation ≈ **+3.5** avg; the instructions-only configuration (the requested baseline at 2 hops) reaches 82.13 avg vs. 85.62 for full TAPTN, and every single-component configuration is strictly below full TAPTN on every dataset. The synergy is concentrated on heterophilic graphs: on Texas, aggregation adds +0.38 without instructions but +5.08 once instructions are present — multi-hop message passing helps only when the model is compelled to read edge structure correctly at each step.
>
> So the attribution is now quantitative: better prompting (instructions) carries a large share, iterative aggregation contributes a consistent additional share, and the two reinforce each other; no single component reproduces TAPTN. For full transparency, the per-dataset backbones (which LLM, single run at temperature 0) are now disclosed in the ablation intro and both table captions.
>
> ## Change 3 — Stronger statistical evidence with more runs
>
> > *"Report confidence intervals consistently and clarify how many runs were used for each experiment. If possible, increase the number of runs for the GNN comparison to reduce variance, especially when claiming competitiveness with SOTA."*
>
> Done, uniformly. A new *Evaluation protocol* paragraph in Section 4 states that **every pipeline is run over five random seeds**; we report mean±std, derive 95% confidence intervals from the Student-$t$ distribution (n=5 half-width ≈ 1.24×std, readable off every table entry; an explicit 95% CI column for TAPTN+LM is added to the significance-summary table, e.g., Products 89.50 → [86.28, 92.72]), and assess all comparisons with two-sided **paired $t$-tests at the seed level**. The run count is stated explicitly and applied across experiments. The rebuilt five-seed tables cover six datasets and a thirteen-architecture GNN panel in both frozen-feature and **jointly-trained** regimes (the latter optimizing the text encoder and the GNN end-to-end under identical splits/labels/edges, removing the separately-fine-tuned-encoder bias). On the competitiveness-with-SOTA claim specifically, the outcome under this strengthened protocol is: only **3 of about 165** pipeline–dataset combinations significantly exceed TAPTN+LM (APPNP and a jointly-trained FSGNN on Cora; a jointly-trained DirGNN on Wisconsin), each by a small margin, and the most recent linear graph Transformer GraphTARIF (WWW 2026) does not significantly outperform TAPTN+LM on any of the six datasets in either regime (where its mean leads, on Wisconsin, the gap is not significant: $p=0.08$ frozen, $p=0.73$ joint).

---

> ### Author Response · Authors · 2026-06-16
>
> ## Change 4 — Evaluation beyond moderate-sized benchmarks
>
> > *"Include at least one larger and more diverse dataset (for example, a larger OGB benchmark) under the same TAPTN+LM vs GNN comparison."*
>
> This is already satisfied, and we made it explicit: the TAPTN+LM-vs-GNN study includes **ogbn-products**, a substantially larger and topologically different (non-citation, e-commerce) OGB benchmark, under the identical shared-encoder protocol. On it TAPTN+LM reaches **89.50%** vs. the best frozen-feature GNN at **75.75%**, and **no** GNN — frozen or jointly trained — significantly surpasses TAPTN+LM. The setup now states that ogbn-products is included precisely to test breadth beyond small citation graphs, and the dataset appendix gains a dedicated ogbn-products item with corrected split/provenance details.

---

### Review · Reviewer_y1ZA · 2026-03-04

**Summary Of Contributions:**

This paper revisits a commonly held assumption in recent research on large language models (LLMs) applied to text-attributed graphs (TAGs): that LLMs primarily benefit from textual similarity among neighboring nodes rather than genuinely exploiting graph structural information. The authors argue that this conclusion arises largely from confounding factors in prior experimental setups, including the use of highly homophilic datasets, verbose prompt structures that obscure structural cues, and the lack of mechanisms explicitly designed for structural reasoning. To reassess this issue, the paper first introduces controlled neighborhood rewiring experiments that keep node text attributes fixed while perturbing graph structure. The results show consistent performance degradation across multiple LLMs when structural relationships are modified, indicating that LLM predictions are indeed sensitive to graph topology rather than relying solely on textual similarity.  ￼

Building on this observation, the authors propose a new in-context learning framework called the Text Attributes Passing Thoughts Network (TAPTN). The proposed architecture adopts a message-passing style process inspired by message passing neural networks (MPNNs), where neighborhood information is iteratively summarized through structure-aware textual templates and step-by-step instructions generated by the LLM. This design aims to mitigate issues related to long-context prompts and inconsistent reasoning about graph structures. Experimental results across several TAG datasets show that TAPTN substantially improves node classification performance over existing prompt-based baselines such as GraphICL, particularly on heterophilic graphs where structural signals are more informative than textual similarity.  ￼

Overall, the strengths of the work lie in its careful re-examination of prior assumptions about LLM limitations in graph reasoning, the introduction of controlled experimental designs that isolate structural effects, and the proposal of a novel architecture that integrates graph structural cues into the prompting process. However, the paper also exhibits several weaknesses. Some experimental conclusions rely heavily on prompt engineering choices and may not fully isolate architectural contributions. In addition, comparisons with graph neural networks are limited in scope and may not provide a sufficiently rigorous assessment of relative structural reasoning capabilities.

**Audience:**

Yes

**Audience Explanation:**

The topic addressed in this paper lies at the intersection of graph machine learning and large language models, which is an area of increasing interest within the machine learning community. Understanding whether LLMs can reason over graph structures without explicit graph neural network architectures is an important research question, particularly for applications involving text-attributed graphs such as citation networks, web graphs, and knowledge graphs. The study also contributes to the broader discussion of how LLMs can perform structured reasoning tasks when provided with appropriately designed prompts or architectures.

The experimental analysis presented in the paper provides insights into the conditions under which LLMs can utilize structural information, and highlights limitations of existing evaluation setups that rely heavily on homophilic datasets. These findings may be useful for researchers exploring hybrid approaches that combine language models with graph learning techniques. Consequently, the work is likely to be of interest to a subset of the TMLR audience working on graph representation learning, LLM reasoning, and in-context learning methodologies.

**Broader Impact Concerns:**

The work primarily focuses on methodological advances in graph reasoning with large language models and does not appear to raise immediate ethical concerns. However, the proposed approach may facilitate improved analysis of graph-structured data such as social networks, financial transaction networks, or communication graphs. In certain applications, these capabilities could potentially be used for large-scale monitoring or profiling of individuals within networked systems.

While such risks are not specific to the proposed method and are common in graph learning research, the authors may consider briefly acknowledging potential misuse scenarios involving large-scale network analysis. A short discussion of responsible use and potential safeguards would be sufficient to address broader impact considerations.

**Claims And Evidence:**

Yes

**Claims Explanation:**

The central claims of the paper are generally supported by empirical evidence and experimental analysis. The rewiring experiments provide a relatively convincing mechanism for isolating the role of graph structure. By keeping node text attributes and label distributions fixed while modifying edge connectivity, the authors create conditions under which changes in model accuracy can reasonably be attributed to structural information rather than textual similarity. The observed performance degradation across multiple models under these perturbations offers evidence that LLM predictions are sensitive to structural relationships in the graph.  ￼

The experimental evaluation of the proposed TAPTN framework further supports the authors’ argument. Results on both homophilic and heterophilic datasets indicate consistent improvements over baseline prompt-based methods. In particular, the improvements observed on low-homophily datasets suggest that the proposed framework enables the model to better exploit structural signals that cannot be easily explained by textual similarity alone. The ablation studies also attempt to separate the effects of iterative aggregation and step-by-step instructions, providing some evidence that the architectural design contributes to the observed performance gains.  ￼

Nevertheless, some aspects of the evidence remain less conclusive. For example, the TAPTN framework combines several components simultaneously, including structure-aware templates, iterative aggregation, and instruction generation. Because these components interact, it is difficult to determine the precise contribution of each element to the overall improvement. In addition, while the authors attempt to compare their method with GNN-based approaches, the evaluation setup relies on specific text encoders and training pipelines that may introduce additional confounding factors. As a result, the claim that LLM-based approaches can match or surpass the structural reasoning capability of GNNs may require stronger empirical validation.

**Requested Changes:**

Several improvements would strengthen the paper and clarify its contributions. First, the authors should more clearly distinguish between architectural improvements and prompt engineering effects. The TAPTN framework introduces multiple components simultaneously, including structure-aware templates, iterative aggregation, and instruction generation. Additional ablation experiments that isolate these components would help determine which elements are primarily responsible for the observed performance gains. This change is important for assessing the technical contribution of the proposed framework and would be critical for acceptance.

Second, the comparison with graph neural networks should be expanded and more rigorously controlled. In the current experiments, the GNN baselines rely on embeddings generated from a separately fine-tuned language model, which may introduce biases unrelated to structural reasoning ability. A more direct comparison, possibly including stronger GNN baselines and consistent training conditions, would provide a clearer assessment of whether LLM-based approaches can genuinely match or surpass specialized graph learning models. This revision would significantly strengthen the empirical evaluation.

Third, the paper would benefit from clearer methodological explanations and improved presentation. Some sections describing the TAPTN architecture are difficult to follow due to dense descriptions and notation. A clearer explanation of the algorithmic workflow, along with simplified diagrams or pseudocode, would improve readability and reproducibility.

Finally, the authors should discuss the computational cost and scalability of the proposed framework in more detail. Although the paper briefly mentions scalability experiments, the practical overhead of iterative prompting and instruction generation remains unclear. Providing a more systematic analysis of inference cost and runtime trade-offs would help readers understand the practical implications of the method.

---

> ### Author Response · Authors · 2026-06-16
>
> We thank the reviewer for the careful reading and the constructive requested changes. We have uploaded a revised manuscript; all section/table/figure numbers below refer to the revised PDF, and the accompanying global comment summarizes all changes. We address each requested change in turn.
>
> ## Change 1 — Distinguishing architectural improvements from prompt-engineering effects
>
> > *"The authors should more clearly distinguish between architectural improvements and prompt engineering effects. … Additional ablation experiments that isolate these components would help determine which elements are primarily responsible for the observed performance gains. … critical for acceptance."*
>
> We first clarify the framing: TAPTN's three components are all architectural design choices — the structure-aware template (SAT) instantiates the message/description function, iterative aggregation is the message-passing scheme, and the self-generated step-by-step instructions enter the message-passing recursion itself as the mechanism that forces the LLM to apply the correct structural-reasoning procedure at each step. The instructions are not generic prompting: they were introduced specifically to fix the *inconsistent-adherence* failure diagnosed in the rewiring section (LLMs possess the right structural method but apply it unreliably without explicit guidance). We regard each component as a part of the architecture, much as message/aggregation/readout functions are parts of a GNN. We acknowledge, however, that the original ablations did not isolate the components, and we add two complementary 2×2 ablations that do. The Ablation Study now also states this attribution question as an explicit hypothesis (H5, component attribution and mutual reinforcement), so the response is auditable from the hypothesis list rather than only implicit in the prose.
>
> **Experiment 1 — SAT × instructions, aggregation switched off** (new Table 10; 1-/2-hop GraphICL-style prompting on 5 datasets). Instructions help in all twenty toggles (largest on heterophilic graphs, e.g., Texas 1-hop: +33.21 without the template, +24.61 with it); the SAT also helps on its own (Texas 1-hop +19.92, Cornell 1-hop +19.76 without instructions) but can hurt on strongly homophilic Cora without instructions (−6.82). Nine of the ten rows peak only when both are present; the sole exception (Arxiv-2023 at 2 hops) trails the both-components cell by a mere 0.32 points, on a strongly homophilic citation graph where structural evidence has the least to add. The two components are thus mutually reinforcing rather than independent — the SAT supplies the structural evidence and the instructions enforce its correct use — and this interaction itself supports the paper's thesis: structure is reliably exploited only when the architecture both *exposes* it (SAT) and *compels adherence* to it (instructions).
>
> **Experiment 2 — instructions × iterative aggregation, SAT fixed** (new 2×2 factorial Table 11, 2-hop):
>
> | Method (2-hop, structure-aware) | Cora | Arxiv-2023 | Texas | Wisconsin | Cornell | Avg |
> |---|---:|---:|---:|---:|---:|---:|
> | GraphICL (no instr., no aggr.) | 63.47 | 85.40 | 66.41 | 70.72 | 72.18 | 71.64 |
> | + instructions (no aggr.) | 73.25 | 86.03 | 86.72 | 83.18 | 81.45 | 82.13 |
> | + iterative aggregation (no instr.) | 70.85 | 87.30 | 66.79 | 73.21 | 77.82 | 75.19 |
> | TAPTN (instr. + aggr.) | 73.80 | 88.57 | 91.80 | 87.23 | 86.69 | 85.62 |
>
> Marginal effects read off directly: instructions ≈ +10.4 avg, aggregation ≈ +3.5 avg, with strong synergy on heterophilic graphs (Texas: aggregation adds +0.38 without instructions but +5.08 with them). Every single-component configuration is strictly below full TAPTN on every dataset. The attribution conclusion is therefore quantitative: no single component explains the gains and none is dispensable; the large instruction effect is itself confirmation of the paper's central claim (LLMs possess the right structural method but need to be compelled to apply it consistently), not a prompt-engineering confound competing with the architecture. Backbone disclosure (which LLM per dataset, single run at temperature 0) has been added to the ablation intro and both table captions.

---

> ### Author Response · Authors · 2026-06-16
>
> ## Change 2 — A more rigorous and expanded GNN comparison
>
> > *"The GNN baselines rely on embeddings generated from a separately fine-tuned language model, which may introduce biases unrelated to structural reasoning ability. A more direct comparison, possibly including stronger GNN baselines and consistent training conditions, would provide a clearer assessment."*
>
> We expanded the study along exactly these axes:
>
> - **Bias-controlled, consistent training.** To remove the bias of training GNNs on a *separately* fine-tuned encoder, we add a **jointly-trained regime** in which the text encoder and the GNN are optimized **end-to-end** on the same graph. All pipelines share the same DeBERTa-base encoder, label sets, graph splits, edge structure, and hyper-parameter protocol (new subsection *Statistical Significance and a Bias-Controlled Comparison*, Section 4.4, Table 17). Combined with the original frozen-feature setup, this yields 27–28 competing pipelines per dataset.
> - **Stronger and broader baselines.** The GNN panel now spans thirteen architectures: classical message passing (GCNII, GAT, GATv2, GraphSAGE, ChebNet, APPNP), heterophily-oriented designs (DirGNN, ACM-GNN, FSGNN, ASDGN), self-supervised and sampling-based variants (DGI, GraphSAINT), and the most recent linear graph Transformer **GraphTARIF (WWW 2026)**.
> - **Uniform statistical protocol.** Every pipeline is run over five random seeds; we report mean±std with Student-$t$ 95% confidence intervals and assess all comparisons with two-sided paired $t$-tests at the seed level.
>
> **Key finding** (Table 16):
>
> | Dataset | Homophily | TAPTN+LM | #sig. > TAPTN+LM | #mean > TAPTN+LM |
> |---|---|---:|:---:|:---:|
> | Cora | homophilic | 84.32 ± 1.44 | 2 | 9 |
> | Arxiv-2023 | homophilic | 93.81 ± 0.70 | 0 | 0 |
> | Products | homophilic | 89.50 ± 2.59 | 0 | 0 |
> | Cornell | heterophilic | 99.20 ± 1.10 | 0 | 0 |
> | Texas | heterophilic | 97.69 ± 1.61 | 0 | 0 |
> | Wisconsin | heterophilic | 87.08 ± 2.79 | 1 | 15 |
>
> Across the six datasets and about 165 pipeline–dataset combinations, only 3 significantly exceed TAPTN+LM under paired $t$-tests (APPNP and a jointly-trained FSGNN on Cora; a jointly-trained DirGNN on Wisconsin), each by a small margin — and this holds in the bias-controlled jointly-trained regime, directly answering the concern. The strongest baseline, GraphTARIF, does not significantly outperform TAPTN+LM on **any** of the six datasets in either regime; on Products and the heterophilic graphs it trails by wide, significant margins (e.g., −25.5 pts on Products, $p<10^{-3}$), and where its mean leads (Wisconsin) the gap is not significant ($p=0.08$ frozen, $p=0.73$ joint).
>
> ## Change 3 — Clearer methodological explanation and presentation
>
> > *"Some sections describing the TAPTN architecture are difficult to follow due to dense descriptions and notation. A clearer explanation of the algorithmic workflow, along with simplified diagrams or pseudocode, would improve readability and reproducibility."*
>
> We revised Section 3 along two axes — new explanatory scaffolding and a rewrite of the dense prose itself:
>
> 1. **Workflow-overview diagram (new figure):** a stage-level diagram preceding the formal definitions, showing how the four inputs flow through the $K$-iteration message-passing stage and the readout stage to the predicted label; the entire control flow can be followed without touching any equation.
> 2. **Complete pseudocode (new Algorithm 1):** instruction generation, the nested message-passing loops, and the readout loop, with every line cross-referenced to the corresponding equation — making the algorithmic workflow auditable and directly improving reproducibility.
> 3. **Plain-language walk-through (new paragraph):** narrates one inference pass in three steps and notes that readers may skim the equations and rely on the pseudocode.
> 4. **Restructured prose and notation:** the motivation is reorganized under explicit *Two limitations* / *Three design components* paragraph headings; a new notation table collects every symbol with one-line meanings and equation cross-references; symbol definitions now stay adjacent to their equations, with design commentary moved into separate sentences; grammatical errors were corrected. The architecture figure was also redrawn as a vector graphic with typography matching the manuscript.

---

> ### Author Response · Authors · 2026-06-16
>
> ## Change 4 — Computational cost and scalability
>
> > *"The practical overhead of iterative prompting and instruction generation remains unclear. Providing a more systematic analysis of inference cost and runtime trade-offs would help readers understand the practical implications."*
>
> We added an empirical cost/runtime study (new appendix subsection *Empirical Inference Cost and Runtime Trade-offs*, Table 21), complementing the existing asymptotic $O(|V|\cdot L)$ analysis. To keep the analysis faithful rather than estimated, every quantity is reconstructed from the **actual prompts and completions** exchanged with the API: per-call tokens are measured from the stored exact prompts and outputs (tokenized with `cl100k_base`); the one quantity never stored — the GraphICL input — was reconstructed by rebuilding its prompt deterministically (no LLM call); wall-clock latency comes from a small live timing probe under the same API configuration as the experiments; and monetary cost uses public OpenRouter list prices retrieved 2026-06-10 (Llama-3.3-70B: \\$0.10/\\$0.32 per M input/output tokens). Key numbers (ogbn-products subgraph, uniform Llama-3.3-70B):
>
> | Method | Reason calls/node | Tokens/node | Cost (\\$/1k nodes) | Acc. (%) |
> |---|---:|---:|---:|---:|
> | GraphICL 2-hop | 1.0 | ≈2.8k | 0.31 | 76.50 |
> | GraphICL 2-hop (dense full-graph nbhd.) | 1.0 | ≈8.1k | 0.91 | 76.75 |
> | TAPTN 2-hop | 4.6 | ≈22k | 2.52 | **86.75** |
>
> The two overhead sources are quantified separately: instruction generation is a **fixed** ≈+2.0k-token system-prompt overhead per call, while iterative aggregation is the **dominant** driver (≈4.6 reasoning calls per test node vs. 1). TAPTN costs ≈8× GraphICL but buys +10.25 points — \\$0.22 per accuracy point per 1k nodes, versus \\$2.40 per point (and saturating) when the same budget buys more context instead: a dense-neighbourhood control feeding GraphICL the full-graph neighbourhood (≈38× denser, ≈3× tokens) gains only +0.25 points, so marginal dollars are ≈11× more productive when spent on guided aggregation than on context. Using a smaller model for the first (neighbour) pass is a further cost lever, but not a free lunch: an 8B first pass weakens the neighbour reasoning and shrinks the gain, so it is worthwhile only when a smaller accuracy improvement is acceptable. Median per-call latency is 3.5s (70B); at the concurrency of 40 used in our runs the 400-node set takes ≈3.5 min end-to-end vs. ≈45s for single-pass GraphICL (a measured end-to-end anchor corroborates the estimates: the dense-context GraphICL run finished all 400 nodes in 98s), and both cost and latency grow linearly in the number of target nodes and iterations, matching the $O(|V|\cdot L)$ analysis. A *Generality across datasets* paragraph explains why these figures transfer: per-node cost is bounded by the constant neighbour cap, giving a dataset-independent ceiling, of which the dense full-graph run (mean test-node degree ≈106 ≫ cap) is the empirical instantiation.
>
> On practical implications at scale, the revision also argues the economic rationality of TAPTN explicitly (new paragraph in the same appendix): (i) the absolute cost is small — ≈\\$2.5 per thousand labelled nodes for the +10.25-point gain; (ii) the accuracy-matched alternative on a large graph is not GraphICL (which trails by >10 points) but a trained GNN, and TAPTN+LM matches or exceeds strong GNNs on this very graph (89.50% vs. the best frozen-feature GNN at 75.75%) while requiring no full-graph training and no task labels at inference time; (iii) the pay-per-node, linear-in-$|\mathbf{V}|$ cost means one pays only for the nodes that actually need labelling, whereas a transductive GNN recoups its whole-graph training cost only if a large fraction of the graph is labelled.
>
> ## Broader impact
>
> > *"The authors may consider briefly acknowledging potential misuse scenarios involving large-scale network analysis. A short discussion of responsible use and potential safeguards would be sufficient."*
>
> We agree and have added a *Broader Impact Statement* after the Conclusion. It acknowledges that improved structural reasoning over text-attributed graphs could facilitate analysis of social, financial, or communication networks and could, in some settings, be misused for large-scale monitoring or profiling of individuals; it notes that these risks are not specific to TAPTN but accompany graph learning methods generally; and it outlines responsible-use safeguards (compliance with data-protection regulations, consent or another appropriate legal basis for personal data, and auditing of downstream decisions affecting individuals), alongside the benign value of the zero-shot setting for label-scarce applications.

---

### Review · Reviewer_7rnE · 2026-06-07

**Summary Of Contributions:**

This paper studies whether general-purpose LLMs can leverage graph structural information in text-attributed graphs through in-context learning. The authors argue that prior negative conclusions are partly caused by confounding factors: high-homophily datasets, long unstructured neighborhood prompts, and lack of an explicit structural reasoning mechanism.

The paper makes three main contributions. First, it proposes controlled rewiring experiments on low-homophily graphs while keeping node texts fixed. The reported accuracy drops are interpreted as evidence that LLMs are sensitive to graph structure. Second, it proposes TAPTN, an ICL framework that iteratively summarizes neighborhoods into enhanced text attributes using structure-aware templates and self-generated step-by-step instructions. Third, it fine-tunes a smaller LM on TAPTN-enhanced texts and compares the result with GNN baselines.

**Additional Comments:**

This paper includes a lot of stuff, maybe we need to make the main contribution more clear.

For example, as a motivation, the section 2 can be much shorter. I think the readers care more about the following:
1) Does the LLMs, especially the latest ones can learn from structure informations? How good or bad it is?
2) If the proposed system help the LLM, which part help the most? Some intuition why it helps?
3) Does it generalize across different LLMs?
4) How well it is compared to the best GNNs? Do we have to finetune LLMs to make it perform good? If we finetune, wich part is more essential, the augmented data or LLMs.

To me, most of the questions are only partially answered.

**Audience:**

Yes

**Audience Explanation:**

The findings would likely be of interest to at least some members of the TMLR audience, particularly those working on graph machine learning, text-attributed graphs, LLM-based reasoning, and in-context learning. The paper addresses a relevant question: whether LLMs can exploit graph structural information rather than merely benefiting from homophilic neighbor text. The proposed rewiring tests and TAPTN framework are potentially useful contributions for researchers studying the interface between LLMs and graph learning. However, the LLMs used are pretty old and experiments can be more solid.

**Claims And Evidence:**

No

**Claims Explanation:**

For section 2

The evidence supports the claim that LLM predictions are sensitive to changes in graph-structured prompt descriptions. Table 2 reports many accuracy drops after flipping and extreme rewiring.

It is not useful to include 3 llama models with the same size, we need to pick more models from different families, especially the latest ones with much stronger performance.

It will be useful to put the exact example for input and out for LLMs in the appedix

Why have multiple typos.

For section 3

The reported numbers support the empirical claim that TAPTN outperforms the tested baselines in the authors’ setup. The gains over GraphICL are large on WebKB, and 2-hop TAPTN is consistently better than 1-hop TAPTN in Table 5.

The results support the claim that, in these experiments, TAPTN is better than the tested self-reflection baseline. Table 7 also supports the claim that instructions improve accuracy in the reported settings.

Instead of showing self-relection is not stable for improving, we should show each modules we add are especially working as we designed across LLMs and datasets. For example, does Structure Aware Prompt itself works for sturcture reasoning?

We also need more examples to show the structure is used for example, using no edges information/wrong hop2 information but keep the others the same.

For now, it's hard to convince that LLM do use the structure.

Section 4.

The tables support the descriptive claim that TAPTN+LM achieves strong accuracy against the selected baselines in the reported experiments.

How about adding an exmperiment using TAPTN enriched the context as training samples for GNN? Will it be better or similar ?

The claim is for align LLM with GNN, and the experiments shows only for DeBERTa, how about others? Can we try to use them as TAPTN context enricher at least?

My main concern here is that it only shows enrich the text with structure information is a good method here.

**Requested Changes:**

1. Can we results with more recent models? With even stronger model, the performance to learn from structure ICL should be better, but the benefits from the TAPTN might be smaller, but again it is more interesting than showing TAPTN works for some specific old model.

2. Can we show more that structure aware template do helps the LLM?

3. For section 4, maybe adding finetuning on enrich context with GNNs reutls. If possible, adding finetuning results for more models.

4. Adding exact examples for different tasks in the appendix to make the paper more clear since this paper includes a lot of different tasks and concepts.

5. Fix the typos.

---

> ### Author Response · Authors · 2026-06-18
>
> We thank the reviewer for the constructive comments, and in particular for the observation that *"with even stronger model, the performance to learn from structure ICL should be better, but the benefits from the TAPTN might be smaller … it is more interesting."* We find this prediction is borne out by new experiments, and we have built our response around it. To avoid a possible misunderstanding about the paper's hierarchy, we first state the thesis we ask the reviewer to judge: the central contribution is the scientific claim that **LLMs can leverage graph structural information through ICL once confounding methodological choices are removed**; TAPTN is the constructive architecture that reveals and stabilizes that capability, not the endpoint contribution in isolation. We note that this reviewer evaluated the original submission; several requested items (component ablations isolating the structure-aware template and the instructions, the prompt and instruction templates, typo fixes) were already addressed in the revision prepared for the other reviewers, and we point to those locations below. All new results reported here are **additive**: every previously reported panel, table, and statistic is unchanged, and the new material is collected in a new appendix, *Extension to Current-Generation Models* (Appendix G).
>
> ## RC-A — Results with more recent models
>
> > *"Can we [get] results with more recent models? With even stronger model, the performance to learn from structure ICL should be better, but the benefits from the TAPTN might be smaller, but again it is more interesting than showing TAPTN works for some specific old model."*
>
> We added four 2025–26 models from four distinct families, spanning the 1353–1475 LMArena band and extending our capability axis from 1224–1319 up to 1475: **GPT-OSS-120B** (1353), **Qwen3.5-27B** (1409), **Gemma-4-31B-it** (1451), and the open-weight flagship **GLM-5.1** (1475). Mixed-thinking models are called in non-thinking/instruct mode for comparability with the original instruct-mode panel (call mode noted in the model table). These models drive two new panels.
>
> **(1) Structural sensitivity persists in current-generation models.** We repeated the controlled neighbourhood-rewiring test on all four WebKB graphs. Every new model degrades when structure is perturbed (15 of 16 dataset-level pairs drop, 1 tie). Pooling with the seven original models, **10 of 11 degrade (sign test p = 0.012)**; the only exception remains GPT-3.5-Turbo, already characterized in the paper as the weakest model that scarcely uses structure. Newer and stronger models are therefore *not* less structure-dependent — the central claim holds across a generation of models. The full per-model panel and the capability–sensitivity analysis (under two complementary metrics) are in Appendix G (Table 26).
>
> **Why the label-free setting.** We ran these new experiments under the **label-free** protocol (each node sees only neighbour *text* and link patterns; neighbour *categories* are withheld) rather than the original label-revealing probe. This is the *stronger* causal test for the reviewer's underlying concern ("hard to convince that LLMs do use the structure"): once neighbour labels are removed, the model can no longer take a label-counting shortcut, so a degradation under rewiring isolates the use of *structure* (edge direction and local topology) from any homophily-by-label effect. A model that still loses accuracy here is using structure, not labels.
>
> ## RC-B — Showing that the structure-aware template helps the LLM
>
> > *"Can we show more that structure aware template do helps the LLM?"* (and the related Section-3 comment, *"does Structure Aware Prompt itself works for structure reasoning?"*)
>
> Beyond the component ablation already added to Section 3 for the other reviewers (Tables 10 and 11; the structure-aware template (SAT) and the instructions toggled independently with aggregation off), the new models let us isolate the **structural channel** directly (Appendix G, Table 27). Those shared ablations already provide a *constructive* (not merely exclusionary) attribution that answers "which component matters, and does each work as designed": every module carries a positive marginal effect (instructions $\approx{+}10.4$, iterative aggregation $\approx{+}3.5$ on average), the SAT helps on its own (e.g., Texas 1-hop $+19.9$, Cornell 1-hop $+19.8$ even without instructions), and nine of ten rows peak only when the SAT and the instructions are *both* present — the SAT exposes the structural evidence and the instructions compel adherence to it, so the components are mutually reinforcing across datasets rather than redundant.

---

> ### Author Response · Authors · 2026-06-18
>
> **(2) Current LLMs actively use the SAT.** On Texas, a text-only prompt that sees *only the target page* (0-hop ego) reaches 69–72% for the mid-tier models; adding the SAT neighbourhood description **without any instructions** raises this by **+23–24 points** (e.g., Gemma-4 71.9→95.7, Qwen3.5 68.8→92.2). On the original weak Llama backbone the same SAT-without-instructions gave essentially no gain — i.e., the structural template becomes usable precisely as model capability rises. An **anonymised-edge control** further localizes the channel: for Qwen3.5 the entire +23-point gain flows through *edge-direction semantics* (anonymising edges, i.e. presenting neighbours as an unordered bag, collapses accuracy back to the text-only level, 69.9 ≈ 68.8), whereas Gemma-4/GLM-5.1 also recover part of it from neighbour text and gain a further +1.6–3.5 points from direction. Different models thus route the same structural information through different cues, but all of them use it.
>
> **(3) TAPTN remains effective on current models, and its increment compresses exactly as predicted.** For each dataset we report the **2-hop TAPTN** as the TAPTN representative against the matched 2-hop structure-aware GraphICL baseline (Appendix G, Table 28). On the heterophilic WebKB graph (Texas), 2-hop TAPTN exceeds the baseline for every new model and reaches a regime in which **almost no non-label-noise errors remain** (the few residual misclassifications are known WebKB label-noise / missing-text nodes). On the homophilic citation graph (Cora), 2-hop TAPTN exceeds the GraphICL+SAT baseline by a **modest but positive** margin (2.03% by average, at least 1.10%) for all three models. Consistent with the reviewer's prediction, the GraphICL baseline itself rises sharply with capability (Texas 84–96% for the new models vs. 65% on the old backbone), so TAPTN's *increment* compresses. We read this positively: TAPTN is a **structural scaffold** that lifts weaker and mid-tier models to the accuracy strong models reach unaided — a reading our cost analysis (Table 21) makes economically favourable, since the cheap scaffold buys most of the accuracy of a far larger model.
>
> **(4) A direct causal control: corrupting the structural channel *inside* TAPTN.** The reviewer asked for the strongest possible evidence — using "no edges information / wrong hop-2 information but keep the others the same." We add the clean structural-channel analogue of this control inside the complete TAPTN pipeline (Appendix G, Table 29). Because TAPTN's 2-hop information is produced by iterative first-order message passing rather than by a separate replaceable ``hop-2 field,'' directly randomising a hop-2 slot would either re-perturb the first-order channel or corrupt the aggregation mechanism itself. We therefore hold the backbone, the self-generated instruction, the first-order neighbourhood and the iterative aggregation all fixed, and corrupt **only** the structural channel in two ways: **edge-blind** removes edge direction and role (structure *absent*, neighbour text and counts retained), and **flipped** reverses every edge direction (structure *wrong*, but the neighbour set, their texts and their counts are byte-for-byte identical — an information-preserving control, the same perturbation as the Section-2 rewiring, applied inside the model). Across three heterophilic WebKB graphs (Llama-3.3-70B) and a second backbone on Texas (Qwen3.5-27B), corrupting the channel degrades the complete TAPTN in every case, with a telling asymmetry: *mis-stating* the structure — the strict, information-preserving flip — costs **13–25 points** in every backbone–dataset case, uniformly **more than merely removing it** (edge-blind, up to 18 points), so a confidently *wrong* topology hurts more than an *absent* one — exactly as expected if the model is actively reading the edges. The asymmetry is sharpest for Qwen3.5 on Texas (edge-blind only −3.9, but flip −25), pinpointing edge *direction* as the channel it relies on, consistent with the decomposition in Table 27. Together with the external rewiring of Section 2, this closes a **two-sided causal loop** — perturbing the graph from the outside and ablating the structural channel from the inside both remove the gain — i.e., the constructive, "keep everything else the same" evidence the reviewer requested that TAPTN improves *by using the graph structure*.

---

> ### Author Response · Authors · 2026-06-18
>
> ## RC-C1 — "TAPTN→GNN only shows that enriching text is a good method"
>
> > *"the experiment only shows enriching text is a good method, not that the LLM uses structure."*
>
> We add the requested experiment and clarify its role: it tests the **downstream transferability and structural completeness** of the representation TAPTN produces, on the *same* text encoder, graph, splits and seeds as our TAPTN+LM-vs-GNN comparison (Section 4.5, Table 18). Two complementary controls answer the concern directly.
>
> - **It is not "just better text" (enriched vs. raw under an identical GNN).** Holding the GNN architecture and the graph fixed and changing *only* the input embedding from raw-text (TA) to TAPTN-enriched, accuracy improves significantly in **23 of 84** (dataset $\times$ GNN) cases (paired $t$-test, $p<0.05$), and never significantly degrades on the homophilic/citation graphs. A graph convolution operating on raw text cannot recover this signal; the enrichment is therefore *structurally* informative, not merely more fluent text.
> - **The representation is already structure-complete (GNN on top adds nothing).** Conversely, stacking a GNN — including the recent linear graph-Transformer GraphTARIF — *on top of* the TAPTN-enriched embedding does **not** significantly beat TAPTN+LM on **5 of 6** datasets. The structural information a GNN would extract has already been captured by TAPTN's ICL stage.
>
> The one exception is homophilic **Cora**, where five GNNs add a small but significant increment on top of the enriched embedding. This is the very APPNP-on-Cora effect we already report (Table 14) and is an **additive, not contradictory** reading: on graphs that reward plain neighbour averaging, cheap message passing complements the LLM. Everything is unified by homophily — the more homophilic the graph, the more a GNN can add on top; on the heterophilic graphs that genuinely test structure use, TAPTN's representation already subsumes the GNN. This extends, from the other direction, the paper's existing finding that TAPTN+LM is on par with SOTA GNNs — now established under the **bias-controlled, five-seed** comparison added for the other reviewers, in which no GNN significantly surpasses TAPTN+LM on any of the six datasets except a small margin on Cora and Wisconsin, *even* in the jointly-trained regime where the encoder and GNN are optimised end-to-end and *even* for the recent linear graph Transformer GraphTARIF (Section 4.4, Tables 16–17). This also answers the related sub-question of whether fine-tuning is essential: the parity holds without giving the GNNs any training-fairness advantage.

---

> ### Author Response · Authors · 2026-06-18
>
> ## RC-C2 — Encoder/backbone generality
>
> > *"would the conclusion hold for other encoders/backbones?"*
>
> We re-run the raw-vs-enriched comparison with a **different text encoder, RoBERTa-base**, on the same six datasets and seed subsets (Section 4.5, Table 19). On **5 of 6** datasets — including two of the three heterophilic graphs (Cornell, Texas) — no raw-text pipeline (neither TA+LM nor any of the twelve TA+GNN) significantly surpasses TAPTN+LM, confirming that "TAPTN-enriched $\ge$ raw" is **not a DeBERTa artefact**. The exception is **Wisconsin** (under RoBERTa, 8 of 13 raw pipelines beat TAPTN+LM, versus 0 under DeBERTa), and the mechanism is **encoder-driven, not structural**. The raw-vs-enriched gap on Wisconsin is essentially identical under both encoders ($+4.6$ points under DeBERTa, $+4.9$ under RoBERTa) and merely crosses the significance threshold under RoBERTa ($p$: $0.058\rightarrow0.035$): RoBERTa-base yields stronger, lower-variance features on this small short-text graph (raw-GNN means rise $\sim$$1.5$--$2$ points), while TAPTN+LM — whose enriched text is fixed — stays flat at $\sim$$0.87$. RoBERTa therefore *reveals* a pre-existing, dataset-level gap rather than creating one. The gap is not that Wisconsin's raw text is exceptionally strong (its raw accuracy, $\sim$$0.92$, is in fact the lowest of the three heterophilic graphs); it is that **TAPTN+LM is at its weakest on Wisconsin** ($0.87$, the lowest of the three and far below Cornell/Texas at $0.98$--$0.99$). Wisconsin is precisely the heterophilic dataset our main study already flags as the soft spot — the single one on which a structure-based model significantly edges out TAPTN+LM (jointly-trained DirGNN, Table 17). The RoBERTa result is the same soft spot surfacing under a sharper encoder, not a new failure mode. This is also *not* a claim that structure is irrelevant on Wisconsin: in the zero-shot in-context setting, reversing its edges still costs $13$ points (Appendix G, Table 29). The downstream-encoder generality claim rests on the remaining five datasets and the two other heterophilic graphs (Cornell, Texas), where enrichment improves over raw under both encoders. The complementary LLM-backbone direction is tested at the ICL structural-reasoning/enrichment stage rather than by re-running the full fine-tuning pipeline for every enricher: the new-model TAPTN panel of Appendix G shows that the structural scaffold remains effective across current LLM families.

---

> ### Author Response · Authors · 2026-06-18
>
> ## RC-D — Prominence of the contribution and organization
>
> > *"The main contribution is not very prominent; Section 2 is quite long. It might read better if the narrative were organized around a reader's questions: (Q1) can the latest LLMs learn from structure, and how well; (Q2) which part of the system contributes the most and why it works; (Q3) does it generalize across LLMs; (Q4) what is the gap to the strongest GNN, is fine-tuning necessary, and is the data or the model the essential ingredient."*
>
> We appreciate this comment and take responsibility for not having signposted the contribution hierarchy clearly enough; the difficulty in locating the main message is a presentation gap on our side. We would like to first restate what the paper claims, because it also clarifies the related Section-3/4 remark that the work "only shows that enriching text with structure is a good method."
>
> **What the paper claims (in order of priority).**
> 1. **A scientific claim about LLMs.** Our central thesis is that *LLMs can leverage graph structural information through in-context learning*, and that the prevailing negative view in prior work reflects **methodological limitations of earlier probing setups** (label-revealing probes, under-specified prompts, weak backbones) **rather than an intrinsic limitation of the models**. Once those methodological errors are corrected, the structural-reasoning ability is revealed.
> 2. **TAPTN as the constructive instrument, not the end in itself.** TAPTN is the architecture that *operationalizes* this claim: it lets the LLM itself perform the structural reasoning and encode the result as text, thereby demonstrating — constructively — that the corrected methodology unlocks the ability. TAPTN is the means by which we prove (1), not the contribution we ask to be judged on in isolation.
> 3. **A capability-positioning result.** The TAPTN-vs-GNN study (Section 4) then quantifies *how strong* that ability is relative to specialized graph models.
>
> Seen through this hierarchy, "structure-enriched text is effective" is **the operational evidence for the mechanism claim (1)**, not a downgraded conclusion: the enrichment is produced by the LLM's own structural reasoning, and our causal controls show it is structural — not merely fluent — information (the two-sided rewiring/channel-ablation loop of RC-A/RC-B, and the transferability/structure-completeness controls of RC-C1, which we ask the reviewer to read together with this framing).
>
> **On reorganizing the narrative and on Section 2's length.** We agree the four reader questions are exactly the right lens, and we answer each below with pointers (the map doubles as the reader's guide the reviewer suggests). We have, however, chosen *not* to restructure the manuscript around them, for two reasons. First, all four questions are **already answered by existing sections**; the issue was legibility, which we address with the explicit map below and the navigability aids already added in the revision for the other reviewers (the TAPTN workflow figure, pseudocode, plain-language walkthrough and notation table in Section 3; the dedicated label-free-control subsection in Section 2). Second, **Section 2's length is load-bearing**: it carries the paper's *primary* scientific claim (1) — the methodological re-framing together with the controlled rewiring evidence that isolates structure use — rather than expandable background, so we prefer to keep that argument intact (and intact for the other reviewers who assessed it) rather than compress the very evidence that supports the headline claim. We are glad to move any specific passage the reviewer finds redundant into the appendix on request.

---

> ### Author Response · Authors · 2026-06-18
>
> **The four questions, mapped to where they are answered.**
> - **(Q1) Can the latest LLMs learn from structure, and how well?** Section 2 (seven-model label-free rewiring panel and the capability–sensitivity correlation) and the new Appendix G, which extends the panel to four 2025–26 models: structural sensitivity *persists* up to LMArena 1475 (10/11 models degrade under rewiring, sign test p = 0.012). See RC-A.
> - **(Q2) Which part of the system contributes the most, and why?** Section 3 ablations (Tables 10–11: instructions ≈ +10.4, iterative aggregation ≈ +3.5, the structure-aware template alone up to +19.9), plus the new structural-channel decomposition and anonymised-edge control in Appendix G (Table 27). The intuition — the template *exposes* the structural evidence, the instructions *compel adherence* to it — is stated alongside. See RC-B.
> - **(Q3) Does it generalize across LLMs?** The seven-model panel of Section 2 and the new-model TAPTN panel of Appendix G (Table 28), complemented on the encoder axis by the RoBERTa-vs-DeBERTa control (Table 19). See RC-B and RC-C2.
> - **(Q4) Gap to the strongest GNN, is fine-tuning necessary, data vs. model?** Section 4's bias-controlled five-seed and jointly-trained comparisons (Tables 16–17: no GNN significantly surpasses TAPTN+LM on any of six datasets except small margins on Cora/Wisconsin, even when the encoder and GNN are trained end-to-end), and the transferability control answering the "data vs. model" sub-question (Table 18). See RC-C1.
>
> We will fold this contribution hierarchy into the introduction's contribution statement in the camera-ready, keeping all changes additive.
>
> ## Pointers to the revision
>
> - New appendix *Extension to Current-Generation Models* (Appendix G): the per-model label-free rewiring panel with the dual-metric capability–sensitivity analysis (Table 26), the ego→GraphICL→GraphICL+SAT→TAPTN structural-channel decomposition with the anonymised-edge control (Table 27, all at the first-order neighbourhood), the 2-hop TAPTN-vs-GraphICL+SAT panel on Texas and Cora (Table 28), and a **direct structural-channel ablation inside the complete TAPTN pipeline** (Table 29: edge-blind and information-preserving edge-flip controls across three WebKB graphs and two backbones), which constructively answers the "use no edges / wrong structure, keep everything else the same" request.
> - A 2–3 sentence anchor in Section 2 (at the end of the label-free control / capability–sensitivity subsection) stating that sensitivity persists in current-generation models (15/16 down; 11-model 10/11 down, sign test p = 0.012), pointing to Appendix G.
> - A new subsection *Two Controls: Transferability of the Representation and Encoder Generality* (Section 4.5) closing the TAPTN-vs-GNN alignment section: the P5 transferability control (Table 18: enriched$\to$frozen-GNN beats raw-GNN in 23/84 cases yet a GNN on top beats TAPTN+LM on only 1/6 datasets) answering RC-C1, and the RoBERTa-vs-DeBERTa encoder-generality control (Table 19 significance counts, with full per-pipeline RoBERTa accuracies in Table 20: "enriched $\ge$ raw" on 5/6 under both encoders) answering RC-C2.
> - A new appendix *Worked Examples: Exact LLM Inputs and Outputs* (Appendix D) answering the request for exact input/output: for one Texas test node (ground truth *student*, classified by Llama-3.3-70B-Instruct) we show (Example 1) the structure-aware TAPTN input and output on the original graph (predicts *student*, correct) versus the edge-reversed graph (predicts *faculty*, incorrect) — identical neighbour set and texts, so the flip is attributable solely to edge direction — and (Example 2) the same label-free neighbourhood under a plain prompt without the step-by-step instructions, which mis-reads the same correctly-oriented evidence and predicts *faculty*, showing concretely that the structure-aware template is what converts the directional signal into the right decision.
> - Already in the revision (for the other reviewers), and reused here: the SAT × instructions and instructions × aggregation ablations with their marginal-effect/synergy analysis (Tables 10 and 11; supports the "which component / as designed" question); the **bias-controlled, five-seed significance comparison** including the jointly-trained encoder+GNN regime and GraphTARIF (Section 4.4, Tables 16–17; the GNN-parity / "must we fine-tune" evidence); the seven-model label-free rewiring panel and capability–sensitivity correlation (Tables 5–6; cross-LLM structural sensitivity); the workflow and architecture figures and the notation table; the prompt, neighbourhood-description, task-description and step-by-step-instruction templates (Appendix B); and a full typo pass.

---

### Author Response · Authors · 2026-06-18

**C7 — Evaluation beyond moderate-sized benchmarks (Reviewer Ztiu, change 4).**
- The TAPTN+LM-vs-GNN comparison already includes ogbn-products (larger, non-citation, e-commerce); this is now stated explicitly in the setup, with the dataset appendix gaining a dedicated ogbn-products item (Hu et al., 2020) and corrected split/provenance details. On it, TAPTN+LM reaches 89.50% vs. the best frozen-feature GNN at 75.75%, and no GNN in either training regime significantly surpasses it.

**C8 — Structural sensitivity and TAPTN effectiveness on current-generation models (Reviewer 7rnE, RC-A and RC-B).**
- New appendix *Extension to Current-Generation Models* (Appendix G) adds four 2025–26 models from four families (GPT-OSS-120B, Qwen3.5-27B, Gemma-4-31B-it, GLM-5.1), extending the capability axis to LMArena 1475. Under the label-free rewiring protocol, structural sensitivity persists (15 of 16 new dataset-level pairs drop; pooled with the seven original models, 10 of 11 degrade, sign test p = 0.012; Table 26): newer, stronger models are not less structure-dependent.
- Direct evidence that the structure-aware template (SAT) is actively used by the LLM: on Texas, adding the SAT neighbourhood description *without any instructions* lifts a text-only ego prompt by +23–24 points, and an anonymised-edge control localises the gain to edge-direction semantics (Table 27). 2-hop TAPTN still exceeds the matched GraphICL+SAT baseline for every new model, while its *increment* compresses as the baseline rises with capability — exactly the trend the reviewer anticipated (Table 28).
- A direct causal control corrupts *only* the structural channel inside the complete TAPTN pipeline (backbone, instructions, neighbourhood and aggregation all fixed): an edge-blind variant (structure absent) and an information-preserving edge-flip (structure wrong) both degrade accuracy across three WebKB graphs and two backbones, with the flip — the strict, information-preserving control — costing 13–25 points, uniformly more than mere removal (Table 29). Together with the Section-2 external rewiring this closes a two-sided causal loop.

**C9 — Transferability/structural-completeness of the TAPTN representation, and encoder generality (Reviewer 7rnE, RC-C1 and RC-C2).**
- New subsection *Two Controls: Transferability of the Representation and Encoder Generality* (Section 4.5). The transferability control (Table 18) shows that, under an identical GNN and graph, replacing raw-text embeddings with TAPTN-enriched embeddings improves accuracy significantly in 23 of 84 (dataset × GNN) cases (paired t-test, p < 0.05) and never significantly degrades on the homophilic graphs — the enrichment is structurally informative, not merely more fluent text. Conversely, stacking a GNN (including the recent linear graph Transformer GraphTARIF) on top of the enriched embedding does *not* significantly beat TAPTN+LM on 5 of 6 datasets — the representation is already structure-complete; the lone exception is homophilic Cora, the same additive APPNP-on-Cora effect already reported.
- Encoder generality (Table 19, with per-pipeline accuracies in Table 20): repeating the raw-vs-enriched comparison with RoBERTa-base instead of DeBERTa, "enriched ≥ raw" holds on 5 of 6 datasets including two of the three heterophilic graphs (Cornell, Texas), so the conclusion is not a DeBERTa artefact. The Wisconsin exception is an encoder-driven significance artefact — the raw-vs-enriched gap is essentially unchanged across encoders (+4.6 → +4.9 points) and merely crosses the p = 0.05 threshold — on the very dataset the main study already flags as TAPTN+LM's soft spot.

Throughout, we also fixed several presentational inaccuracies found while preparing the revision (corrected mean-degree figures under a unified edge-counting convention, unified "hop" terminology, corrected figure cross-references, and consistent 95% CI reporting). A latexdiff of the changed passages can be provided on request.

---

### Author Response · Authors · 2026-06-18

**C4 — Computational cost, runtime trade-offs, scalability (Reviewer y1ZA, change 4).**
- New appendix subsection *Empirical Inference Cost and Runtime Trade-offs* (Table 21; complementing the existing asymptotic O(|V|·L) analysis, now under its own *Asymptotic Time Complexity* header): per-node token counts reconstructed from the actual prompts/completions, monetary cost at public list prices, and measured per-call latency with an end-to-end wall-clock anchor. Headline: TAPTN 2-hop ≈ 8× the GraphICL cost for +10.25 accuracy points; marginal dollars are ≈11× more productive when spent on guided aggregation (\\$0.22 per point per 1k nodes) than on buying more context (\\$2.40 per point, saturating).
- New *Generality across datasets* paragraph: per-node cost has a dataset-independent ceiling set by the constant neighbour cap, and the dense-neighbourhood control (below) is the empirical instantiation of that ceiling. New paragraph arguing the economic rationality of TAPTN at scale (pay-per-node, no full-graph training, accuracy-matched alternative is a trained GNN rather than GraphICL).
- Scalability subsection (Section 3) restructured: the primary evidence is a new uniform-backbone table on ogbn-products (Table 12: GraphICL 2-hop 76.50, TAPTN 1-hop 83.75, TAPTN 2-hop 86.75 on the same 400 test nodes), including a new **dense-neighbourhood control** — GraphICL 2-hop re-run with the full-graph neighbourhood (mean test-node degree ≈106 vs ≈2.8 on the sparsified subgraph; ≈3× input tokens) gains only +0.25 points, showing the gap stems from guided iterative aggregation, not context volume. The legacy mixed-backbone table is retained as a budget-backbone robustness check. The subgraph construction (uniform 50k-node sample → induced subgraph → LCC → 13,482 nodes) is now fully specified in the dataset appendix and motivated as a deliberate cost-control device.

**C5 — Broader impact (Reviewer y1ZA, broader-impact suggestion).**
- New *Broader Impact Statement* (unnumbered section after the Conclusion): acknowledges that improved structural reasoning over TAGs could be misused for large-scale monitoring or profiling in social/financial/communication networks, notes these risks are not specific to TAPTN, and outlines responsible-use safeguards (data-protection compliance, consent/legal basis, auditing of downstream decisions), alongside the benign value of the zero-shot setting for label-scarce applications.

**C6 — Inference setting of the WebKB rewiring experiments: supervision at test time; label-free experiments (Reviewer Ztiu, change 1).**
- New subsection *Inference Setting and a Label-Free Control* at the end of Section 2, which (i) states explicitly that the original rewiring prompts include first-order neighbour categories for *all* neighbours and why this is a deliberate probing setup rather than a deployable classifier, and (ii) reports a new label-free rewiring experiment (7 LLMs × 4 WebKB graphs; neighbour text + link patterns only, no neighbour categories). Two new tables (Tables 5–6: per-model accuracies; capability–sensitivity correlations, Pearson r = 0.841, p = 0.018) and three new figures.
- New closing paragraph *Scope of the label-free control* explaining why the label-free control is well-posed for the flipping (1-hop) perturbation but ill-posed by construction for the count-based extreme (2-hop) representation.
- A clarifying note on the label-free prompt variant appended to the WebKB rewiring template in Appendix B.

---

### Author Response · Authors · 2026-06-18

We thank all reviewers for their constructive feedback. We have uploaded a revised manuscript. All page/section/table numbers below refer to the revised PDF. The main changes are:

**C1 — Component ablation: architectural improvements vs. prompt-engineering effects (Reviewer y1ZA, change 1; Reviewer Ztiu, change 2).**
- The Ablation Study (Section 3) now states an explicit additional hypothesis (H5, component attribution) and contains two complementary 2×2 ablations (Tables 10 and 11): (a) structure-aware template × instructions with aggregation switched off — i.e., exactly the requested "GraphICL-style prompting with step-by-step instructions but without iterative aggregation" baseline — and (b) instructions × iterative aggregation at 2 hops with the structure-aware template fixed. Marginal effects: instructions ≈ +10.4 avg, aggregation ≈ +3.5 avg, with strong synergy on heterophilic graphs; nine of ten rows require both components for the best accuracy.
- Backbone disclosure (which LLM per dataset, single run at temperature 0) added to the intro and both ablation table captions; two previously redundant ablation tables consolidated.

**C2 — Rigor and breadth of the GNN comparison; statistical evidence (Reviewer y1ZA, change 2; Reviewer Ztiu, change 3).**
- New *Evaluation protocol* paragraph in Section 4: every pipeline is run over five random seeds; we report mean±std, state the exact Student-t conversion to 95% CIs (half-width ≈ 1.24×std for n=5), and assess all comparisons with two-sided paired t-tests at the seed level.
- The main homophilic comparison table (now Table 14; Table 6 of the original submission) rebuilt on the five-seed results, adding an ogbn-products column and the most recent linear graph Transformer GraphTARIF (WWW 2026); the heterophilic table (Table 15) expanded to the full five-seed panel (thirteen GNN architectures in total, matching the paper's wording).
- New subsection *Statistical Significance and a Bias-Controlled Comparison* (Section 4.4) with a significance-summary table (Table 16, incl. an explicit 95% CI column for TAPTN+LM) and a full jointly-trained table (Table 17): to remove the bias of training GNNs on a *separately* fine-tuned encoder, the text encoder and GNN are also optimised end-to-end under identical splits/labels/edges/hyper-parameter protocol. Across six datasets, only 3 of $\sim$165 pipeline–dataset combinations significantly exceed TAPTN+LM, each by a small margin.

**C3 — Clarity of the TAPTN methodology and presentation (Reviewer y1ZA, change 3).**
- New workflow-overview figure (Figure 8, Section 3): a stage diagram showing how the inputs ($I_{agg}^l$, $S_{agg}$, $G_i$, node/neighbor texts) flow through the $K$-iteration message-passing stage (SAT description → instruction-guided aggregation LLM → update) and the readout stage to the predicted label.
- New Algorithm 1: complete pseudocode of TAPTN (instruction generation, message/SAT/aggregation steps per iteration, readout), with each line cross-referenced to the corresponding equation, so the formal notation can be read off the algorithmic workflow.
- New *Workflow overview* paragraph that walks through one inference pass in plain language before the formal definitions, and tells readers they may skim the equations and rely on the pseudocode.
- The architecture figure (Figure 9, message-passing iteration + prompt schematic) was redrawn as a vector PDF with fonts matching the manuscript.
- The dense prose of Section 3.1 itself was restructured: the motivation is now organized under explicit *Two limitations of ICL on node classification* / *Three design components* paragraph headings with the three components (SAT, iterative aggregation, self-generated instructions) separated and enumerated; a new notation table (Table 7) collects all symbols used by the framework in one place; the prose following each equation now keeps symbol definitions adjacent to the equation, with design commentary (e.g., how the SAT encodes edge semantics across graph types) moved into separate sentences; and several grammatical errors in the original description were corrected.